# Graphical-model framework for automated annotation of cell identities in dense cellular images

**Shivesh Chaudhary[1], Sol Ah Lee[1], Yueyi Li[1], Dhaval S Patel[1], Hang Lu[1,2]\***

[1]School of Chemical & Biomolecular Engineering, Georgia Institute of Technology, Atlanta, United States; [2]Petit Institute for Bioengineering and Bioscience, Georgia Institute of Technology, Atlanta, United States

**Abstract** Although identifying cell names in dense image stacks is critical in analyzing functional whole-brain data enabling comparison across experiments, unbiased identification is very difficult, and relies heavily on researchers' experiences. Here, we present a probabilistic-graphical-model framework, CRF_ID, based on Conditional Random Fields, for unbiased and automated cell identification. CRF_ID focuses on maximizing intrinsic similarity between shapes. Compared to existing methods, CRF_ID achieves higher accuracy on simulated and ground-truth experimental datasets, and better robustness against challenging noise conditions common in experimental data. CRF_ID can further boost accuracy by building atlases from annotated data in highly computationally efficient manner, and by easily adding new features (e.g. from new strains). We demonstrate cell annotation in *Caenorhabditis elegans* images across strains, animal orientations, and tasks including gene-expression localization, multi-cellular and whole-brain functional imaging experiments. Together, these successes demonstrate that unbiased cell annotation can facilitate biological discovery, and this approach may be valuable to annotation tasks for other systems.

**\*For correspondence:**
hang.lu@gatech.edu

**Competing interests:** The authors declare that no competing interests exist.

## Introduction

Annotation of anatomical structures at cellular resolution in large image sets is a common data analysis step in many studies in *Caenorhabditis elegans* such as gene expression pattern analysis (*Long et al., 2009*; *Murray, 2008*), lineage tracing (*Bao et al., 2006*), multi-cell calcium imaging and whole-brain imaging (*Schrödel et al., 2013*; *Kato et al., 2015*; *Venkatachalam et al., 2016*; *Nguyen et al., 2016*). It is necessary for cellular resolution comparison of data across animals, trials, and experimental conditions. Particularly in whole-brain functional imaging, meaningful interpretation of population activity critically depends on cell identities as they facilitate the incorporation of existing knowledge about the system (*Kato et al., 2015*). Cell identities are also needed for applying common statistical data analysis methods such as Principal Component Analysis, Tensor Component Analysis, demixed-Principal Component Analysis (*Williams et al., 2018*; *Kobak et al., 2016*) etc as data across experiments needs to be indexed and pooled by cell identities before applying these methods.

While accurate annotation of cell identities in images is critical, this task is difficult. Typically, the use of cell-specific markers as landmarks delivers good accuracy, but has the cost of having to engineer cell-specific reagents without interfering with phenotypes of interest, which is not guaranteed. Further, even with markers such as the recently developed impressive reagents in the NeuroPAL collection (*Yemini et al., 2021*), there is still a need to automate the cell identification process. In the absence of markers, cells are identified by comparing images to a reference atlas such as WormAtlas (*Altun and Hall, 2009*) and OpenWorm (*Szigeti et al., 2014*) atlas. However, there are severe limitations from both using reference atlas and the presence of noise in data. Reference atlases assume a

static and often single view of the anatomy; in contrast, anatomical features vary across individuals. Moreover, due to variations in experimental conditions during acquisition such as exact resolution and orientation of animals, image data often do not match the static atlases, making manual cell identification extremely difficult if not infeasible. Separately, two kinds of noise are prevalent in data. First, individual-to-individual variability in cell positions compared to positions in atlas (position noise). Second, mismatch between number of cells in image and atlas (count noise). Count noise is primarily caused by variability in the expression levels of the reporter used to label cells across animals (i.e. mosaicism), incomplete coverage of promoter to label desired cells, and limits in the computational methods to detect cells. In each of these cases, fewer cells are detected in the images than cells in the atlas. Empirical data have shown that in normalized coordinates, a cell's position can deviate from the atlas position by more than the cell's distance to its tenth' nearest neighbor in the image (*Yemini et al., 2021*; *Toyoshima et al., 2016*). Further, our data, as well as data from other labs, have shown that 30–50% of cells in atlases may be missing from images (*Kato et al., 2015*; *Venkatachalam et al., 2016*; *Nguyen et al., 2016*). As a result of the large position and count noise common in data, identifying densely packed cells in head ganglion images of *C. elegans* by manually comparing images to the atlas is extremely difficult, even for experienced researchers. Further, manual annotation is labor intensive. Therefore, there is a critical need for automated methods for cell identification.

Previous computational methods for cell identification in *C. elegans* images (*Long et al., 2009*; *Long et al., 2008*; *Qu et al., 2011*; *Aerni et al., 2013*) focused on identifying sparsely distributed cells with stereotypical positions in young larvae animals. Tools for identification of cells in whole-brain datasets, that is in dense head ganglion, do not exist. Further, previous methods (*Long et al., 2009*; *Yemini et al., 2021*; *Qu et al., 2011*; *Aerni et al., 2013*; *Toyoshima, 2019*; *Scholz, 2018*) do not explicitly address the challenges imposed by the presence of position and count noise in the data. All previous methods either are registration-based or formulate a linear assignment problem; objective functions in these methods minimize a first-order constraint such as the distances between cell-specific features in images and atlases. Thus, these methods maximize only extrinsic similarity (*Bronstein et al., 2007*) between images and atlas, which is highly sensitive to count noise, position noise, and pre-alignment of spaces in which the image and the atlas exist (i.e. orientations of animals in images and atlases). With the amount of position and count noise commonly observed in experimental data, registration-based methods produce large matching errors.

An alternative criterion proposed for topology-invariant matching of shapes is to maximize intrinsic similarity (*Bronstein et al., 2007*; *Bronstein et al., 2009*), orthogonal to extrinsic similarity. This approach has advantages because noise that affects extrinsic similarity does not necessarily imply worse intrinsic similarity. For instance, although cell positions in an image may deviate from their positions in the atlas (large extrinsic noise), geometrical relationships among them are largely maintained (low intrinsic noise). As a specific example, although absolute positions of the cell bodies of AIBL and RIML in an image may deviate greatly from their atlas positions, AIBL soma stays anterior to RIML soma. Therefore intrinsic similarity is more robust against noises, independent of the pre-alignment of spaces, and inherently captures dependencies between cell label assignments that registration methods do not consider.

To directly optimize for intrinsic similarity and dependencies between label assignments, we cast the cell annotation problem as a Structured Prediction Problem (*Bakir, 2007*; *Nowozin, 2010*; *Caelli and Caetano, 2005*; *Kappes et al., 2015*) and build a Conditional Random Fields (CRF) model (*Lafferty et al., 2001*) to solve it. The model directly optimizes cell-label dependencies by maximizing intrinsic and extrinsic similarities between images and atlases. One major advantage, as shown using both synthetic data with realistic properties (e.g. statistics from real data) and manually annotated experimental ground-truth datasets, is that CRF_ID achieves higher accuracy compared to existing methods. Further, CRF_ID outperforms existing methods in handling both position noise and count noise common in experimental data across all challenging noise levels.

To further improve accuracy, we took two approaches. First, we took advantage of spatially distributed (fluorescently labeled) landmark cells. These landmark cells act as additional constraints on the model, thus aiding in optimization, and helping in pre- as well post-prediction analysis. Second, we developed a methodology to build data-driven atlases that capture the statistics of the experimentally observed data for better prediction. We provide a set of computational tools for automatic and unbiased annotation of cell identities in fluorescence image data, and efficient building of data-

driven atlases using fully or partially annotated image sets. We show the utility of our approach in several contexts: determining gene expression patterns with no prior expectations, tracking activities of multiple cells during calcium imaging, and identifying cells in whole-brain imaging videos. For the whole-brain imaging experiments, our annotation framework enabled us to analyze the simultaneously recorded response of *C. elegans* head ganglion to food stimulus and identify two distinct groups of cells whose activities correlated with distinct variables – food sensation and locomotion.

## Results

### Cell annotation formulation using structured prediction framework

Our automated cell annotation algorithm is formulated using Conditional Random Fields (CRF). CRF is a graphical model-based framework widely used for structured/relational learning tasks in Natural Language Processing and Computer Vision community (*Bakir, 2007*; *Nowozin, 2010*; *Lafferty et al., 2001*; *Sutton and McCallum, 2010*). The goal of structured learning tasks is to predict labels for structured objects such as graphs. In our neuron annotation problem, we assume that our starting point is a 3D image stack of the *C. elegans* head ganglion (*Figure 1A(i)*) in which neurons have already been detected (*Figure 1A(ii)*), either manually or by automated segmentation, and we want to match each neuronal cell body or nucleus to an identity label (a biological name). Hence, we have $N$ detected neuronal cell bodies $\{x_1, \ldots, x_N\}$ that form the set of observed variables $\mathbf{x} = \{x_i\}_1^N$, and their 3D coordinates, $p_i \in \mathbb{R}^3$, $i \in \{1, \ldots, N\}$. We also have a neuron atlas that provides a set of labels $\mathcal{L} = \{l_1, \ldots, l_K\}$ (biological names) of the neurons and positional relationships among them. Note that the number of neurons in the atlas is greater than the number of neurons detected in the image stack in all datasets, that is $K > N$. The goal is to annotate a label $y_j \in \mathcal{L}$ to each neuron in the image stack. The problem is similar to structured labeling (*Nowozin, 2010*) since the labels to be assigned to neurons are dependent on each other. For example, if a certain neuron is assigned label AVAL, then the neurons that can be assigned label RMEL become restricted since only the cells anterior to AVAL can be assigned RMEL label.

We use CRF-based formulation to directly optimize for such dependencies and automatically assign names to each cell. Briefly, a node $v_i$ is associated with each observed variable $x_i$ (i.e. segmented neuron in image data) forming the set of variables $V = \{v_i\}_1^N$ in the model. Then, CRF models a conditional joint probability distribution $P(\frac{\mathbf{y}}{\mathbf{x}})$ over product space $\mathcal{Y} = \mathcal{Y}_1 \times \ldots \times \mathcal{Y}_N$ of labels assigned to $V$ given observations $\mathbf{x}$, where each $\mathcal{Y}_i = \mathcal{L}, i \in \{1, \ldots, N\}$ and $\mathbf{y} \in \mathcal{Y}$ is a particular assignment of labels to $V$. $\mathcal{Y}$ contains non-optimal and optimal assignments. In CRF, label dependencies among various nodes are encoded by the structure of an underlying undirected graph $\mathrm{G}(V, \mathcal{C})$ defined over nodes $V$, where $\mathcal{C}$ denotes the set of cliques in graph $\mathrm{G}$. With the underlying graph structure, the conditional joint probability distribution $P(\frac{\mathbf{y}}{\mathbf{x}})$ over full label space $\mathcal{Y}$ factorizes over cliques in $\mathrm{G}$, making it tractable to the model:

$$P\left(\frac{\mathbf{y}}{\mathbf{x}}\right) = \frac{1}{Z} \prod_c \Phi_c(y_c; \mathbf{x}) \tag{1}$$

Here, $Z$ is the normalization constant with $Z = \sum_{\mathbf{y} \in \mathcal{Y}} \prod_{c \in \mathcal{C}} \Phi_c(y_c; \mathbf{x})$ and $\Phi_c$ denotes clique potential if nodes in clique $c$ are assigned label $y_c \in \mathcal{Y}_c = \mathcal{Y}_1 \times \ldots \times \mathcal{Y}_c$. In our model, the fully connected graph structure considers only pairwise dependencies between every pair of neurons. Thus, the graph structure of our model becomes $\mathrm{G}(V, \mathcal{E})$, with the node set $V$ containing nodes $v_i$ associated with each segmented neuron and pairwise edges between all nodes that form the edge set $\mathcal{E}$. The potential functions in our model are node-potentials $\Phi_i$ and edge-potentials $\Phi_{ij}$. These functions are parameterized with unary feature functions $f_a : \mathbf{x} \times \mathcal{Y}_i \to R$ and pairwise feature functions $f_b : \mathbf{x} \times \mathcal{Y}_i \times \mathcal{Y}_j \to R$, respectively:

$$\Phi_i(m; i, \mathbf{x}) = \exp\left(\sum_a \lambda_a f_a(m; i, \mathbf{x})\right) \tag{2}$$

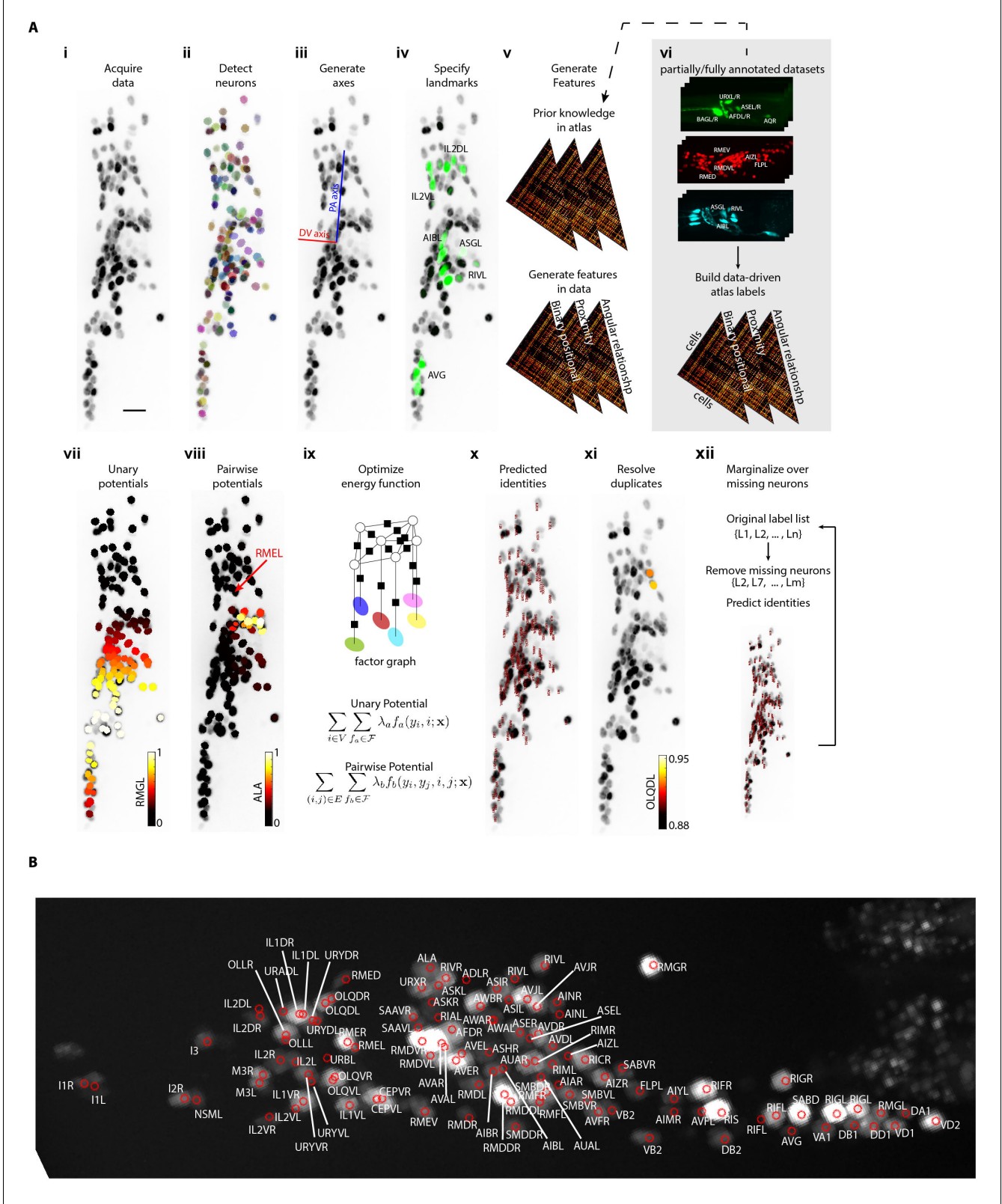

**Figure 1.** CRF_ID annotation framework automatically predicts cell identities in image stacks. (**A**) Steps in CRF_ID framework applied to neuron imaging in *C. elegans*. (i) Max-projection of a 3D image stack showing head ganglion neurons whose biological names (identities) are to be determined. (ii) Automatically detected cells (Materials and methods) shown as overlaid colored regions on the raw image. (iii) Coordinate axes are generated automatically (Note S1). (iv) Identities of landmark cells if available are specified. (v) Unary and pairwise positional relationship features are
*Figure 1 continued on next page*

*Figure 1 continued*

calculated in data. These features are compared against same features in atlas. (vi) Atlas can be easily built from fully or partially annotated dataset from various sources using the tools provided with framework. (vii) An example of unary potentials showing the affinity of each cell taking the label RMGL. (viii) An example of dependencies encoded by pairwise potentials, showing the affinity of each cell taking the label ALA given the arrow-pointed cell is assigned the label RMEL. (ix) Identities are predicted by simultaneous optimization of all potentials such that assigned labels maximally preserve the empirical knowledge available from atlases. (x) Predicted identities. (xi) Duplicate assignment of labels is handled using a label consistency score calculated for each cell (Appendix 1–Extended methods S1). (xii) The process is repeated with different combinations of missing cells to marginalize over missing cells (Note S1). Finally, top candidate label list is generated for each cell. (B) An example of automatically predicted identities (top picks) for each cell.

The online version of this article includes the following video and figure supplement(s) for figure 1:

**Figure supplement 1.** Schematic description of various features in the CRF model that relate to intrinsic similarity and extrinsic similarity.

**Figure supplement 2.** Additional examples of unary and pairwise potentials and label consistency scores calculated for each cell.

**Figure 1—video 1.** Identities predicted automatically by the CRF_ID framework in head ganglion stack.

https://elifesciences.org/articles/60321#fig1video1

$$\Phi_{ij}(m,n;i,j,\mathbf{x}) = \exp\left(\sum_b \lambda_b f_b(m,n;i,j,\mathbf{x})\right) \tag{3}$$

where $m \in \mathcal{L}$ and $n \in \mathcal{L}$ are labels in atlas. Note, there are $a$ unary features and $b$ pairwise features with weights $\lambda_a$ and $\lambda_b$ respectively to define node and edge potentials.

While unary features account for extrinsic similarity and cell-specific features, pairwise features account for intrinsic similarity. To maximize accuracy, we encode pairwise dependencies between all pairs of cells in the form of several geometrical relationship features (*Figure 1—figure supplement 1*). Optimal identities of all neurons $\mathbf{y} \in \mathcal{Y}$ is obtained by maximizing the joint-distribution $P(\frac{\mathbf{y}}{\mathbf{x}})$. This is equivalent to maximizing the following energy function.

$$y = \arg max_{y \in \mathcal{Y}} \sum_{i \in V} \sum_a \lambda_a f_a(m;i,\mathbf{x}) + \sum_{e_{ij} \in \mathcal{E}} \sum_b \lambda_b f_b(m,n;i,j,\mathbf{x}) \tag{4}$$

Optimizing this energy function over fully connected graphs (more specifically graphs with loops) is known to be an NP-hard problem (*Kohli et al., 2009*). However, approximate inference algorithms are widely used in CRF community as they provide reasonable solutions. We implemented a popular method called Loopy Belief Propagation (*Murphy et al., 1999*) to infer the most probable labeling over all cells, as well as marginal distributions of label assignments for each cell.

The features used in the base version of the model are geometrical relationship features that ensure identities assigned to cells in image are consistent with the atlas in terms of satisfying pairwise geometrical relationships. These features include binary positional relationship feature, proximity relationship feature, and angular relationship feature (*Figure 1—figure supplement 1*, Appendix 1–Extended methods S1.2). All these features are a variant of the quadratic Gromov-Wasserstein distance used in matching metric spaces (*Bronstein et al., 2009*; *Peyre et al., 2016*) and shapes (*Solomon et al., 2016*; *Mémoli, 2011*). Briefly, binary positional relationship features encode that as an example, if cell $i$ is anterior, dorsal and to the right of cell $j$ in image stack, then identities assigned to these cells should satisfy these relationships in the atlas. Proximity relationship features ensure that if cell $i$ is spatially near to cell $j$ in image stack, then identities of spatially distant cells in atlas would not be assigned to these cells. Finally, angular relationships ensure that identities assigned to cells $i$ and $j$ should satisfy fine-scale directional relationships as well, and not just simple binary relationships. We show that the CRF model can be easily updated to include additional features such as cells with known identities (landmark cells) and fluorescent spectral codes of cells. We demonstrate this by incorporating landmark cells and spectral information of cells in the model and show improvement in accuracy.

A critical component for the success of automated cell identification methods is data-driven atlas. Static atlases such as OpenWorm atlas provide a single observation of positional relationships among cells. For instance, if cell RMEL is to the left of cell AVAL in OpenWorm atlas, then the model assumes that RMEL is to the left of AVAL with 100% probability. In contrast, in observed experimental data RMEL may be observed to be left of AVAL with 80% probability (e.g. in 8 out of 10

annotated experimental datasets). Thus, data-driven atlases relax the hard-coded constraint of 100% probability imposed by static atlas and accounts for the statistics that is observed experimentally for all positional relationship features (*Figure 1—figure supplement 1*). Note, data-driven atlas built in our framework is considerably different from those built by registration-based methods. While the latter atlases store probabilistic positions of cells, atlases built by our framework store only probabilistic pairwise positional relationship features among cells, thus more generalizable. We show that building such data-driven atlases is easy for our CRF model (Appendix 1–Extended methods S1.7). We demonstrate this by building several data-driven atlases from different data sources containing various features, showing considerable improvement in accuracy. Further, building data-driven atlases is computationally cheap in CRF_ID, requiring only simple averaging operations; thus, it is scalable to build atlases from large-scale annotated data that may become available in future.

## Computational workflow for automatic cell identification

Our annotation framework consists of four major steps (*Figure 1A*; Appendix 1–Extended methods S1). First, cells are automatically detected in input image channels using a Gaussian Mixture-based segmentation method (see Materials and methods – Whole-brain data analysis). Cells with known identities (landmarks cells) are also detected in this step and their identities are specified. We designed the framework to be flexible on several fronts: (1) easily using manual segmentations of image channels or segmenting on the run; (2) integrating landmark information from any number of image channels; (3) specifying identities of landmark cells on the run or from existing fully or partially annotated files generated with other tools such as Vaa3D (*Peng et al., 2010*). In the second step, a head coordinate is generated by solving an optimization problem with considerations of the directional consistency of axes (see Appendix 1-Extended methods S1.3). With this coordinate system, we next define cell-specific features (unary potentials) and co-dependent features (pairwise potentials) in the data (*Figure 1—figure supplement 2A,B*). The base version of the model uses only pairwise relationship features for all pairs of cells, including binary positional relationships, angular relationship, and proximity relationship between cells in images (*Figure 1—figure supplement 1*). However, additional unary features such as landmarks and color information can be easily added in the model. By encoding these features among all pairs of cells, our fully connected CRF model accounts for label dependencies between each cell pair to maximize accuracy. The atlas used for prediction may be a standard atlas such as the OpenWorm (*Szigeti et al., 2014*) atlas or it can be easily built from fully or partially annotated datasets from various sources using the tools provided with our framework (see Appendix 1–Extended methods S1.7). In the third step, identities are automatically predicted for all cells by optimizing the CRF energy function consisting of unary and pairwise potentials, which in our formulation is equivalent to maximizing the intrinsic similarity between data and the atlas (see Appendix 1–Extended methods S1.4). Duplicate assignments are resolved by calculating a label-consistency score for each neuron, removing duplicate assignments with low scores (*Figure 1—figure supplement 2C,D*, see Appendix 1–Extended methods S1.5) and re-running the optimization. After the third step, the code outputs top predicted label for each cell. Next, an optional fourth step can be performed to account for missing neurons in image stack. In this step, full atlas is subsampled to remove fixed number of labels from atlas by either sampling uniformly or based on prior confidence values available on missing rate of labels in images (see Appendix 1–Extended methods S1.5). Subsampled atlas assumes that labels removed are missing from the image and thus ensures that those labels cannot be assigned to any cell in the image. The sampling procedure is repeated, and identities are predicted in each run. We perform these runs in parallel on computing clusters. Lastly, identities predicted across each run are pooled to generate top candidate identities for each cell (*Figure 1B*; *Figure 1—video 1*; see Appendix 1–Extended methods S1.6). Thus, there are two modes of running the framework – single-run mode that outputs only top label for each cell and parallel-run mode that outputs multiple top candidate labels for each cell. We make the software suite freely available at https://github.com/shiveshc/CRF_Cell_ID (*Chaudhary, 2021*; copy archived at swh:1:rev:aeeeb3f98039f4b9100c72d63de25f73354ec526).

## Prediction accuracy is bound by position and count noise in data

Given the broad utility of image annotation, we envision our workflow to apply to a variety of problems where experimental constraints and algorithm performance requirements may be diverse. For

example, experimental data across different tasks inherently contains noise contributed by various sources in varying amounts that can affect annotation accuracy. These sources of noises include the following: (1) deviation between cell positions in images and positions in atlases, which is position noise, (2) smaller count of cells in images than number of cells in atlas due to missing cells in images, which is count noise, and (3) absence of cells with known identities, i.e. known landmarks. We set out to determine general principles of how these noises may affect cell identification accuracy across various tasks. We used two different kinds of data: synthetic data generated from OpenWorm 3D atlas (*Szigeti et al., 2014*; *Figure 2—figure supplement 1A,B* and *Figure 2—figure supplement 2*) and experimental data generated using NeuroPAL strains (*Yemini et al., 2021*), consisting of annotated ground-truth of nine animals with ~100 uniquely identified neurons (*Figure 2—figure supplement 3*). While experimental data enables the assessment of prediction accuracy in real scenarios, synthetic data enable us to tune the amount of noise contributed from various sources and dissect their effects on accuracy independently.

To assess the effects of position noise and count noise on prediction accuracy, we simulated four scenarios using the synthetic data (*Figure 2—figure supplement 1C*). In the absence of any noise, relative positional relationship features predicted neuron identities with perfect accuracy (scenario one in *Figure 2—figure supplement 1C*), thus demonstrating the suitability of co-dependent features and CRF_ID framework for the annotation task. We found that both position noise and count noise affect accuracy significantly (*Figure 2—figure supplement 1C,D*) with position noise having a larger effect (compare scenarios 1–2 with 3–4 in *Figure 2—figure supplement 1C*). As mentioned before, count noise is primarily caused by inefficiencies of either the reporter used to label cells or inaccuracies of the cell detection algorithm used, thus leading to fewer cells detected in the images than cells in atlases. Results on both synthetic data and real data show that 10–15% improvement in prediction accuracy can be attained by simply improving reagents and eliminating count noise (*Figure 2—figure supplement 1D*). Next, we tested the effect of landmarks (cells with known identities) on annotation accuracy (*Figure 2—figure supplement 1E*). We hypothesized that landmarks will improve accuracy by acting as additional constraints on the optimization while the algorithm searches for the optimal arrangement of labels for non-landmark cells. Indeed, we found, in both experimental data and synthetic data, randomly chosen landmarks increased prediction accuracy by ~10–15%. It is possible that strategic choices of landmarks could further improve accuracy.

Another advantage of simulations using synthetic data is that by quantifying accuracy across the application of extreme-case of empirically observed noises, they can be used to obtain expected accuracy bounds for real scenarios. We obtained such bounds (shown as gray regions in *Figure 2—figure supplement 1F*) based on observed position noise in experimental data (*Figure 2—figure supplement 2*). Notably, the prediction results for experimental data lay close to the estimated bounds using synthetic data (*Figure 2—figure supplement 1F*). Together, good agreement between results obtained on synthetic and experimental data suggest that the general trends uncovered using synthetic data of how various noises affect accuracy are applicable to experimental data.

Next, with this knowledge, we tuned the features in the model, and we compared prediction accuracy for several combinations of positional relationship features. Among all co-dependent positional relationship features, the angular relationship feature by itself or when combined with PA, LR, and DV binary position relationship features performed best (*Figure 2—figure supplement 4A*). To account for missing cells, we developed a method that considers missing neurons as a latent state in the model (similar to hidden-state CRF *Quattoni et al., 2007*) and predicts identities by marginalizing over latent states (see Appendix 1–Extended methods S1.6). Compared to the base case that assumes all cells are present in data, simulating missing neurons significantly increased the prediction accuracy (*Figure 2—figure supplement 4B*) on experimental data.

## Identity assignment using intrinsic features in CRF_ID outperforms other methods

We next characterized the performance of our CRF_ID framework by predicting the identities of cells in manually annotated ground-truth datasets (*Figure 2A*). To specify prior knowledge, we built data-driven atlases combining positional information of cells from annotated ground-truth datasets and OpenWorm atlas (Appendix 1–Extended methods S1.7). To predict cell identities in each ground-truth dataset, separate leave-one-out atlases were built keeping the test dataset held out. Building such data-driven atlases for our framework is extremely computationally efficient, requiring simple

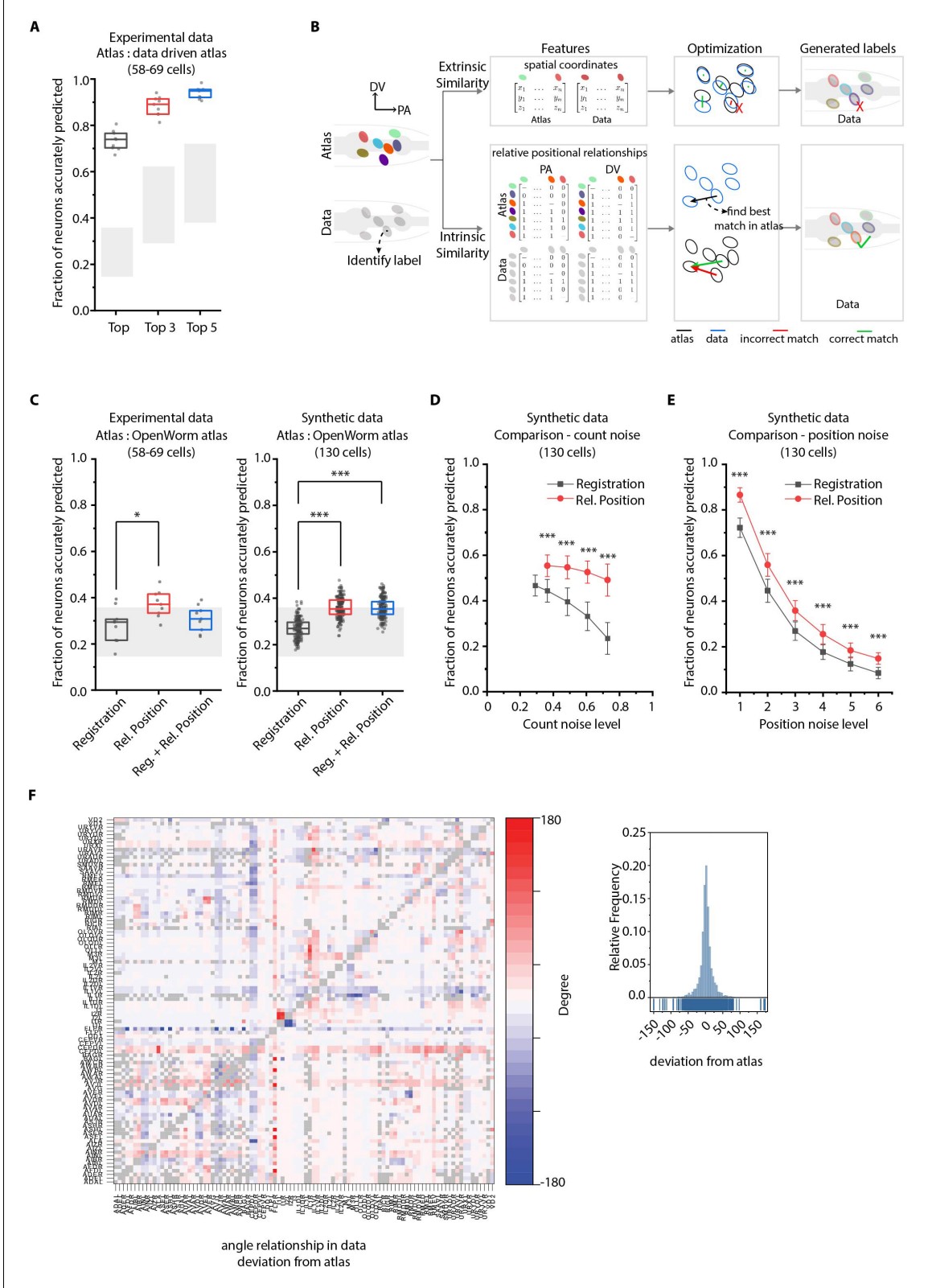

**Figure 2.** CRF_ID annotation framework outperforms other approaches. (**A**) CRF_ID framework achieves high prediction accuracy (average 73.5% for top labels) using data-driven atlases without using color information. Results shown for whole-brain experimental ground truth data (n = 9 animals). Prediction was performed using separate leave-one-out data-driven atlases built for each animal dataset with test dataset held out. Gray regions indicate bounds on prediction accuracy obtained using simulations on synthetic data (see *Figure 2—figure supplement 1F*). Experimental data comes

*Figure 2 continued on next page*

*Figure 2 continued*

from strain OH15495. Top, middle, and bottom lines in box plot indicate 75th percentile, median, and 25th percentile of data, respectively. (B) Schematic highlighting key difference between registration-based methods and our CRF_ID framework. (C) Prediction accuracy comparison across methods for ground truth experimental data (n = 9, *p<0.05, Bonferroni paired comparison test) and synthetic data (n = 190–200 runs for each method, ***p<0.001, Bonferroni paired comparison test). OpenWorm atlas was used for predictions. Accuracy results shown for top predicted labels. Experimental data comes from strain OH15495. For synthetic data, random but realistic levels of position and count noise applied in each run. Gray regions indicate bounds on prediction accuracy obtained using simulations on synthetic data (see *Figure 2—figure supplement 1F*). Top, middle, and bottom lines in box plot indicate 75th percentile, median, and 25th percentile of data, respectively. (D) Comparison of methods across count noise levels (defined as percentage of cells in atlas that are missing from data) using synthetic data. (n = 150–200 runs for Rel. Position for each noise level, n = ~1000 runs for Registration for each noise level, ***p<0.001, Bonferroni paired comparison test). OpenWorm atlas was used for prediction. Accuracy results shown for top predicted labels. For a fixed count noise level, random cells were set as missing in each run. Markers and error bars indicate mean ± standard deviation. (E) Comparison of methods across position noise levels using synthetic data. (n = 190–200 runs for each method for each noise level, ***p<0.001, Bonferroni paired comparison test). OpenWorm atlas was used for prediction. Accuracy results shown for top predicted labels. For a fixed position noise level, random position noise was applied to cells in each run. Different noise levels correspond to different variances of zero-mean gaussian noise added to positions of cells (see section Materials and methods – Generating synthetic data for framework tuning and comparison against other methods). Noise levels 3 and 6 correspond to the lower bound and upper bound noise levels shown in *Figure 2—figure supplement 1F*. Markers and error bars indicate mean ± standard deviation. (F) Pairwise positional relationships among cells are more consistent with OpenWorm atlas even though the absolute positions of cells vary across worms. (Left) average deviation of angular relationship measured in ground truth data (n = 9) from the angular relationship in static atlas. (Right) distribution of all deviations in left panel (total of 8516 relationships) is sparse and centered around 0 deviation, thus indicating angular relationships are consistent with atlas.

The online version of this article includes the following figure supplement(s) for figure 2:

**Figure supplement 1.** Performance characterization using synthetic data.
**Figure supplement 2.** Method of applying position noise to the atlas to generate synthetic data.
**Figure supplement 3.** Details of manually annotated experimental ground-truth datasets.
**Figure supplement 4.** Model tuning/characterization – features selection and simulating missing cells.
**Figure supplement 5.** CRF_ID framework with relative positional features outperforms registration method.
**Figure supplement 6.** Variability in absolute positions of cells and relative positional features in experimental data compared to the static atlas.
**Figure supplement 7.** Comparison of optimization runtimes of CRF_ID framework with a registration method CPD (*Myronenko and Song, 2010*).

averaging operations; thus, new atlases can be built from thousands of annotated images very quickly. With data-driven atlases (of only eight annotated set, one for each test dataset), 74% of cells were correctly identified by the top label prediction in the ground-truth data set, which exceeds the state of the art. Further, 88% and 94% of cells had true identities within the top 3 and the top 5 predicted labels, respectively (*Figure 2A*). Note that with using only positional relationship features in the data-driven atlas, this case is equivalent to predicting identities in experimental whole-brain datasets without color information. More importantly, automated annotation is unbiased because, in principle, the framework can combine manual annotations of cell identities of several users (possibly across labs) in the form of data-driven atlases and can predict identities such that positional relationships in the atlas are maximally preserved. Thus, automated annotation removes individual biases in annotating cells. Further, it greatly supports researchers with no prior experience.

We next compared our method against registration-based methods popular for automatic cell annotation (*Long et al., 2009*; *Long et al., 2008*; *Aerni et al., 2013*; *Toyoshima, 2019*; *Scholz, 2018*) (see Appendix– S1.9 Registration methods do not consider intrinsic similarity features such as relative positional relationships and S2.1 Registration). For fair comparison across methods, all methods used OpenWorm atlas as reference for prediction. The major difference between our framework and previous methods is the use of intrinsic similarity compared to extrinsic similarities in previous methods in the annotation task (*Figure 2B*, *Figure 1—figure supplement 1*). Remarkably, for both experimental and synthetic data, CRF_ID using relative positional features performs the best (*Figure 2C*; *Figure 2—figure supplement 5A*). Note that the decrease in accuracy compared to *Figure 2A* here is due to using static OpenWorm atlas, further highlighting the importance of building data-driven atlases. Notably, CRF_ID outperforms registration-based method across all levels of count noise and position noise in data (*Figure 2D,E*; *Figure 2—figure supplement 5B,C*). The accuracy of registration-based methods falls rapidly with increasing count noise levels, whereas CRF_ID is highly robust, maintaining higher accuracy even when up to 75% of cells in atlas were missing from data. This has important practical implications as the amount of count noise observed in experimental data may vary significantly across reagents, imaging conditions etc. Further, neuron

positions being highly variable across individual animals have been shown (*Yemini et al., 2021*), and confirmed by our datasets as well (*Figure 2—figure supplement 6A*). Because cell positions on average can deviate from their atlas position by more than the distance to their tenth nearest neighbor (*Figure 2—figure supplement 6B*), we expect that this variability introduces large matching errors in registration-based methods. In contrast, most pair-wise relationships are preserved despite the variability of absolute positions (*Figure 2F*; *Figure 2—figure supplement 6C,D*). Interestingly, a hybrid objective function that combines registration using absolute positions with relative position features in CRF_ID framework corrupts the annotation performance (*Figure 2—figure supplement 5A*), likely due to competing effects in the objective function. This again highlights the fact that higher accuracy is achieved by positional relationship features in CRF_ID method.

Next, to compare the computational efficiency of CRF_ID framework with that of registration based methods, we compared the optimization step runtimes of the single-run mode of CRF_ID framework with that of a popular registration method (Coherent Point Drift *Myronenko and Song, 2010*; *Figure 2—figure supplement 7A*). The computational speed of both methods scales with the number of cells to be annotated in images and the number of cells in the atlas. As expected, CRF_ID framework is computationally more expensive compared to CPD, because it optimizes both unary and pairwise potentials. Nonetheless, the optimization runtime of CRF_ID framework for multi-cell calcium imaging use-case (10–50 cells in image) is on the order of 0.1–10 s, on a desktop computer (see Materials and methods – Runtime comparison), when full head ganglion atlas (206 cells) is used for annotation. We emphasize that using full head ganglion atlas for cell identity annotation in whole-brain imaging is important because without prior knowledge of which cells are missing in images, full atlas provides unbiased opportunity to cells in images to take any label from the atlas. In contrast, if only a partial atlas or partially annotated data set is used as atlas, the labels absent in atlas will never get assigned to any cell in images, thus potentially biasing the annotation. In practice, faster runtimes can be achieved in multi-cell calcium imaging and whole-brain imaging case with the use of smaller atlases based on prior knowledge of cells expected in strains. Further, the multiple-run mode of CRF_ID framework can be parallelized using multiple CPU workers. Thus, higher accuracy compared to registration based methods combined with reasonable speeds makes CRF_ID favorable for cell annotation tasks.

## Cell annotation in gene-expression pattern analysis

We next demonstrate the utility of our framework for gene-expression localization analyses, which is important for many problems, for example mapping the cellular atlas of neurotransmitters (*Gendrel et al., 2016*; *Pereira et al., 2015*), receptors (*Vidal et al., 2018*), and neuropeptides (*Bentley et al., 2016*). Conventional methods, for example screening a list of cell-specific marker lines that overlap in expression with the reporter, are laborious and scale poorly with the number of cells expressing the genes of interest and the number of new genes for which expression patterns are to be determined. Our cell annotation framework can considerably reduce manual efforts by generating a small list of candidate identities for each cell expressing the reporter. Subsequently, researchers can easily verify or prune the candidate list. To demonstrate this use case, we imaged a strain with multiple cells labeled with GFP and predicted candidate identities for each cell (*Figure 3A*). Determining cell identities in this case is difficult due to large count noise along with position noise: since the full list of labels in the atlas is much bigger than few cells in the reporter strain (equivalent to scenario four in *Figure 2—figure supplement 1C*). Thus, several degenerate (equally probable) solutions are possible. To avoid accuracy decrease in such cases, we directly predicted the candidate identities of all cells marked with pan-neuronal red fluorescent protein (RFP) using full whole-brain atlas and subsequently assessed the accuracy of only cells of interest, that is those marked with GFP. Our framework accurately generated a candidate list for cells across all datasets (n = 21 animals); 85% of cells had true identities within the top five labels chosen by the framework. In comparison, the candidate list generated by the registration method achieved only 61% accuracy (*Figure 3B*).

## Cell annotation in multi-cell functional imaging experiments

We next demonstrate the utility of our algorithm in another important application - annotating cell identities in multi-cell calcium functional imaging in vivo (*Figure 4*). Automation in this case

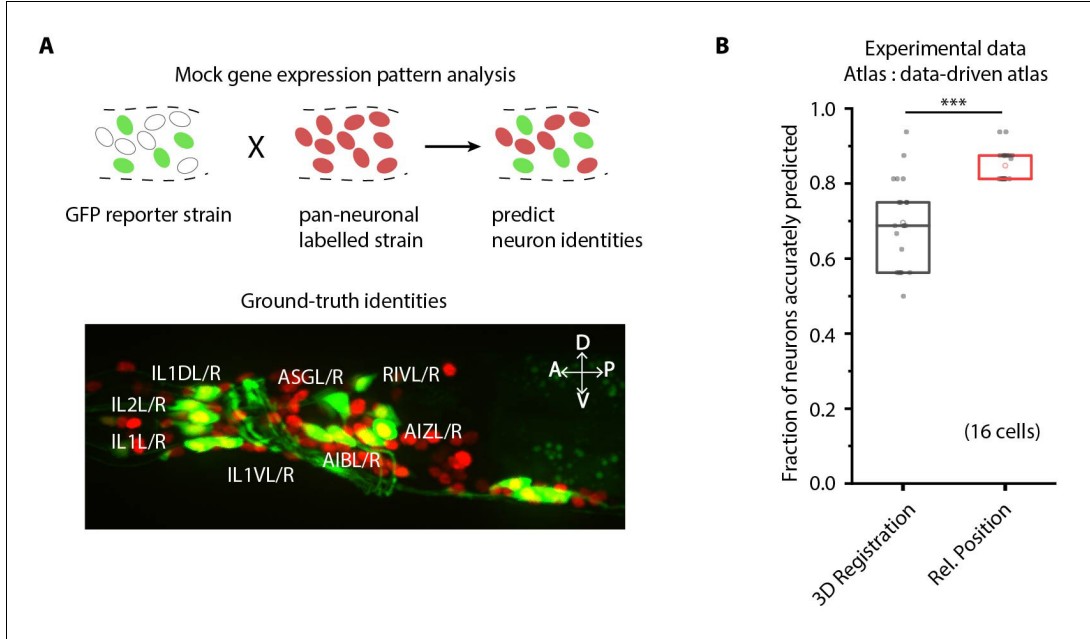

**Figure 3.** CRF_ID framework predicts identities for gene expression pattern analyses. (**A**) (Top) Schematic showing a fluorescent reporter strain with GFP expressed in cells for which names need to be determined. Since no candidate labels are known a priori neurons labels are predicted for all cells marked with pan-neuronally expressed RFP using full whole-brain atlas. (Bottom) A proxy strain AML5 [rab-3p(prom1)::2xNLS::TagRFP; odr-2b:: GFP] with pan-neuronal RFP and 19 cells labeled with GFP was used to assess prediction accuracy. (**B**) CRF_ID framework with relative position features outperforms registration method (n = 21 animals) (***p<0.001, Bonferroni paired comparison test). Accuracy shown for top five labels predicted by both methods. Experimental data comes from strain AML5. Top, middle, and bottom lines in box plot indicate 75th percentile, median, and 25th percentile of data, respectively.

dramatically reduces labor associated with cell annotation for many time points, across trials, animals, and experiments. We used a strain carrying GFP in multiple cells as a proxy for GCaMP-labeled strains for illustration purposes (*Figure 4A*). Given the known candidate list of labels that can be assigned (i.e. no count noise), the configurational space is small, which makes the task easy (similar to scenario three in *Figure 2—figure supplement 1C*). Indeed, our annotation framework identified neurons with high accuracy (98%, n = 35 animals). In comparison, the registration method predicted identities with lower accuracy (88%) even with the small label assignment space (*Figure 4B*). In reality, some neurons may be undetected in the data due to expression mosaicism or low-calcium transients thus adding count noise to data (equivalent to scenario 4 in *Figure 2—figure supplement 1C*). We thus simulated this case by randomly removing up to a third of total neurons from the images and predicting identities of remaining cells using the full label list (*Figure 4C*; *Figure 4—figure supplement 1A*). Even under these conditions, the accuracy of our method remains high (88%), significantly outperforming registration method (81%) (*Figure 4—video 1*). In practice, the performance can be further compensated for by using multiple frames from each video, which we are not doing here in the mock experiment.

To further facilitate annotation accuracy, we explored the utility of landmarks with known identities. Landmarks can also help in establishing a coordinate system in images and guiding post-prediction correction. Because the combinatorial space of potential landmarks is very large (~$10^{14}$ for 10 landmarks out of ~200 cells in the head), we asked what properties landmarks should have. We found that landmarks distributed throughout the head or in lateral ganglion perform better in predicting identities of neurons in all regions of the brain (*Figure 4—figure supplement 2*; Materials and methods). As a test case, we developed strains with spatially distributed, sparse neuronal landmarks using CyOFP (see Material and methods - Construction of landmark strains), which by itself can assist researchers in manual cell identification tasks. When crossed with pan-neuronally expressing GCaMP/RFP reagents, the strains can be used for whole-brain imaging (*Figure 4D*) by

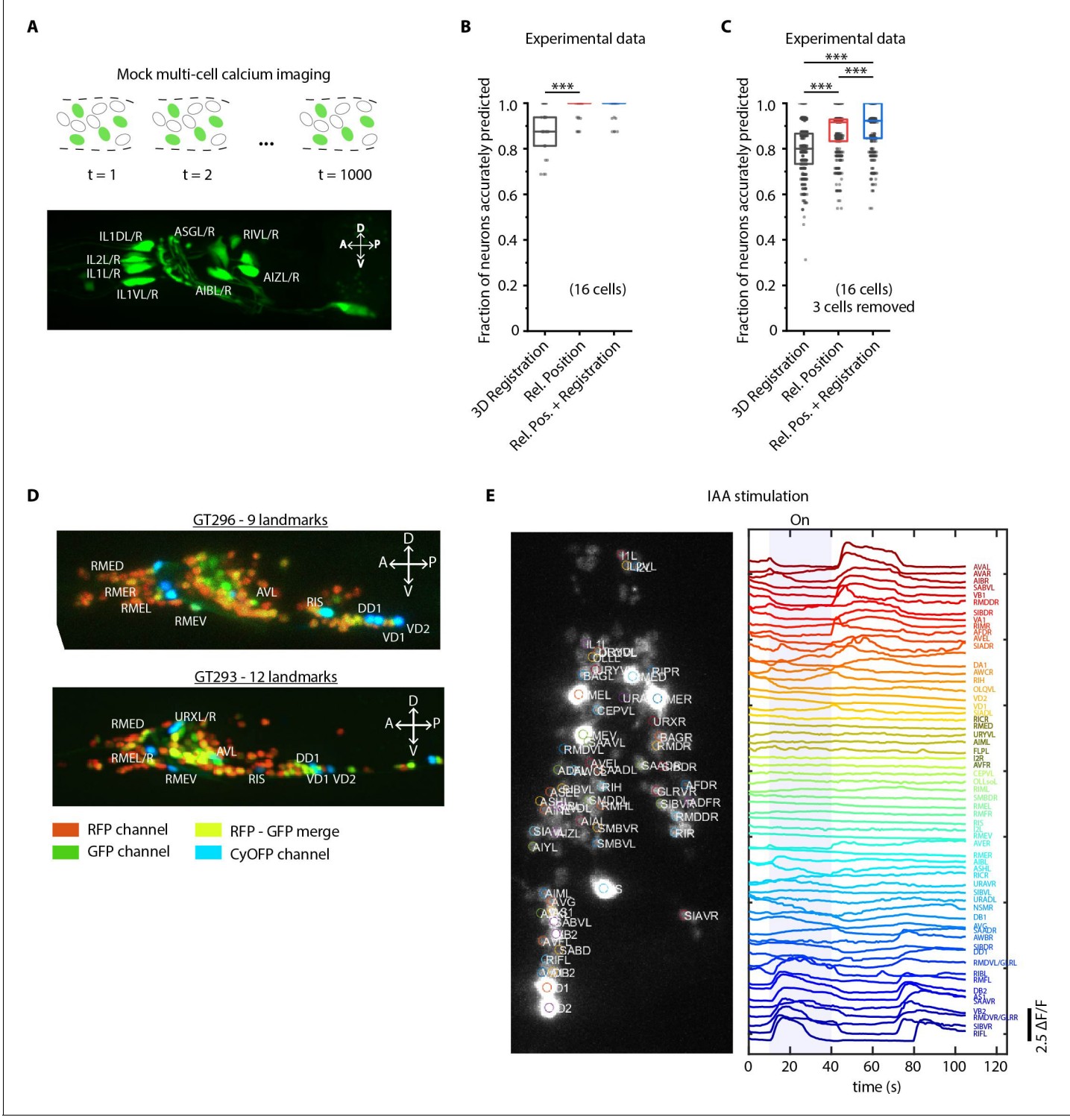

**Figure 4.** Cell identity prediction in mock multi-cell calcium imaging experiments and landmark strain. (**A**) (Top) schematic showing automatic identification of cells in multi-cell calcium imaging videos for high-throughput analysis. (Bottom) A mock strain with GFP-labeled cells was used as an illustration of GCaMP imaging. Only green channel of AML5 strain was used for this purpose. (**B**) CRF_ID framework outperforms registration method (n = 35 animals, ***p<0.001, Bonferroni paired comparison test). OpenWorm atlas was used for prediction. Accuracy results shown for top predicted labels. Experimental data comes from strain AML5 (only green channel used). Top, middle, and bottom lines in box plot indicate 75th percentile, median, and 25th percentile of data, respectively. (**C**) Prediction accuracy comparison for the case of missing cells in images (count noise). ***p<0.001, Bonferroni paired comparison test. Total n = 700 runs were performed across 35 animals for each method with 3 out 16 randomly selected cells

*Figure 4 continued on next page*

*Figure 4 continued*

removed in each run. For fair comparison, cells removed across methods were the same. OpenWorm atlas was used for prediction. Accuracy results shown for top predicted labels. Experimental data comes from strain AML5 (only green channel used). Top, middle, and bottom lines in box plot indicate 75th percentile, median, and 25th percentile of data, respectively. (D) Max-projection of 3D image stacks showing CyOFP labeled landmark cells in head ganglion (pseudo-colored as cyan): animals carrying [unc47p::NLS::CyOFP1::egl-13NLS] (GT296 strain) with nine landmarks (top), and animals carrying [unc-47p::NLS::CyOFP1::egl-13NLS; gcy-32p::NLS::CyOFP1::egl-13NLS] with 12 landmarks (bottom). (E) (Left) max-projection of a 3D image stack from whole-brain activity recording showing head ganglion cells and identities predicted by CRF_ID framework (Top labels). Animal is immobilized in a microfluidic device channel and IAA stimulus is applied to the nose tip. (Right) GCaMP6s activity traces extracted by tracking cells over time in the same 108 s recording and their corresponding identities. Blue shaded region shows IAA stimulation period. Experimental data comes from strain GT296.

The online version of this article includes the following video and figure supplement(s) for figure 4:

**Figure supplement 1.** Relative position features perform better than registration in handling missing cells in images.

**Figure supplement 2.** Spatially distributed landmarks or landmarks in lateral ganglion perform best in supporting CRF_ID framework for predicting identities.

**Figure supplement 3.** Microfluidic device used in chemical stimulation experiments and characterization.

**Figure 4—video 1.** Comparison between the CRF_ID framework and the registration method for predicting identities in case of missing cells.

https://elifesciences.org/articles/60321#fig4video1

using only two channels. This has two advantages: CyOFP can be imaged 'for free' while imaging GCaMP and RFP simultaneously, thus the landmarks providing a concurrent reference in all frames; this strategy also leaves other channels open for optogenetic manipulations and voltage imaging (*Piatkevich et al., 2019*; *Piatkevich et al., 2018*).

We next tested this strategy in a simple whole-brain imaging experiment. Isoamyl alcohol (IAA) is a well-known component of the bacterial metabolites that *C. elegans* senses and responds to *Chalasani et al., 2007*; *L'Etoile and Bargmann, 2000*; *Bargmann et al., 1993*. We recorded neuronal responses to a step-change in IAA concentration using a microfluidic system (*Cho et al., 2020*; *Figure 4—figure supplement 3*). We observed both odor-specific responses and spontaneous activities (*Figure 4E*). More importantly, neurons with algorithm-assigned identities demonstrate expected behavior. For instance, we identified the sensory neuron AWC, and detected an off-response to IAA, consistent with known AWC behavior. In addition, the predicted interneurons (e.g. AVA, RIB, and AIB) also demonstrate previously known activity patterns (*Kato et al., 2015*).

We also tested worms' responses to periodic stimuli of a more complex and naturalistic input – supernatant of bacterial culture (*Figure 5*, *Figure 5—video 1*). A periodic input (5 s On and 5 s Off for eight cycles) entrains many neurons as expected, therefore allowing us to better separate the odor-elicited responses from spontaneous activities (*Figure 5A*). We generated the candidate identities for all recorded neurons (*Figure 5—figure supplement 1A*). Notably, several highly entrained neurons were identified as sensory neurons known to respond to food stimuli (*Liu et al., 2019*; *Wakabayashi et al., 2009*; *Zaslaver et al., 2015*; *Figure 5C*), some of which responded to the onset of the stimuli and some to the withdrawal of the stimuli (*Figure 5D*). The power spectrum of these neurons showed a strong frequency component at 0.1 Hz as expected (*Figure 5B*).

Next, to examine the latent dynamics in the whole-brain activities during the entire experiment, we used traditional Principal Component Analysis (PCA) and Sparse Principal Component Analysis (sPCA) (*Zou et al., 2006*). The overall dynamics are low-dimensional with top three traditional PCs capturing 70% of the variance (*Figure 5—figure supplement 1B*). In comparison, while the top 3 sparse PCs (SPCs) explain 43% of the variance in the data, they enable meaningful interpretation of the latent dynamics by eliminating mixing of activity profiles in PCs (*Figure 5E*). SPC1 shows a systematic decline of the signals, presumably related to photobleaching of the fluorophores; both SPC2 and SPC3 illustrate spontaneous activities with different temporal dynamics. With automatic annotation, we were able to identify cell classes belonging to each SPC (*Figure 5—figure supplement 1C*). We then analyzed the relationship between motion and neuron activities. In our microfluidic device, the animals are not fully immobilized. By tracking landmarks on the body; we observed propagating waves along the body (*Figure 5F*; *Figure 5—figure supplement 1D*, *Figure 5—video 2*). Interestingly, cells participating in SPC2 showed significantly higher mutual information with motion than any other component (*Figure 5G*). Examining the connection between activities of neurons that drive SPC2 and animal motion demonstrates that these neurons are indeed correlated or

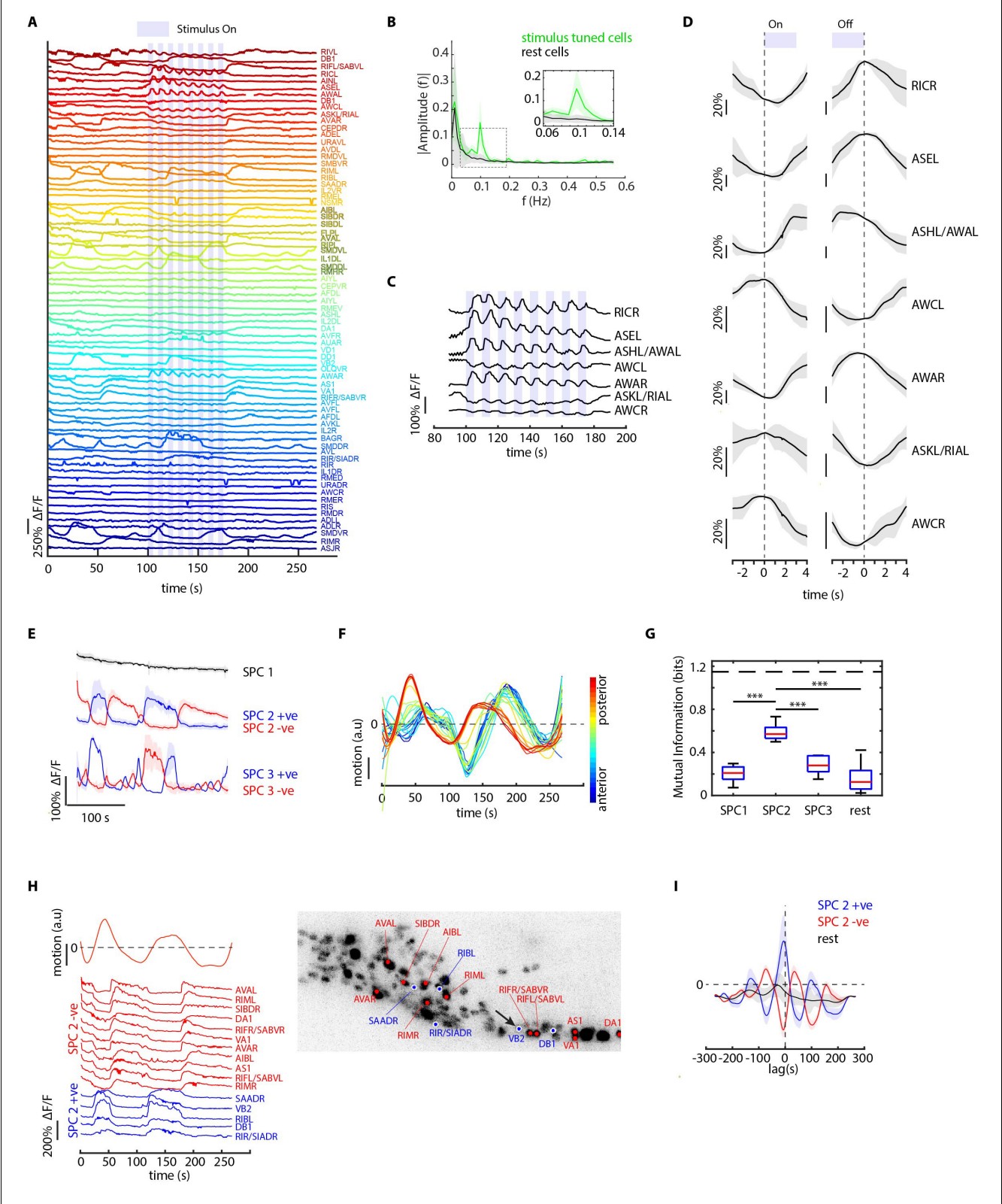

**Figure 5.** CRF_ID framework identifies neurons representing sensory and motor activities in whole-brain recording. (**A**) GCaMP6s activity traces of 73 cells automatically tracked throughout a 278 s long whole-brain recording and the corresponding predicted identities (top labels). Periodic stimulus (5 sec-on – 5 sec-off) of bacteria (*E. coli* OP50) supernatant was applied starting at 100 s (shaded blue regions). Experimental data comes from strain GT296. (**B**) Power spectrum of neuron activity traces during the stimulation period for all cells. Cells entrained by 0.1 Hz periodic stimulus show

*Figure 5 continued on next page*

*Figure 5 continued*

significant amplitude for 0.1 Hz frequency component (green). (**C**) Activity traces of cells entrained by periodic stimulus shown for the stimulation period. Blue shaded regions indicate stimulus ON, unshaded region indicate stimulus OFF. Identities predicted by the framework are labeled. (**D**) Average ON and OFF responses of cells entrained by periodic stimulus across trials. The black line indicates mean and gray shading indicates ± s.e. m. (**E**) Average activities of neurons with significant non-zeros weights in the first three sparse principal components (SPCs). Activities within each component are stereotypical and different components show distinct temporal dynamics. Cells with positive weights (blue) and negative weights (red) in SPC2 and SPC3 showed anti-correlated activity. Out of the 67 non-stimulus-tuned cells, 19 had non-zero weights in SPC1, 16 cells had non-zero weights in SPC2, and 5 cells had non-zero weights in SPC3. SPC1, SPC2, and SPC3 weights of cells are shown in *Figure 5—figure supplement 1*. Shading indicates mean ± s.e.m of activity. (**F**) Velocity (motion/second) traces of cells along anterior-posterior (AP) axis (blue to red) show phase shift in velocity indicating motion in device shows signatures of wave propagation. (**G**) Cells with non-zero weights in SPC2 show high mutual information with worm velocity compared to cells grouped in other SPCs (*** denotes p<0.001, Bonferroni paired comparison test). Median (red line), 25th and 75th percentiles (box) and range (whiskers). Dashed line indicates entropy of velocity (maximum limit of mutual information between velocity and any random variable). Velocity of cell indicated by the black arrow in panel H right was used for mutual information analysis. (**H**) Activity traces of 16 cells (with significant non-zero weights) in SPC2 and corresponding identities predicted by the framework. Red traces for cells with negative weights in SPC2, blue traces for cells with positive weights in SPC2. Worm motion/second shown on top. (Right) max projection of 3D image stack showing head ganglion neurons and cells with positive weights (blue) and negative weights (red) in SPC2. Motion/second of cell indicated with arrow is shown in left panel. (**I**) Cross-correlation analysis between velocity and cells with non-zero weights in SPC2 shows a strong correlation between neuron activities and velocity. In comparison, other cells show low correlation. Velocity of cell indicated by arrow in panel H right was used for cross-correlation analysis.

The online version of this article includes the following video and figure supplement(s) for figure 5:

**Figure supplement 1.** Further analysis of data in periodic food stimulation and whole-brain imaging experiment.

**Figure 5—video 1.** Whole-brain functional imaging with bacteria supernatant stimulation.

https://elifesciences.org/articles/60321#fig5video1

**Figure 5—video 2.** Wave propagation in animal and correlation of neuron activities to worm motion.

https://elifesciences.org/articles/60321#fig5video2

---

anti-correlated with the motion we detected (*Figure 5H*); notably, these neurons included several command interneurons such as AVA, RIM, and motor neurons such as VA and DA (*Kato et al., 2015*; *Figure 5H*). Cross-correlation analysis between motion and neuron activities showed that neurons are activated ahead of motion (*Figure 5I*); when a lag is added to the neuron activities, the mutual information of SPC2 neurons with motion is maximum at the same delay observed in the cross-correlation analysis (*Figure 5—figure supplement 1E*). These experiments together demonstrate the power of the approach, which enabled previously difficult simultaneous analyses of several sensory, inter-, and motor neurons' activities to natural food stimulus. Thus, automatic identity prediction enabled meaningful interpretation of the whole-brain data.

## CRF framework is broadly applicable to wider conditions

Another important advantage of the CRF_ID framework is its flexibility to incorporate additional information to improve the identification accuracy, by simply adding new terms in the objective function without disturbing the weights of existing features. Here we demonstrate this idea by using the recently developed NeuroPAL (*Yemini et al., 2021*) that provides a unique chromatic code to each neuron (*Figure 6A*). The chromatic code was included as a unary feature in the model (see Appendix 1–Extended methods S2.6). Using manually curated ground-truth data, we compared different methods. These methods included different orthogonal feature combinations, as used by previous approaches, thus providing insights into which features perform best in predicting cell identities (*Figure 6B*, see Appendix 1–Extended methods S2). For fair comparison across methods, static OpenWorm atlas was used across all methods. For methods that use color information, we built data-driven color atlases (Appendix 1–Extended methods S2.4) using all datasets except the test dataset: leave-one-out color atlases. Unsurprisingly, registration performs poorly (with or without color information); color alone is not sufficient, and color combined with spatial features improves the accuracy (whether registration or relative position is used). Notably, the best performing model uses relative position features in combination with color and without registration term (*Figure 6B*; *Figure 6—figure supplement 1A*), achieving 67.5% accuracy for the top-label prediction. Further, for 85.3% of the neurons, the true identity is within the top three labels.

Next, to assess true potential of CRF_ID framework, instead of using OpenWorm atlas, we used data-driven positional relationship atlases, so that the predictions are now performed with data-driven atlases for both positional relationships and color. To test the generalizability of the method

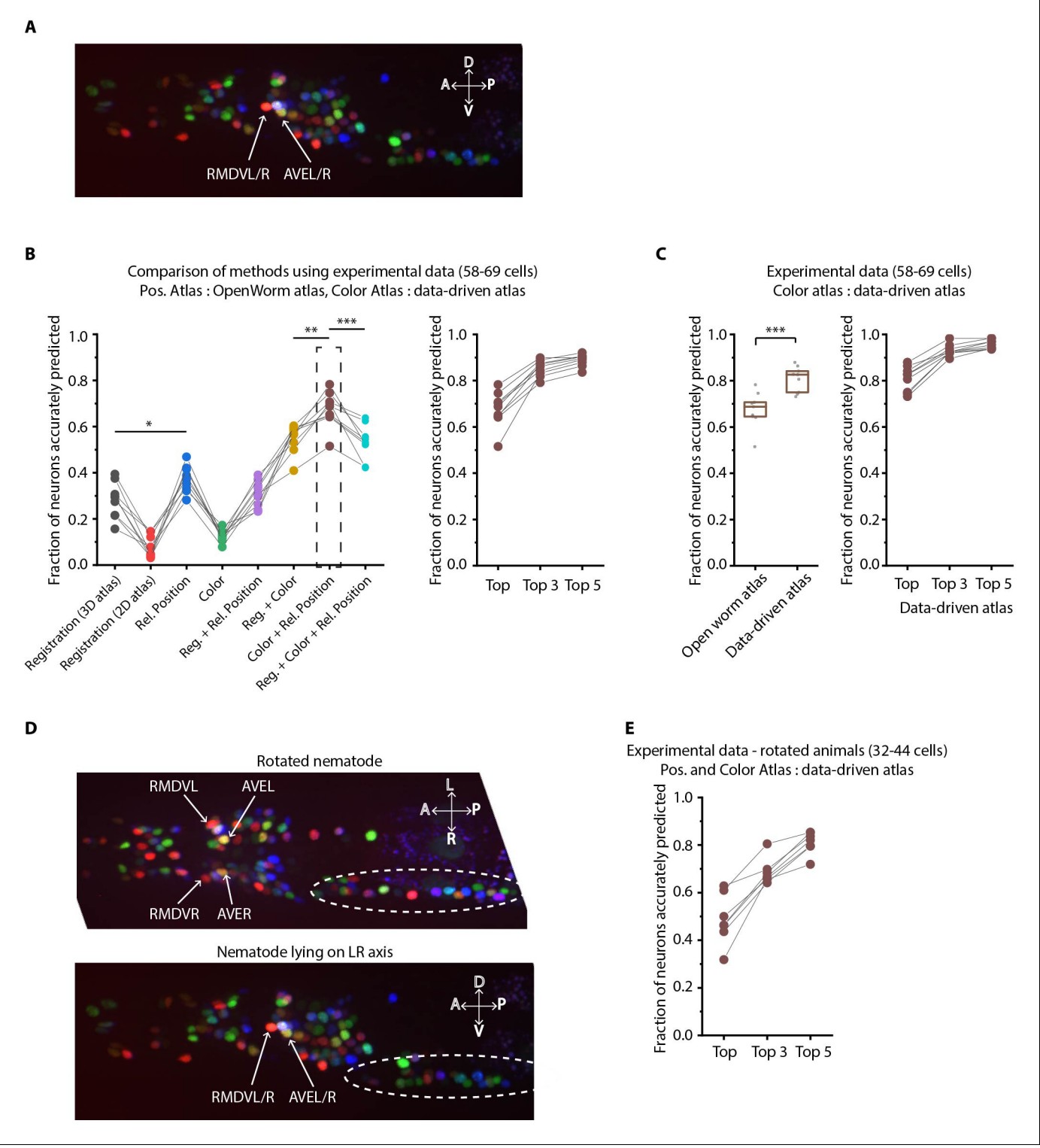

**Figure 6.** Annotation framework is generalizable and compatible with different strains and imaging scenarios. (**A**) A representative image (max-projection of 3D stack) of head ganglion neurons in NeuroPAL strain OH15495. (**B**) (Left) comparison of prediction accuracy for various methods that use different information. CRF_ID framework that combines relative position features along with color information performs best (n = 9 animals, *p<0.05, **p<0.01, ***p<0.001, Bonferroni paired comparison test). (Right) the best performing method predicts cell identities with high accuracy. OpenWorm static atlas was used for all methods. Color atlas was built using experimental data with test data held out. Ensemble of color atlases that combine two different color matching methods were used for prediction. Accuracy results shown for top predicted labels. Experimental data comes from strain OH15495. (**C**) (Left) annotation framework can easily incorporate information from annotated data in the form of data-driven atlas, which

*Figure 6 continued on next page*

*Figure 6 continued*

improves prediction accuracy (***p<0.001, Bonferroni paired comparison test). Prediction was performed using leave-one-out data-driven atlases for both positional relationship features and color. Accuracy shown for top predicted labels. Ensemble of color atlases that combine two different color matching methods were used for prediction. (Right) accuracy achieved by top, top 3, and top 5 labels. Experimental data comes from strain OH15495. Top, middle, and bottom lines in box plot indicate 75th percentile, median and 25th percentile of data, respectively. (D) An example image of head ganglion neurons in NeuroPAL strain for rotated animal (nematode lying on DV axis). In contrast, animal lying on the LR axis is shown below. The locations of RMDVL/R, AVEL/R cells in the two images are highlighted for contrasts. Dashed ellipses indicate positions of cells in retrovesicular ganglion, showing that the rotated animal is not rigidly rotated. Experimental data comes from strain OH15495. (E) Top-label prediction accuracies for non-rigidly rotated animal. n = 7 animals. Experimental data comes from strain OH15495 and OH15500. Prediction was performed using leave-one-out data-driven atlases for both positional relationship features and color. Accuracy shown for top predicted labels. Ensemble of color atlases that combine two different color matching methods were used for prediction.

The online version of this article includes the following figure supplement(s) for figure 6:

**Figure supplement 1.** Additional results on prediction performance of CRF_ID method on NeuroPAL data: comparison against registration method and utility of ensemble of color atlases.

**Figure supplement 2.** Example annotations predicted by the CRF_ID framework for animals imaged lying on the LR axis.

**Figure supplement 3.** Example annotations predicted by the CRF_ID framework for animals twisted about the anterior-posterior axis (note the anterior and lateral ganglions show clear left-right separation whereas retrovesicular ganglion instead of being in the middle is more toward one of the left or right sides).

on unseen datasets, we compared the accuracy of CRF_ID framework across several kinds of data-driven atlases (*Figure 6—figure supplement 1B*). These included the following: (1) positional relationship and color atlases, which include information from all datasets including test dataset, (2) color information comes from all datasets and leave-one-out atlases for positional relationships built with test dataset held out, (3) positional relationship information, which comes from all datasets and leave-one-out color atlases, and (4) leave-one-out atlases for both positional relationships and color. The analysis revealed that accuracy falls more sharply with using color leave-one-out atlases compared to the leave-one-out positional relationship atlases. This implies that in the datasets used, positional relationship features are more consistent compared to color features. Thus, leave-one-out positional relationships atlases can represent positional relationships among cells in test datasets. Further, to assess the contribution of color information to prediction accuracy, we compared the accuracy of the case using both positional relationship and color leave-one-out atlases (*Figure 6—figure supplement 1B* last column) to the case where predictions were performed using only leave-one-out positional relationship atlases shown earlier in *Figure 2A*. We found that color contributed little to improving accuracy. This is because in the datasets used, the color variability in raw RGB values across animals is far greater than the position variability across animals; hence, the distribution of color features in the training data does not match the distribution of features in the test data. This could be due to inherent variations in fluorophore expressions across animals, or variations in imaging settings (i.e. exposure time of each channel, laser power etc.) across sessions. Thus, a naive approach of building color atlas by directly aggregating RGB values from training images contributed little to improving accuracy. The problem of mismatched feature distributions in test data compared to training data is commonly solved by domain adaptation methods in machine-learning community. We adopted a simple domain adaptation strategy for dealing with color discrepancies and developed a two-step method (Appendix 1–Extended methods S2.4). First, we aligned the distributions of RGB values in training datasets to the test dataset by several methods such as simple normalization of color channels, histogram matching color channels in training images to test data set, contrast and gamma adjustment of image channels, and transforming the color space of all images with color invariants (*Finlayson et al., 1998*; *Figure 6—figure supplement 1C*). Note that this alignment does not rely on cell identity information at all. These color alignment methods by themselves or in combination with other methods improved accuracy for some datasets but not all datasets. Second, we used an ensemble of leave-one-out color atlases for prediction, that is predictions were performed using multiple leave-one-out color atlases each built with a different color alignment technique. The ensemble, in comparison to single atlases, provides a combination of color features from aligned color distributions, thus improving accuracy. The two-step method improves accuracy by 6% over the naïve approach (*Figure 6—figure supplement 1D*). Overall, a significant improvement in the model accuracy was achieved by using data-driven atlas to account for biological variability in both the positional relationships and color (*Figure 6C*; *Figure 6—figure*

*supplement 2*). Using the data-driven atlas, accuracy improved to 81% (top labels); more than 93% of the neurons have their true identities in the top three labels chosen by the model. We expect that more datasets for the atlas will continue to improve the accuracy.

Lastly, we show that our framework is equipped to work with realistic complex scenarios of animals imaged in different orientations, often not rigid rotations (*Figure 6D*). Identifying cells in these cases is challenging: manual annotation using the 2D-atlas (*Altun and Hall, 2009*) is not possible since it lacks left-right information; further, due to low-z sampling of image stacks, segmented positions of cells along z-axis are noisier. These challenges can be addressed by using the data-driven atlas. We manually annotated data collected for animals imaged with varying degrees of non-rigid rotations and built data-driven atlases for positional relationships and color. Further, we combined rotated animals' atlas with the previous atlas built from animals imaged in lateral orientation to build a super atlas. With the test data held out in atlases, the prediction accuracy of top labels was 48.8%, and the accuracy was 68.7% for top three labels (*Figure 6E* ), which are reasonable for practical purposes. In this case too, aligning the color distributions in atlas to the test data set and using ensemble of color atlases with different alignment techniques helped to significantly improve accuracy over the naïve approach to build color atlases (*Figure 6—figure supplement 1E*).

## Discussion

Annotating anatomical features and cellular identities in biological images are important tasks for many applications. Here, we demonstrated our CRF_ID framework is suitable for fluorescently labeled cells in 3D images for many applications. Using both ground-truth experimental data of whole-brain image stacks and synthetic data generated from an atlas, we showed that our framework is more accurate compared to existing approaches. We demonstrated using real examples how the pipeline can be used for analysis of gene expression pattern for instance, and for neuron identification from dense multi-cell or whole-brain imaging experiments. Further, our CRF_ID framework significantly speeds up the cell identification compared to manual labeling while reducing bias.

With the pipeline, we address several challenges. There is ample evidence that anatomy varies from individual to individual, and from condition to condition. This variability, or position noise, is a major source of roadblock in effectively applying previous methods to annotate the whole-brain recording data. Because our framework leverages intrinsic similarity (*Bronstein et al., 2007*), it performs better than registration methods in handling position noise (*Figure 2E*; *Figure 2—figure supplement 5C*). Further, CRF_ID formulation is more accurate in handling count noise that is cases of missing or undetectable cells in images (*Figure 2D*; *Figure 4C*; *Figure 2—figure supplement 5B*; *Figure 4—figure supplement 1A*), because the missing neurons do not upset the relationships among the detectable neurons in the CRF_ID formulation while missing neurons introduce large uncertainty in registration methods. Lastly, CRF_ID method predicts identities with sufficient accuracy for different postural orientations of the worms often seen in our microfluidic experiments. We expect that this superiority is maintained for any data that have relational information preserved, this is the case virtually in all biological samples where tissues are connected by matrix materials, such as in other whole-brain recordings or for registration of fixed tissues.

Building and using data-driven atlases in the pipeline is simple and yet highly effective. We expect that data from more animals, different orientations, age, and imaging techniques will further improve the generalizability. Since building such data-driven atlas for our framework requires only cheap mathematical operations (Appendix 1–Extended methods S1.7), incorporating more data is quite simple and easily scalable. In contrast, previous registration-based methods may require simultaneous or batch-wise registration of multiple images to one reference image; this would require solving multiple constrained regression problems on increasingly large data sets, thus rendering them computationally unscalable. Further, without systematic methodology of which image should be chosen as reference image, atlas gets biased toward the chosen reference image or by the order in which blocks of images are registered to the reference image. Tackling this challenge is an active field of research (*Wang et al., 2008*; *Evangelidis et al., 2014*). In comparison, in CRF method, atlas building is unbiased toward any image set because there is no concept of reference image. Additionally, atlas can be built from all images simultaneously because of the cheapness of mathematical operations.

Another major advantage of data-driven atlases in our framework is that the atlases can be built incrementally using partially or fully annotated datasets, for example using lines that label partial and distinct subsets of cells. In comparison, registration-based methods cannot build atlas from lines that label distinct subset of cells. This is because registration-based methods build probabilistic spatial atlases by first establishing correspondence among cells in images and subsequently registering images. However, this is not possible if the cells in different images do not have any overlapping cells or have very few overlapping cells. In comparison, atlases built in CRF_ID framework store probabilistic positional relationship features among cells observed within each image. Hence, correspondence between images is not required. Thus, in principle, CRF_ID framework can combine manually annotated data across different lines, generated by different researchers (and across labs) in the form of data-driven atlases. Automated annotation using such atlases removes individual biases in annotating cells. Further, it greatly supports researchers with no prior experiences with cell identification. We expect that using our framework, large-scale atlases can be built in the future with community contributed data.

Finally, a distinction of CRF_ID framework is its ability to build and annotate with complete atlases covering all cells. This is made possible by the efficient utilization of data, even from strains with non-overlapping cells. Annotating against a complete atlas is crucial because commonly in practice, no prior information is available on exactly which cells are missing from the images before annotation. Registration-based or unary potential-based methods are limited in building atlas by the availability of overlapping strains. Thus, in these methods, cells that are missing in the atlas can never be assigned to cells in images; hence these methods do not perform completely unbiased annotation. In comparison, CRF_ID framework uses a complete atlas to assign any possible label in the atlas to cells in the images, thus performing unbiased annotation, resulting in better handling of count noise in images.

CRF framework offers the flexibility of combining arbitrary unary features with arbitrary pairwise features for cell annotation task. We demonstrate the utility of such flexibility by combining color information of cells in NeuroPAL strain with positional relationship features and show higher accuracy compared to other methods. Our experiments show that in order to be able to utilize color information of cells in NeuroPAL for automatic annotation of cell identities, color consistency across animals needs to be maintained, either experimentally or by *post hoc* corrections. Experimentally, consistent protocol/imaging settings across sessions should be maintained as much as possible. Even with consistent protocol, color variation may exist due to inherent differences across animals in relative expressions of fluorophores that define the chromatic code of cells. This can be tackled by (1) collecting large volume of data to capture each cells' full RGB variations and (2) using computational domain adaptation techniques. More advancement in image color transfer and domain adaptation techniques will further improve accuracy in future.

While we only considered pairwise features in the current formulation, feature functions with arbitrary dependency can be included in the model that may further improve prediction accuracy (*Kohli et al., 2009*; *Najafi et al., 2014*). Advances in structured energy minimization field (*Kohli et al., 2009*; *Komodakis and Paragios, 2009*; *Krähenbühl and Koltun, 2011*) will enable tackling the increased complexity of combinatorial optimization in these cases. Our workflow borrows techniques developed in metric object/shape matching literature for annotation in biological images. Log-linear parameterization in our framework makes the model a member of the exponential families (*Wainwright and Jordan, 2007*); thus, the objective function in our framework has interesting connections with max-entropy models and with the entropy-regularized optimal transport objective functions (*Solomon et al., 2016*; *Nitzan et al., 2019*). Therefore, improvements in computational speed can be achieved by borrowing fast optimization techniques for quadratic assignment problems developed in optimal transport literature. Advances in these fields will continue to improve the method development in image analysis.

We anticipate that by using our generalizable formulation, similar pipelines can be set up to annotate more image sets in other organisms and build atlases. Data in many anatomical annotation problems (e.g. brain atlas construction, registering images from different modalities, comparing animals or related species to one another for developmental studies) share a similar property, in that the anatomical features of interest maintain a cohesion from sample to sample. This underlining cohesion lends itself to the CRF framework. As we have shown, the pipeline is extremely flexible in incorporating new information. Thus, framework should be easily modifiable catering to the data

demands in other organisms including features besides landmarks and spectral information such as cellular morphology and expected cellular activities (e.g. calcium transients). Because the only inputs to our framework are segmented anatomical regions in images and positional relationships among them, information already available in data across organisms (*Robie et al., 2017*; *Kim et al., 2015*; *Chen et al., 2019*; *Ronneberger et al., 2012*), the framework proposed here should be generally useful for many problems in model organisms such as *Drosophila* (*Robie et al., 2017*; *Vaadia et al., 2019*), zebrafish (*Ronneberger et al., 2012*), mammalian brains (*Kim et al., 2015*; *Chen et al., 2019*). Besides fluorescence, the pipeline should also be able to work with data from other modalities including EM, live imaging, and fluorescence imaging from cleared tissues.

# Materials and methods

## Reagents
For all experiments, animals were cultured using standard techniques (*Stiernagle, 2006*). A detailed list of strains used is provided below.

| Name | Genotype | Experiments | Reference |
|------|----------|-------------|-----------|
| GT290 | aEx18[unc-47p::NLS::CyOFP1::egl-13NLS] | Strain with nine neuronal landmarks in head | This work |
| GT298 | aEx22[unc-47p::NLS::CyOFP1::egl-13NLS + gcy-32p::NLS::CyOFP1::egl-13NLS] | Strain with 12 neuronal landmarks in head | This work |
| AML32 | wtfIs5 [rab-3p::NLS::GCaMP6s + rab-3p::NLS::tagRFP] | Strain used to make whole-brain imaging strain with CyOFP labeled landmarks GT296 | *Nguyen et al., 2016* |
| AML70 | wtfIs5 [rab-3p::NLS::GCaMP6s + rab-3p::NLS::tagRFP]; lite-1(ce314) X | Strain used to make whole-brain imaging strain with CyOFP labeled landmarks GT293 | *Scholz, 2018* |
| KG1180 | lite-1(ce314) X | Strain used to make whole-brain imaging strain with CyOFP labeled landmarks GT296 | CGC |
| GT296 | wtfIs5 [rab-3p::NLS::GCaMP6s + rab-3p::NLS::tagRFP]; aEx18[unc-47p::NLS::CyOFP1::egl-13NLS]; lite-1(ce314) X | Strain used for whole-brain functional imaging experiments (*Figures 4* and *5*) and quantifying cell position variability. | This work |
| GT293 | wtfIs5 [rab-3p::NLS::GCaMP6s + rab-3p::NLS::tagRFP]; lite-1(ce314) X; aEx22[unc-47p::NLS::CyOFP1::egl-13NLS + gcy-32p::NLS::CyOFP1::egl-13NLS] | Strain used for quantifying cell position variability. | This work |
| AML 5 | otIs355 [rab-3p(prom1)::2xNLS::TagRFP] IV. kyIs51 [odr-2p::GFP + lin-15(+)] | Strain used for mock gene-expression pattern analysis and mock multi-cell calcium imaging experiments | *Nguyen et al., 2016* |

*Continued on next page*

*Continued*

| Name | Genotype | Experiments | Reference |
|------|----------|-------------|-----------|
| OH15495 | otIs696 [UPN::NLS::TagRFP-T + acr-5::NLS::mTagBFP2::H2B + flp-1::NLS::mTagBFP2::H2B + flp-6::NLS::mTagBFP2::H2B + flp-18::NLS::mTagBFP2::H2B + flp-19::NLS::mTagBFP2::H2B + flp-26::NLS::mTagBFP2::H2B + gcy-18::NLS::mTagBFP2::H2B + ggr-3::NLS::mTagBFP2::H2B + lim-4::NLS::mTagBFP2::H2B + pdfr-1::NLS::mTagBFP2::H2B + srab-20::NLS::mTagBFP2::H2B + unc-25::NLS::mTagBFP2::H2B + cho-1::NLS::CyOFP1::H2B + flp-13::NLS::CyOFP1::H2B + flp-20::NLS::CyOFP1::H2B + gcy-36::NLS::CyOFP1::H2B + gpa-1::NLS::CyOFP1::H2B + nlp-12::NLS::CyOFP1::H2B + nmr-1::NLS::CyOFP1::H2B + ocr-1::NLS::CyOFP1::H2B + osm-9::NLS::CyOFP1::H2B + srh-79::NLS::CyOFP1::H2B + sri-1::NLS::CyOFP1::H2B + srsx-3::NLS::CyOFP1::H2B + unc-8::NLS::CyOFP1::H2B + acr-2::NLS::mNeptune2.5 + ceh-2::NLS::mNeptune2.5 + dat-1::NLS::mNeptune2.5 + dhc-3::NLS::mNeptune2.5 + eat-4::NLS::mNeptune2.5 + flp-3::NLS::mNeptune2.5 + gcy-35::NLS::mNeptune2.5 + glr-1::NLS::mNeptune2.5 + gcy-21::NLS::CyOFP1::H2B::T2A::NLS::mTagBFP2::H2B + klp-6::NLS::mNeptune2.5::T2A::NLS::CyOFP1::H2B + lim-6::NLS::mNeptune2.5::T2A::NLS::CyOFP1::H2B + mbr-1::NLS::mNeptune2.5::T2A::NLS::mTagBFP2::H2B + mec-3::NLS::CyOFP1::H2B::T2A::NLS::mTagBFP2::H2B + odr-1::NLS::mNeptune2.5::T2A::NLS::mTagBFP2::H2B + srab-20::NLS::mNeptune2.5::T2A::NLS::mTagBFP2::H2B] | NeuroPAL strain demonstrating the ease of incorporating color information, and thus demonstrating generalizability | *Yemini et al., 2021* |

Continued

| Name | Genotype | Experiments | Reference |
|---|---|---|---|
| OH15500 | otIs672 [rab-3::NLS::GCaMP6s + arrd-4:NLS:::GCaMP6s]. otIs669 [UPN::NLS::TagRFP-T + acr-5::NLS::mTagBFP2 ::H2B + flp-1::NLS::mTagBFP2::H2B + flp-6::NLS::mTagBFP2::H2B + flp-18::NLS::mTagBFP2::H2B + flp-19::NLS::mTagBFP2::H2B + flp-26::NLS::mTagBFP2::H2B + gcy-18::NLS::mTagBFP2::H2B + ggr-3::NLS::mTagBFP2::H2B + lim-4::NLS::mTagBFP2::H2B + pdfr-1::NLS::mTagBFP2::H2B + srab-20::NLS::mTagBFP2::H2B + unc-25::NLS::mTagBFP2::H2B + cho-1::NLS::CyOFP1::H2B + flp-13::NLS::CyOFP1::H2B + flp-20::NLS::CyOFP1::H2B + gcy-36::NLS::CyOFP1::H2B + gpa-1::NLS::CyOFP1::H2B + nlp-12::NLS::CyOFP1::H2B + nmr-1::NLS::CyOFP1::H2B + ocr-1::NLS::CyOFP1::H2B + osm-9::NLS::CyOFP1::H2B + srh-79::NLS::CyOFP1::H2B + sri-1::NLS::CyOFP1::H2B + srsx-3::NLS::CyOFP1::H2B + unc-8::NLS::CyOFP1::H2B + acr-2::NLS::mNeptune2.5 + ceh-2::NLS::mNeptune2.5 + dat-1::NLS::mNeptune2.5 + dhc-3::NLS::mNeptune2.5 + eat-4::NLS::mNeptune2.5 + flp-3::NLS::mNeptune2.5 + gcy-35::NLS::mNeptune2.5 + glr-1::NLS::mNeptune2.5 + gcy-21::NLS::CyOFP1::H2B::T2A::NLS:: mTagBFP2::H2B + klp-6::NLS::mNeptune2.5::T2A::NLS::CyOFP1::H2B + lim-6::NLS::mNeptune2.5::T2A::NLS:: CyOFP1::H2B + mbr-1::NLS::mNeptune2.5:: T2A::NLS::mTagBFP2::H2B + mec-3::NLS::CyOFP1::H2B::T2A::NLS::m TagBFP2::H2B + odr-1::NLS::mNeptune2.5:: T2A::NLS::mTagBFP2::H2B + srab-20::NLS::mNeptune2.5::T2A::NLS::m TagBFP2::H2B] V | NeuroPAL strain demonstrating the ease of incorporating color information, and thus demonstrating generalizability | *Yemini et al., 2021* |

## Imaging

All imagings were performed using either a Perkin Elmer spinning disk confocal microscope (1.3 NA, 40x, oil objective) or Brucker Opterra II Swept field confocal microscope (0.75 NA, 40x, Plan Fluor air objective), with an EMCCD camera.

To acquire data used for framework validation and comparison against other methods (*Figure 2*), gene expression pattern analysis (*Figure 3*), multi-cell calcium imaging (*Figure 4*), imaging landmark strain (*Figure 4*) and NeuroPAL imaging (*Figure 6*), animals were synchronized to L4 stage and were imaged in an array microfluidic device (*Lee et al., 2014*). A single 3D stack was acquired with either 0.5 μm or 1 μm spacing between z-planes and 10 ms exposure time (except for NeuroPAL strain where exposure times of different channels were chosen based on the guidelines provided in Neuro-PAL manuals *Yemini et al., 2021*).

Whole-brain functional recording data while providing chemical stimulus were acquired using a microfluidic device designed for applying chemical stimulation (*Cho et al., 2020*) to the nose-tip of the animal. Here, image stacks were acquired with 1 μm spacing between z-planes and 10 ms

exposure for each z-plane. This enabled recording videos at 1.1 volumes/s while imaging two channels simultaneously (GCaMP and RFP). Animals were synchronized to Day-1 adult stage.

## Generating synthetic data for framework tuning and comparison against other methods

Synthetic data was generated using the freely available 3D atlas at OpenWorm (*Szigeti et al., 2014*). Atlas available at Worm Atlas (*Altun and Hall, 2009*) was not used as it provides only a 2D view. To mimic the conditions encountered in experimental data, two noise perturbations were applied to the 3D atlas (*Figure 2—figure supplement 2*). First, due to inherent biological variability, positions of cells observed in images do not exactly match the positions in atlas. Thus, position noise was applied to each cell in the atlas. This noise was sampled from a normal distribution with zero mean and fixed variance $\Sigma = diag\left(\left[\sigma_x, \sigma_y, \sigma_z\right]\right)$. Here $\sigma_x$, $\sigma_y$ and $\sigma_z$ denote variances along $x$, $y$ and $z$ image dimensions and $diag(\mathrm{x})$ denotes diagonalizing vector $\mathrm{x}$. Hence, the position of $i^{th}$ cell $p_i \in R^3$ in synthetic data was defined as $p_i = p_{i,atlas} + \epsilon$, $\epsilon \sim \mathcal{N}(0, \Sigma)$. Here $p_{i,atlas}$ is the position of the $i^{th}$ cell in the atlas. To determine the variance $\Sigma$, we quantified the variance of cell positions observed in experimental data (*Figure 2—figure supplement 2A, C, E*) using the strains GT293, GT295 with neuronal landmarks. We calculated the 25[th] percentile and 75[th] percentile of the variance across all cells across all animals (n = 31) to define the lower bound and upper bound position noise levels observed in experimental data. However, this variability cannot be directly applied to the atlas due to different spatial scales in images and atlas. Thus, we first normalized the observed variance of cell positions in images with inter-cell distances in images and then scaled it according to the inter-cell distances in atlas (*Figure 2—figure supplement 2B,D,F,G,H*) to define lower bound and upper bound noise to be applied to the atlas. More position noise levels such as those in *Figure 2E* and *Figure 2—figure supplement 5* were set by varying $\Sigma$ between zero and upper-bound noise level.

Second, although there are 195–200 neurons in head ganglion in *C. elegans,* only 100–130 cells were detected in most image stacks. Remaining cells are not detected either due to low-expression levels of fluorophores, variability in expression levels of genetic marker used to label cells (mosaicism, incomplete coverage etc.) or inability of segmentation methods to resolve densely packed cells. This increases the complexity of determining the labels of cells. To illustrate this, matching 195 cells in an image to 195 cells in the atlas is easier as only one or very few possible configurations of label assignments exist that maximally preserves the positional relationships among cells. In contrast, in the case of matching 100 cells in an image to 195 cells in atlas, many possible labeling arrangements may exist that will equally preserve the positional relationships among cells. Thus, to simulate count noise in synthetic data, randomly selected cells in atlas were marked as missing and synthetic data was generated from the atlas with remaining cells. Hence, identities were predicted for remaining cells only in synthetic data using the full atlas. Number of cells set as missing was set by the count noise level parameter, defined as the fraction of total cells in the atlas that were set as missing. Since no prior information was available on which regions of the head ganglion had more cells missing, we selected the missing cells uniformly across brain regions.

Finally, bounds on prediction accuracy (shown as gray regions in *Figure 2*, *Figure 2—figure supplement 1*) were obtained as the average prediction accuracy across runs obtained on synthetic data by applying lower bound and upper bound position noise.

## Generating ground-truth data for framework tuning and comparison against other methods

NeuroPAL reagents OH15495 and OH15500 were used to generate ground-truth data. 3D image stacks were acquired following the guidelines provided in NeuroPAL manual (*Yemini et al., 2021*). Identities were annotated in image stacks using the example annotations provided in NeuroPAL manual. Individual channel image stacks were read in MATLAB, gamma and contrast were adjusted for each channel individually so that the color of cells in the RGB image formed by combining the individual channels matched as much as possible (perceptually) with the colors of cells in NeuroPAL manuals. To annotate identities in the 3D stack, Vaa3D software was used (*Peng et al., 2010*).

## Model comparison against previous methods

Detailed description of the methodology used for each method that our CRF_ID framework was compared against is provided in Appendix 1–Extended methods S2. Note, for fair comparisons, standard 3D OpenWorm atlas was used by all methods as the reference: either for defining positions of cells (used by registration methods) or for defining positional relationships among cells (used by the CRF_ID framework).

## Simulations for choosing landmark locations

Landmarks (cell with known identities) improve prediction accuracy by constraining the optimization problem as it forces the CRF_ID framework to choose optimal labels for all cells such that they preserve their positional relationships with the cells with fixed identities. However, choosing an optimal set of landmarks is difficult. This is because the combinatorial space of choosing landmarks is huge (~$10^{14}$ for 10 landmark cells out of 195 in head ganglion). Simulating each such combination and predicting identities is not computationally tractable. Thus, we asked which regions of the brain landmark cells should lie in. We divided the head ganglion region into three groups: anterior group consisting of anterior ganglion, middle group consisting of lateral, dorsal and ventral ganglion, and posterior group consisting of retrovesicular ganglion. Two hundred runs were performed for each group with 15 randomly selected landmarks in each run. We constrained the landmarks cells to lie in a specific group and assessed how well the landmarks in that group perform in predicting the identities of cells in other regions. Overall, landmarks in anterior and posterior groups performed badly in predicting identities of cells in posterior and anterior groups respectively. Landmarks in the middle group and landmarks spatially distributed throughout the head performed equally (*Figure 4—figure supplement 2*). We chose landmarks spatially distributed throughout the head due to practical advantages: spatially distributed landmarks can be easily identified manually in image stacks thus can be used as inputs to the CRF_ID framework. In contrast, cells in middle group are densely packed and may not be identified easily. We tested this using several reporter strains with GFP labeled cells. Further, landmarks should be reliably expressed across animals, should have known and verified expression patterns and should label neither too few cells (not useful) nor too many cells (difficult identification). Thus, we chose *unc-47* and *gcy-32* reporters for labeling landmarks.

## Construction of landmark strains

We constructed two different transgenic strains in which nine (GT290) and twelve (GT298) neurons, respectively, were labeled with the fluorescent protein CyOFP1 (*Chu et al., 2016*). Due its long Stokes shift, CyOFP1 can be excited by the same laser line as GCaMP, while emitting red-shifted photons. This conveniently allows us to perform three-color imaging on our two-channel confocal microscope. We designed an optimized version of the CyOFP1 gene using the *C. elegans* Codon Adapter (*Redemann et al., 2011*), which was then synthesized (Integrated DNA Technologies) and sub-cloned into a kanamycin-resistant vector to yield the pDSP11 plasmid. Our CyOFP1 construct contains two different nuclear localization sequences (NLS), SV40 NLS at the N-terminus and EGL-13 NLS at the C-terminus, a strategy which has been shown to more effective at trafficking recombinant proteins to the nucleus of worm cells (*Lyssenko et al., 2007*). The nuclear localization of the CyOFP1 protein allows us to unambiguously identify labeled cells in the densely packed head ganglion of *C. elegans*.

We tested two different labeling strategies in our study. The first used the promoter of the *unc-47* gene to drive expression CyOFP1 in all 26 GABAergic neurons of the worm (*McIntire et al., 1997*). As our study focused on the head ganglion, only the RMEL, RMER, RMEV, RMED, AVL, RIS, DD1, VD1, and VD2 neurons are labeled by this promoter in this region (Strain GT296, *Figure 4D* top panel). Our second strategy used the *unc-47* CyOFP1 construct in combination with a second driven by the promoter of the *gcy-32* gene, which is expressed in the AQR, PQR, and URX neurons (*Yu et al., 1997*), to label twelve cells in the head ganglion (Strain GT293, *Figure 4D* bottom panel). The DNA sequence of each promoter was amplified from N2 (wild type) genomic DNA and incorporated into a NotI-digested linear pDSP11 vector backbone using the NEBuilder HiFi DNA Assembly master mix (New England Biolabs) to yield the vectors pSC1 and pSC2. The following primers were used to amplify the promoters: *unc-47* Forward 5'- cagttacgctggagtctgaggctcgtcctgaatgatatgcCTGCCAATTTGTCCTGTGAATCGT-3' and Reverse 5'- gcctctcccttggaaaccatCTGTAATGAAATAAATG

TGACGCTGTCGT, *gcy-32* Forward 5'- cagttacgctggagtctgaggctcgtcctgaatgatatgcTTGTATAG TGGGAAATACTGAAATATGAAACAAAAAATATAGCTG-3' and Reverse 5'- gcctctcccttggaaaccatTC TATAATACAATCGTGATCTTCGCTTCGG-3'.

To make landmark strains pSC1 and pSC2 were injected into N2 strain to make GT290 and GT298. GT290 and GT298 strains can be crossed with any strain where cells need to be identified. Landmarks in these strains help in defining a coordinate system in head and also improve the accuracy of automatic annotation framework by constraining optimization problem. To make strain GT293 for whole-brain imaging experiments, AML70 was crossed with GT298; *lite-1(ce314)* was confirmed by sequencing. To make strain GT296 for whole-brain imaging experiments, AML32 was crossed with GT290 and subsequently crossed with KG1180, *lite-1(ce314)* was confirmed by sequencing.

## Whole-brain data analysis

All videos were processed using custom software in MATLAB for automatic segmentation and tracking of nuclei in whole-brain image stacks. Tracks for nuclei with minor tracking errors were corrected in post-processing steps. Tracks with large tracking errors were dropped from the data.

### Segmentation

Neurons were automatically segmented in image stacks using a Gaussian Mixture model based segmentation technique. Briefly, intensity local maxima are detected in images to initialize the mixture components and subsequently a 3D gaussian mixture model is fitted to the intensity profiles in image stacks using Expectation-Maximization (EM) algorithm. The number of components in the model and the ellipsoidal shape of each component determines the number of nuclei segmented and their shapes.

### Tracking

Custom software was used for tracking cells. Briefly, segmented nuclei at each timepoint in image stacks are registered to a common reference frame and temporally nearby frames to produce globally and locally consistent matching. Based on these matchings, consistency constraints such as transitivity of matching were imposed in the post-processing step to further improve tracking accuracy. A custom MATLAB GUI was used to quickly manually inspect the accuracy of tracking. Tracks of cells with minor tracking errors were resolved using semi-automated method.

### Cell identification

Identities were predicted using the CRF_ID framework with positional features (Appendix 1– Extended methods S1) using the data-driven atlas. Landmarks cells with known identities were identified in the CyOFP channel and were provided as input to the framework to achieve higher accuracy.

### Identification of stimulus tuned neurons

To identify stimulus tuned neurons, the power spectrum of activities of all cells within the stimulus application window (100 s – 180 s) was calculated using 'fft' function in MATLAB. Cells that showed significant power (>0.08) at 0.1 Hz (due to 5 s on 5 s off stimulus, *Figure 5*) were selected. This criterion identified all cells except two with low response amplitude to the stimulus; however, the response could be manually seen in the video. Thus, these cells were manually selected.

### PCA and Sparse PCA

Principal Component analysis (PCA) of neuron activity time-series data was performed using in-built functions in MATLAB. Sparse Principal component analysis (SPCA) was performed using freely available MATLAB toolbox (*Sjöstrand et al., 2018*).

### Neuron activities correlation to animal motion

To ascertain that the motion of the worm in device has signatures of wave-propagation in freely moving animals, we looked for phase shift in the velocity of the different regions of the animal in the device (similar to phase shift in curvature of body parts of animals seen in freely moving animals

*Stephens et al., 2008*). To calculate the velocity, displacement of randomly selected cells along the anterior-posterior axis of the animal was calculated (*Figure 5—video 2*) based on the tracking of cells. Cell displacements were smoothed using Savitzky-Golay filter. Subsequently, velocity of each cell was calculated by differentiating the displacement of each cell.

Mutual information (MI) of the obtained velocity signal was calculated with (1) stimulus tuned neurons, (2) neurons with significant weights in sparse principal components 1–3, and (3) remaining cells. MI analysis requires estimating the joint probability density of velocity and neuron activity. We used the kernel density estimation (KDE) method to do so. KDE method used Gaussian kernel with bandwidth parameters (that specify the variance of gaussian kernel) set to [0.05, 0.05]. Cells grouped in SPC2 always had the largest mutual information with velocity regardless of the choice of the bandwidth parameter.

## Runtime comparison

To compare optimization runtimes of CRF and registration-based method CPD (*Myronenko and Song, 2010*), synthetic data was generated using OpenWorm atlas as described previously with randomly selected 10, 20, 50, and 130 cells. Annotation was performed using only positional relationship features. Full head ganglion OpenWorm atlas (206 cells) was used for annotation. Simulations were run on standard desktop computer (Intel Xeon CPU E5-1620 v4 @3.5 GHz, 32 GB RAM).

## Statistical analysis

Standard statistical tests were performed using Paired Comparisons App in OriginPro 2020. Details regarding the tests (sample size, significance, method) are reported in figure legends. Following asterisk symbols are used to denote significance level throughout the manuscript - * ($p < 0.05$), ** ($p < 0.01$), *** ($p < 0.001$). Significance level not indicated in figures implies not significantly different (n.s).

## Code and data availability

Code and data used in this study can be accessed at https://github.com/shiveshc/CRF_Cell_ID.git. This repository contains the following (1) All code and individual components necessary for using CRF_ID framework to annotate cells in new data, visualize results, and build new atlases based on annotated data (2) Code to reproduce results for comparison shown against other methods in this study, and (3) all raw datasets used in this study as well as human annotations created for those datasets except whole-brain imaging datasets.

# Acknowledgements

The authors thank the Hobert Lab for providing NeuroPAL strains. The authors acknowledge the funding support of the U.S. NIH (R21DC015652, R01NS096581, NIH R01GM108962, and R01GM088333) and the U.S. NSF (1764406 and 1707401) to HL. Some nematode strains used in this work were provided by the Caenorhabditis Genetics Center (CGC), which is funded by the NIH (P40 OD010440), National Center for Research Resources and the International *C. elegans* Knockout Consortium. SC and HL designed the algorithm, experiments, and methods. SC collected ground-truth validation data. SC, SL collected whole-brain imaging data. SC, SL, YL, and DSP developed strain with neuronal landmarks; SC, SL, YL, and HL analyzed the data. SC and HL wrote the paper. The authors declare no competing interest. All raw datasets (except whole-brain imaging datasets) used in this study as well as ground-truth human annotations created for those datasets are available at https://github.com/shiveshc/CRF_Cell_ID/tree/master/Datasets.

# Additional information

## Funding

| Funder | Grant reference number | Author |
|---|---|---|
| National Institutes of Health | R21DC015652 | Hang Lu |
| National Institutes of Health | R01NS096581 | Hang Lu |

| National Institutes of Health | R01GM088333 | Hang Lu |
| National Science Foundation | 1764406 | Hang Lu |
| National Science Foundation | 1707401 | Hang Lu |
| National Institutes of Health | R01GM108962 | Hang Lu |
| National Institutes of Health | P40OD010440 | Hang Lu |

The funders had no role in study design, data collection and interpretation, or the decision to submit the work for publication.

## Author contributions

Shivesh Chaudhary, Conceptualization, Resources, Data curation, Software, Formal analysis, Validation, Investigation, Visualization, Methodology, Writing - original draft, Writing - review and editing; Sol Ah Lee, Yueyi Li, Investigation; Dhaval S Patel, Methodology; Hang Lu, Conceptualization, Resources, Formal analysis, Supervision, Funding acquisition, Validation, Investigation, Visualization, Methodology, Writing - original draft, Project administration, Writing - review and editing

## Author ORCIDs

Shivesh Chaudhary (iD) https://orcid.org/0000-0002-1928-0933
Hang Lu (iD) https://orcid.org/0000-0002-6881-660X

## Decision letter and Author response

Decision letter https://doi.org/10.7554/eLife.60321.sa1
Author response https://doi.org/10.7554/eLife.60321.sa2

# Additional files

## Supplementary files

• Transparent reporting form

## Data availability

All data generated or analysed during this study are included in the manuscript and supporting files. Source data files are provided at https://github.com/shiveshc/CRF_Cell_ID.git (copy archived at https://archive.softwareheritage.org/swh:1:rev:aeeeb3f98039f4b9100c72d63de25f73354ec526/).

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

## Appendix 1

## Extended methods

S1: Detailed description of CRF_ID model for cell annotation

### S1.1 Advantages of CRF_ID framework

There are several advantages of CRF-based formulation for automatic cell annotation as discussed below.

1. The major advantage of CRF framework is that arbitrary feature relationships $f_k$ among non-independent observations $\mathbf{x}$ can be specified (*Sutton and McCallum, 2010*). This enables optimizing user-defined arbitrary features. Depending on the scope of potential functions $\Phi_c$, these features may be cell specific that is unary features ($\Phi_c = \Phi_i$), pairwise features ($\Phi_c = \Phi_{ij}$) or higher order features. For example position of cells in ganglion, position related to a landmark cell with known identity, color etc.

2. Since we are modeling a probability distribution over labels assigned to neurons, a set of candidate list of names can be generated for each neuron. A straightforward way to do this is by using the marginal probabilities of labels assigned to each cell. Once the normalization constant $Z$ has been calculated, marginal probability of cell $i$ taking a label $y_i$ can be obtained as $P\left(\frac{y_i}{\mathbf{x}}\right) = \sum_{j \in V, j \neq i} P(\mathrm{y}/\mathbf{x})$. Other computationally efficient methods to estimate the marginal probabilities using eigenvector of potential functions *Leordeanu and Hebert, 2005*; *Leordeanu and Hebert, 2009* have been proposed as well. We propose an alternative method to generate such list (see Appendix 1–Extended methods S1.5) taking into account the undetected neurons in image stack.

3. With log-linear parameterization of feature functions as in (*Equation 2*), CRF models belong to exponential family models (*Wainwright and Jordan, 2007*) or max-entropy models. Thus, the joint probability distribution over labels assigned to neurons that we infer is maximally unbiased (maximum entropy) subject to some empirical constraints (sufficient statistics to define the probability distribution). In our case, these empirical constraints are geometrical relationships among cell positions. Interestingly, the maximum entropy nature of the objective function in our model also makes it very similar to the entropy regularized optimal transport problems (*Solomon et al., 2016*; *Nitzan et al., 2019*).

4. CRF framework is a trainable algorithm (*Taskar et al., 2004*; *Taskar et al., 2003*). Thus, if annotated image stacks are available, the weights $\lambda_k$ of the feature functions can be optimized directly such that the cell labels in annotated image stacks match the predicted labels. Further, we show that annotated experimental data can be utilized by building a data-driven atlas and feature functions can updated based on data-driven atlas (S1.7). For pairwise feature functions in our model, building such data-driven atlas requires cheap mathematical operations (averaging).

### S1.2 Features in CRF_ID-based annotation framework

During manual annotation of neuron identities in images, researchers use intuitive features such as positions of neurons in image and atlas, positional relationships among neurons, proximity of neurons to one another, neuron with known identities such as neurons expressing specific markers etc. In this section, we describe how such intuitive features are encoded in our model.

### S1.2.1 Unary potentials – positions along AP axis

In empirical data, we observed that anterior-posterior (AP) positional relationships among the cells are most stereotypical (*Figure 2—figure supplement 6C,D*) and consistent with the 3D atlas. Further since, x-y sampling of image stacks is much higher than z sampling, AP axis' sampling is always higher than LR and DV axes. Thus, positions along AP axes detected by segmentation method are less noisy. Thus, we included a feature based on positions of cell along AP axis as unary feature.

$$f_a(m; i, \boldsymbol{x}) = \exp\left(-\frac{||x_{i,AP} - y_{m,AP}||_2}{\sigma_u^2}\right) \tag{5}$$

Here, $x_{i,AP}$ is the position of cell $i$ along AP axes in data and $y_{m,AP}$ is the position of cell with label

$m$ in atlas. $\sigma_u$ is the bandwidth parameter. To account for scale differences between image and atlas, positions along AP axis are normalized first in image and atlas. Low value of $\sigma_u$ greatly penalizes the deviation of cell position along AP axis in image from atlas position thus favoring that labeling should preserve positions along AP axis. Large value of $\sigma_u$ decreases the effect of deviation of cell position along AP axis. Thus, this feature restricts the assignment of a particular label to certain cells along AP axes making the labeling AP consistent. Low values of $\sigma_u$ also increases the influence (weight) of this unary potential compared to the pairwise potentials in labeling. We set this parameter as 1 to set equal influence of unary and pairwise potentials.

### S1.2.2 Pairwise potentials – binary positional relationships

These features encode that the labels assigned to neuron in an image should preserve the relative positional relationships among them in the atlas. for example if neuron $i$ is to the left of neuron $j$ in the image stack and these neurons are assigned labels $m$ and $n$ respectively, then the neuron with label $m$ in the atlas should be to the left of neuron with label $n$. Similar constraints can also be applied on anterior-posterior relationship and dorsal-ventral relationship of neurons in the image stack. Let $\left[x_{i,AP}, x_{i,LR}, x_{i,DV}\right]$ denote the coordinate of neuron $i$ in the image along anterior-posterior (AP), left-right (LR) and dorsal-ventral (DV) axes, respectively. Similarly, $\left[y_{m,AP}, y_{m,LR}, y_{m,DV}\right]$ be the coordinates of neuron with label $m$ in the atlas. The feature functions $f_{AP}$ is defined as

$$f_{AP}(m,n;i,j,\mathbf{x}) = \lambda_{AP}\begin{cases} 1, & (x_{i,AP} - x_{j,AP})(y_{m,AP} - y_{n,AP}) > 0 \\ 0, & (x_{i,AP} - x_{j,AP})(y_{m,AP} - y_{n,AP}) < 0 \end{cases} \tag{6}$$

Thus, this feature implies that if atlas labels $m$ and $n$ assigned to neurons $i$ and $j$ in image are consistent with the AP positional relationship of cells $i$ and $j$, then the feature value is 1 else 0. Note that the feature values for cells $i$ and $j$ is same (1 or 0) irrespective of the labels $m$ and $n$ assigned to cells. This is true only if annotation is performed using a static atlas or an atlas built from only one data source. We expand more on this in section S1.7 (Building data-driven atlases) and explain how label dependent feature functions are formed with the availability of empirical hand-annotated datasets.

Similarly, features are defined for left-right and dorsal-ventral relationships, $f_{LR}$ and $f_{DV}$, respectively.

$$f_{LR}(m,n;i,j,\mathbf{x}) = \lambda_{LR}\begin{cases} 1, & \left(x_{i,LR} - x_{j,LR}\right)\left(y_{m,LR} - y_{n,LR}\right) > 0 \\ 0, & \left(x_{i,LR} - x_{j,LR}\right)\left(y_{m,LR} - y_{n,LR}\right) < 0 \end{cases} \tag{7}$$

$$f_{DV}(m,n;i,j,\mathbf{x}) = \lambda_{DV}\begin{cases} 1, & \left(x_{i,DV} - x_{j,DV}\right)\left(y_{m,DV} - y_{n,DV}\right) > 0 \\ 0, & \left(x_{i,DV} - x_{j,DV}\right)\left(y_{m,DV} - y_{n,DV}\right) < 0 \end{cases} \tag{8}$$

Here, $\lambda_{AP}$, $\lambda_{LR}$, and $\lambda_{DV}$ are hyperparameters in the model that weigh positional relationship features against other pairwise features in the model. We set these parameters as 1 to give equal weightage to all features.

### S1.2.3 Pairwise potentials – proximity relationships

While manually annotating images by comparing positions of neurons to atlas, researchers often use proximity relationship among neurons that is if neuron $i$ is anatomically far from neuron $j$ then the identities to be assigned to these neurons from atlas should not belong to neighboring or nearby neurons. To encode such intuition in the model, we include proximity feature similar to the Gromov-Wasserstein discrepancy used in shape matching (*Solomon et al., 2016*; *Mémoli, 2011*)

$$f_{proximity}(m,n;i,j,\mathbf{x}) = -\lambda_{proximity}||d(x_i,x_j) - d(y_m,y_n)||_2 \tag{9}$$

Here, $d(x_i,x_j)$ is any distance measure between neurons $i$ and $j$ in the image stack and $d(y_m,y_n)$ is the same distance measure between neurons with labels $m$ and $n$ in the atlas. We use geodesic distances between cells. To calculate geodesic distances, graphs were constructed by connecting each neuron to its nearest six neighbors. $\lambda_{proximity}$ is hyperparameter that weighs proximity relationship feature function against other features in the model. We set $\lambda_{proximity}$ as 1.

We compared geodesic distances rather than Euclidean $\mathcal{L}_2$ distances because geodesic distances are invariant to spatial scale in the image and atlas. This is critical since the scale of spatial distribution of neurons in the atlas is very different (much lower) than those in the images. Also, the spatial scale of distribution of cells may vary across images, depending on the size of worm in images; however, geodesic distances among neurons should be preserved.

### S1.2.4 Pairwise potentials – angular positional relationships

Relative positional relationship features described in S1.2.2 encode information about positional relationships along axes independently that is each feature contains information about positional relationship along one axis only. For example $f_{AP}$ encodes whether neuron $i$ is anterior to or posterior to neuron $j$ and how the labels should be assigned to these cells. A feature that simultaneously accounts for positional relationships along all axes may additionally help in determining identities of neurons. Such a feature could be formed by multiplying AP, LR, and DV positional relationship features. However, a multiplied feature will still contain binary information only about whether neuron $i$ is anterior, dorsal and to the right of neuron $j$ or not. It would not tell anything about fine scale directional relationships. Thus, we formulated an angular relationship feature. Let $p'_i$ and $p'_j$ be the 3D vectors associated with coordinates of neurons $i$ and $j$ in the image stack. Also let $p''_m$ and $p''_n$ be the 3D vectors associated with coordinates of neurons with labels $m$ and $n$ in the atlas. Then the feature is defined as

$$f_{angle}(m,n;i,j,\mathbf{x}) = \frac{\lambda_{angle}\left(1 + \left(\frac{p'_i - p'_j}{\|p'_i - p'_j\|_2} \cdot \frac{p''_m - p''_n}{\|p''_m - p''_n\|_2}\right)\right)}{2} \tag{10}$$

Thus, if the vector $\overrightarrow{p'q'}$ aligns perfectly with the vector $\overrightarrow{p''q''}$, $f_{angle} = 1$ and $f_{angle} = 0$ if the vectors point in completely opposite directions. This feature encodes directional agreement of the labels $m$ and $n$ assigned to neurons $i$ and $j$. Here $\lambda_{angle}$ is hyperparameter that weighs angular relationship feature function against other features in the model. We set $\lambda_{angle}$ as 1.

### S1.3 Defining AP, LR, and DV axes

To compare positional relationships among neurons in image and atlas, it is necessary to define anterior-posterior (AP), left-right (LR), and dorsal-ventral (DV) axes in image as well in atlas. 3D coordinates of neurons along these axes are then used to define features described above. We use two methods to define these axes. In method 1, we use Principal Component Analysis to obtain these axes. Let $\mathbf{p} = [p_1, \ldots p_N] \in \mathbb{R}^{3 \times N}$ be the centered coordinates (zero mean) of $N$ neurons detected in the image stack or atlas. Then the principal components correspond to the eigenvectors of the matrix $\mathbf{pp}^T$. Since the spatial spread of neurons in image as well as in atlas is maximum along AP axes, the first principal component (eigenvector corresponding to maximum eigenvalue) always corresponds to AP axis. Second and third PCs can be assigned to LR and DV axes depending on the orientation of worm in image as described below. Due to rotation of worm about AP axis, the second and third eigenvectors may not always correspond to LR and DV axes. Thus, we designed two methods for these different scenarios –

1. **Worm lying on LR axis** – In this case LR axis is assigned to the third eigenvector. This is because z-sampling of image stacks is much smaller compared to x-y sampling. Thus, the spread of neurons along LR axis is smallest. Figure below shows axes obtained using PCA.
2. **Worm rotated about AP axis** – In this case we developed an alternative method. First, we define LR axis. To do so we use left-right pair of neurons that are easily detectable in image stack. We used RMEL-RMER for landmark strain (*Figure 4*) and RMDVL-RMDVR for NeuroPAL strain (*Figure 6E, F*). Using these neuron-pair coordinates, an LR vector, $\overrightarrow{lr}$ was defined as $\frac{p_r - p_l}{\|p_r - p_l\|_2}$. Next AP vector, $\overrightarrow{ap}$ was determined by solving constrained optimization problem

$$\vec{ap} = \max(\vec{ap}.\vec{v}_1)$$
$$\text{s.t.} \quad \vec{ap}.\vec{lr} = 0 \ and \ \left\|\vec{ap}\right\|_2 = 1$$

(11)

That is a unit vector which is orthogonal to LR vector and in the direction of first principal component $\vec{v}_1$. Next, $\vec{dv}$ vector is obtained by defining a vector orthogonal to both $\vec{ap}$ and $\vec{lr}$ vectors.

Finally, we check the consistency of $\vec{ap}$, $\vec{lr}$, $\vec{dv}$ vectors that is these vectors should point to the anterior, right and ventral of the worm and should satisfy cross product rule to constitute a valid coordinate system. This is necessary because PCA axes are determined up to a multiplication factor of -1 that is coordinate system specified by the principal components (PC1, PC2, PC3) is same as the coordinate system specified by (-PC1, -PC2, -PC3). Thus, a user input is taken in the framework while defining axes. Users can easily click on any neuron in the anterior portion and the posterior portion of the worm image, when asked to do so, to specify PC1 direction.

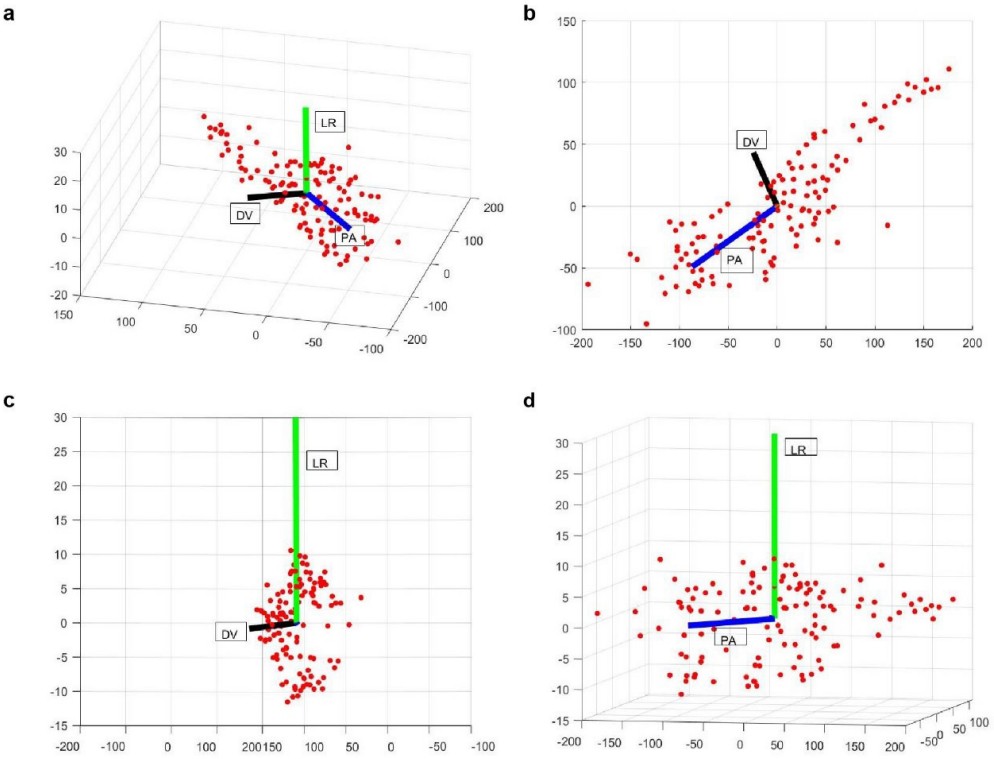

**Appendix 1—figure 1.** Examples of PA (blue), LR (green), and DV (black) axes generated automatically in a whole-brain image stack. Here red dots correspond to the segmented nuclei in image stack. Shown are 3D view (a), XY (b), YZ (c), and XZ (d) views of the image stack.

## S1.4 Inferring neuron identities

To infer most probable identity of neurons, energy function in (*Equation 4*) is to be maximized. Exact inference techniques for maximizing energy functions over arbitrary graph structures, such as the fully connected graph structure of our model, are not available (*Kohli et al., 2009*). Thus, we use a commonly used approximate inference method that has been used successfully in past for several applications, Loopy Belief Propagation (LBP) (*Murphy et al., 1999*; *Ikeda et al., 2004*) to infer optimal labeling that is the maximum of joint-probability distribution $p(\frac{y}{x})$ as well as the marginal probabilities of labels assigned to each node. We implement our model in MATLAB using an open source package for undirected graphical models (*Schmidt, 2007*).

## S1.5 Resolving duplicate assignments with label consistency score

In general, we define pairwise feature functions in the model such that it penalizes duplicate assignments of any label $m$ to cells $i$ and $j$ in image as follows

$$f_{angle}(m,m;i,j,\mathbf{x}) = -\infty \tag{12}$$

Even with this measure, we still see some duplicate assignments when the LBP optimization converges. To resolve such assignments, we mark all cells that were not assigned duplicate labels as assigned. Next, we calculate a label consistency score for each cell that was assigned a duplicate label (examples shown in *Figure 1—figure supplement 1C*). This score measures how consistent the current label assigned to the cell is in terms of preserving its positional relationships with other cells that were not assigned any duplicate labels (i.e. cells marked as assigned). Among all the cells that were assigned same duplicate label, the cell with the highest consistency score is assigned the duplicate label and marked as assigned. Remaining cells are marked as unassigned. After resolving all such duplicate assignments for all labels, optimization is run again only for the unassigned cells while keeping the identities of other cells fixed. We calculate the label consistency score for each cell as follows

$$bin.pos.rel.(i) = \sum_{j \in v'} f_{AP}(m,n;i,j,\mathbf{x}) + f_{LR}(m,n;i,j,\mathbf{x}) + f_{DV}(m,n;i,j,\mathbf{x}) \tag{13}$$

$$ang.rel.(i) = \sum_{j \in v'} f_{angle}(m,n;i,j,\mathbf{x}) \tag{14}$$

$$proximity\,rel.(i) = \sum_{j \in v'} f_{proximity}(m,n;i,j,\mathbf{x}) \tag{15}$$

$$consistency\,score = bin.pos.rel. + ang.rel. + proximity\,rel. \tag{16}$$

Here, $i$ is a cell that was assigned a duplicate label and $V'$ denotes the set of all other cells that were not assigned duplicate labels. Intuitively, correctly predicted cells should have higher consistency score. This is because correctly predicted cells will preserve their relationship with other correctly predicted cells and won't preserve their relationship with incorrectly predicted cells. In contrast, incorrectly predicted cells will neither preserve their relationship with correctly predicted cells nor with incorrectly predicted cells thus having a lower consistency score. This was observed in simulations as well (*Figure 1—figure supplement 1D*). Thus, label consistency score also serves as a good criterion for sorting candidate list of labels predicted by the framework.

Label consistency score is combination of features in the model that define the probability distribution $p(y/\mathbf{x})$ in our model. Thus, another way to look at the label consistency score is that it tries to maximize the pseudo-loglikelihood (*Sutton and McCallum, 2007*; *Hyvärinen, 2006*) of labels to be assigned to all cells with duplicate labels conditioned on the labels of other cells that were not assigned duplicate labels that is $\sum_{i \in dup} \log\left(p(y_i; y_{V'}, \mathbf{x})\right)$. Here $dup$ denotes the set of all cells that were assigned duplicate labels.

## S1.6 Simulating missing cells and generating candidate name list for each cell

The 3D atlas of *C. elegans* neurons we used is freely available (*Szigeti et al., 2014*). There are 195 cells in head ganglion in this atlas, that is the label list from which identities are to be assigned to cells in data has 195 elements. However, empirically we detect only ~120–130 neurons in whole-brain image stacks. Remaining neurons are undetected due to either no/low expression levels of fluorophore in these cells or false-negatives in automated detection algorithm. Approximately similar number of neurons were detected by other labs as well (*Kato et al., 2015*). Further, which cells are undetected is not known a priori. Thus, to take into account missing cells while annotating identities with the model, we define a hidden variable $\mathbf{h} \in \{0,1\}^N$ that specifies the cells missing in images that

is $\mathbf{h_k} = 1$ if cell with label $k$ in atlas is missing in the image. Here $N$ is the number of labels/cells in atlas. With the hidden variable, we model the joint probability distribution $P\left(\frac{y,\mathbf{h}}{\mathbf{x}}\right)$ as

$$P\left(\frac{y,\mathbf{h}}{\mathbf{x}}\right) = \frac{1}{Z}\prod_{i \in V}\Phi_i(m;i,\mathbf{x})\prod_{e_{ij} \in \mathcal{E}}\Phi_{ij}(m,n;i,j,\mathbf{x})\prod_{i \in V}\Phi_i(m,\mathbf{h})\Phi(\mathbf{h},\mathbf{x}) \tag{17}$$

Here $Z = \sum_{y \in \mathcal{Y},\mathbf{h}}\prod_{i \in V}\Phi_i(y_i;i,\mathbf{x})\prod_{e_{ij} \in \mathcal{E}}\Phi_{ij}(y_i,y_j;i,j,\mathbf{x})\prod_{i \in V}\Phi_i(y_i,\mathbf{h})\Phi(\mathbf{h},\mathbf{x})$. Unary potential functions $\Phi_i$ and pairwise potential functions $\Phi_{ij}$ are same as described in S1 and S1.2. Potential function $\Phi_i(m,\mathbf{h})$ captures the dependencies between label $m$ assigned to cell $i$ when the hidden variable is $\mathbf{h}$. We set the potential function as

$$\Phi_i(m,\mathbf{h}) = \begin{cases} 0 & if\ \mathbf{h}_m = 1 \\ 1 & if\ \mathbf{h}_m = 0 \end{cases} \tag{18}$$

Thus, the potential specifies that if $m^{th}$ label in atlas is missing in the image then that label cannot be assigned to any cell in the image. Further, if $m^{th}$ label is not missing in the image then each cell has equal potential to be assigned that label which will be updated based on positional relationship potentials. Further, the potential function $\Phi_i(\mathbf{h},\mathbf{x})$ captures the dependence between the observed image $\mathbf{x}$ and the hidden variable $\mathbf{h}$. Defining this potential function is not trivial for whole-brain images since predicting which cells are missing just based on observed 3D whole-brain image stack is difficult. However, in specific cases confidence values of each label missing in the image may be available, denoted by $\beta_k$, $0 \le \beta_k \le 1$. Such confidence values may be specified based on prior knowledge for example based on expression pattern of fluorophore labeling cells in strain, cells that have low detection probability may have higher parameter $\beta_k$. Using these confidence values, we define $\Phi(\mathbf{h},\mathbf{x})$ as

$$\Phi(\mathbf{h},\mathbf{x}) = \prod_{k=1}^{N}\beta_k^{h_k}(1-\beta_k)^{(1-h_k)} \tag{19}$$

Our goal is to calculate $P\left(\frac{y}{\mathbf{x}}\right)$ which can be obtained by marginalizing (*Equation 17*) over the hidden states $\mathbf{h}$. However, since the number of elements in the space of $\mathbf{h}$ is huge, marginalizing as well as calculating the normalization constant $Z$ is not tractable. But, for specific cases such as we describe below, the calculation can be simplified.

In the absence of any prior information about which labels are missing in images for whole-brain imaging case, we assigned equal confidence value for each label missing in the image that is $\beta_k = \beta$. Thus, for a fixed number $P$ of missing cells in image, $\Phi(\mathbf{h},\mathbf{x}) = \beta^P(1-\beta)^{N-P}$. A consequence of defining potential function this way is that for a given number of missing cells in image, each combination of missing labels in atlas is equally probable as long as labels of missing cells are not assigned to any cell in image

$$P\left(\frac{\mathbf{h}}{y=y,\mathbf{x}}\right) = \frac{P\left(\frac{y,\mathbf{h}}{\mathbf{x}}\right)}{\sum_{\mathbf{h}}P\left(\frac{y,\mathbf{h}}{\mathbf{x}}\right)} \tag{20}$$

$$= \frac{\prod_{i \in V}\Phi_i(m;i,\mathbf{x})\prod_{e_{ij} \in \mathcal{E}}\Phi_{ij}(m,n;i,j,\mathbf{x})\prod_{i \in V}\Phi_i(m,\mathbf{h})\Phi(\mathbf{h},\mathbf{x})}{\sum_{\mathbf{h}}\prod_{i \in V}\Phi_i(m;i,\mathbf{x})\prod_{e_{ij} \in \mathcal{E}}\Phi_{ij}(m,n;i,j,\mathbf{x})\prod_{i \in V}\Phi_i(m,\mathbf{h})\Phi(\mathbf{h},\mathbf{x})} \tag{21}$$

$$= \frac{\prod_{i \in V}\Phi_i(m,\mathbf{h})\Phi(\mathbf{h},\mathbf{x})}{\sum_{\mathbf{h}}\prod_{i \in V}\Phi_i(m,\mathbf{h})\Phi(\mathbf{h},\mathbf{x})} = \frac{1}{|\mathbf{h}'|} \tag{22}$$

Here, $\mathbf{h}' \subset \{0,1\}^N$ such that $\mathbf{h}_k = 0 \forall k^{th}$ *label assigned to cells in y* and $\sum_k \mathbf{h}_k = P$. Further, $|\mathbf{h}'|$ is the number of elements in $\mathbf{h}'$. Thus, we can randomly and uniformly sample $\mathbf{h}$ and keep it fixed while predicting $P\left(\frac{y}{\mathbf{x}}\right)$. Our goal is to calculate $P\left(\frac{y}{\mathbf{x}}\right)$. We do so as

$$\arg\max_{y \in \mathcal{Y}} P\left(\frac{y}{\mathbf{x}}\right) = \sum_{\mathbf{h}} \arg\max_{y \in \mathcal{Y}} P\left(\frac{y, \mathbf{h}}{\mathbf{x}}\right) \tag{23}$$

Therefore, we randomly select $P$ cells in atlas that are considered to be missing in the image data. These $P$ cells are selected uniformly across different regions of the head ganglion following the discussion above. Labels of these $P$ cells are removed from the atlas i.e. the list of possible labels that can be assigned to neurons, and identities of cells are predicted using the remaining atlas. This process is repeated ~1000 times to sample multiple possible combinations of $\mathbf{h}$. Finally, a candidate list of names is generated using (*Equation 23*) that is compiling a list of optimum labels predicted for each cell in each run (by maximizing $P(\frac{y, \mathbf{h}}{\mathbf{x}})$ in each run) and choosing the top frequent labels for each cell across all runs.

## S1.7 Building data-driven consensus atlas

Here we describe the procedure and the intuition behind building data-driven consensus atlas in our framework. We also describe the intuition behind why it is computationally more efficient than building data-driven atlases for registration methods.

First, we describe how features in the model are defined using a static atlas such as OpenWorm atlas and then extend the formulation to building and using data-driven atlas. Positional relationships among cells based on the OpenWorm atlas are stored as matrices of size $N \times N$ where $N$ is the number of cells in atlas. Each cell in matrix records the positional relationship between a pair of cells observed in the atlas. For example, for AP positional relationship matrix, if a column cell such as RMEL is anterior to a row cell such as AIZR in atlas, then the corresponding cell in matrix will denote 1 and otherwise 0. Here, 1 implies that according to the prior information available from Open-Worm, cell RMEL is observed to be anterior to cell AIZR with 100% probability. Let $F_{AP} \in R^{N \times N}$ be the AP positional relationship matrix and $m, n$ be labels in atlas. Then,

$$F_{AP}(m,n) = \begin{cases} 1, & n > m \, along \, AP \\ 0, & n < m \, along \, AP \end{cases} \tag{24}$$

Note that $F_{AP}(n,m) = 1 - F_{AP}(m,n)$. Using the matrix $F_{AP}$, the AP positional relationship feature function $f_{AP}(m,n;i,j,\mathbf{x})$ for cells $i$ and $j$ (as described in S1.2.2) can be defined as

$$f_{AP}(m,n;i,j,\mathbf{x}) = \begin{cases} F_{AP}(m,n) & j > i \, along \, AP \\ F_{AP}(m,n) & j < i \, along \, AP \end{cases} \tag{25}$$

Here, $f_{AP}(m,n;i,j,\mathbf{x})$ denotes the AP positional relationship feature for assigning labels $m$ and $n$ in atlas to cells $i$ and $j$ in image. Note that *Equation 25* is consistent with *Equation 6* as shown below by expanding $F_{AP}(m,n)$ and $F_{AP}(n,m)$ terms using *Equation 24* in *Equation 25*.

$$f_{AP}(m,n;i,j,\mathbf{x}) = \begin{cases} 1, & n > m \cap j > i \, along \, AP \\ 0, & n < m \cap j > i \, along \, AP \\ 1, & m > n \cap j < i \, along \, AP \\ 0, & m < n \cap j < i \, along \, AP \end{cases} \tag{26}$$

Thus, the AP positional relationship matrix $F_{AP}$ stores the prior knowledge available on anterior-posterior relationships among cells. Since, $F_{AP}$ is built using only a single data source that is Open-Worm atlas, all elements in the matrix are either 1 or 0. This implies that for all pairs of cells $m$ and $n$ in atlas, the $F_{AP}$ matrix says that cell $n$ is anterior to cell $m$ with either 100% probability or 0 probability. Also, note that since $F_{AP}$ consists of only 1's and 0's, the AP positional relationship feature, $f_{AP}(m,n;i,j,\mathbf{x})$, also consists of only 1's and 0s, thus it is independent of labels $m$ and $n$ assigned to cells $i$ and $j$.

In contrast to using a single data source, if additional information is available in the form of anno-tated experimental datasets, the prior knowledge on anterior-posterior relationships can be updated. For example it may be possible that cell $n$ is observed to be anterior to cell $m$ with 80% probability in annotated datasets (e.g. in 8 out of 10 experimental datasets) and posterior to cell $n$

with 20% probability. This empirically observed AP positional relationships among cells can be updated in matrix $F_{AP}$ as

$$F_{AP,data-driven}(m,n) = \begin{cases} w_{mn} & n > m\,along\,AP \\ 1 - w_{mn} & n < m\,along\,AP \end{cases} \tag{27}$$

Here, $w_{mn}$, $0 \le w_{mn} \le 1$ is the fraction of annotated datasets in which labels $m$ is annotated to the anterior of label $n$. Thus, instead of specifying a hard constraint based on a single data source static atlas (as in *Equation 24*), $w_{mn}$ specifies a soft constraint based on positional relationships observed in experimental annotated datasets. Formally, $w_{mn}$ is defined as

$$w_{mn} = \frac{1}{D}\sum_{d=1}^{D} I(m,n;\mathbf{x},d) \tag{28}$$

$$I(m,n;\mathbf{x},d) = \begin{cases} 1 & x_n^d > x_m^d \\ 0 & x_n^d < x_m^d \end{cases} \tag{29}$$

Here, $d \in \{d_i\}_1^D$ denotes an annotated data set, $D$ is the total number of annotated datasets used to build atlas, $x_m^d \in \mathbf{x}$ and $x_n^d \in \mathbf{x}$ are the coordinates along AP axis of cells annotated labels $m$ and $n$ in dataset $d$. *Equation 26* can be seen as a generalization of *Equation 24*. Further, the AP positional relationship feature function $f_{AP}(m,n;i,j,\mathbf{x})$ can be updated based on data-driven atlas using *Equation 25*. The above discussion can be similarly extended to LR, DV relationships, angular relationship, and proximity relationship.

Similarly, for angular relationship feature, instead of using a fixed vector $\overrightarrow{p''q''}$ (refer to *Equation 10*) between cells $m$ and $n$ that is provided in static atlas, we use an average vector obtained from annotated experimental datasets.

$$\overrightarrow{p''q''}_{data-driven} = \frac{1}{D}\sum_{d=1}^{D} \frac{p''_{m,d} - p''_{n,d}}{||p''_{m,d} - p''_{n,d}||_2} \tag{30}$$

Here, $p''_{m,d}$ and $p''_{n,d}$ are position coordinates of cells with labels $m$ and $n$ in data $d$. Similarly, for proximity relationship feature (refer to *Equation 9*), instead of using a fixed distance between cells with labels $m$ and $n$ in atlas, we use an average distance obtained from annotated experimental datasets.

$$d(y_m, y_n)_{data-driven} = \frac{1}{D}\sum_{d=1}^{D} d(y_{m,d}, y_{n,d}) \tag{31}$$

Here, $d(y_m, y_n)$ is the distance between cells with labels $m$ and $n$ in data $d$.

The key difference between registration-based methods and our framework in building data-driven atlases lies in the underlying methodology used by these methods to annotate cells. Registration methods annotate cells by maximizing extrinsic similarity between images and atlas. This requires the pre-alignment of spaces in which the image and the atlas exist. Thus, building data-driven atlas for registration-based methods also requires pre-alignment of all annotated images to the same space. This is typically done by simultaneously registering or block-wise registering all annotated images which requires solving several constrained regression problems. In contrast, our framework annotates cells by maximizing intrinsic similarity between images and atlas which is independent to the pre-alignment of images to same space. Thus, in our framework positional relationship features can be calculated for each annotated dataset in its own space and subsequently the features are aggregated together by simple averaging operations, which is computationally efficient, to build data-driven atlases.

## S1.8 Discussion on previous methods using registration to annotate cells

*Qu et al., 2011* and *Long et al., 2009* annotate cells in *C. elegans* images by spatially deforming an atlas with known cell identities and registering it to the image data. Similar method was proposed

by *Aerni et al., 2013*. Here, several other features were used along with location feature such as cell shape, cell size, fluorophore expression level etc. In *Long et al., 2008*, *C. elegans* images were annotated by comparing each cell's location in image to cell locations in multiple template images to generate initial matches. Generated matches were then pruned by checking relative position consistency such as anterior-posterior relationship among cells. Thus, although pairwise positions were used, they were not systematically optimized to predict cell labels and were only used for enforcing consistency in post-registration matching. In *Scholz, 2018*, cell identities were determined by registering image data to the 2D atlas (*Altun and Hall, 2009*). Registering 3D data to 2D atlas makes it difficult to disambiguate the identities along LR axis since the LR neurons are not exactly symmetrical along that axis. Similarly, registration-based cell annotation was proposed in *Toyoshima, 2019*; however, in this case, the authors generated their own atlases by registering several partial atlases with subset of cells labeled in each atlas. Registration-based cell identification was proposed in *Yemini et al., 2021* as well. Here, additional color information was integrated as a feature for registration along with spatial location of cells.

## S1.9 Registration methods do not consider intrinsic similarity features such as relative positional relationships

One of the major reasons of higher cell annotation accuracy achieved by our framework (*Figure 2C, D,E*, *Figure 3B*, *Figure 4B*, *Figure 6B*, *Figure 2—figure supplement 5*) is that our model systematically includes and optimizes pairwise positional relationships between cells to determine cell labels. In comparison, registration-based methods that predict cell identities by registering image stack to the atlas maximize the extrinsic similarity between images and atlas (*Bronstein et al., 2007*). Due to inherent biological variability, spatial distribution of cells in worms differ significantly (*Figure 2—figure supplement 6A,B*) from the positions of cells in atlas while the relative positional relationships are still preserved (*Figure 2—figure supplement 6C,D*). Below, we provide a mathematical argument for why registration-based methods do not include pairwise positional relationships. Further, we show that how registration information can be included in our model.

Following the description in S2.1, the objective function to be optimized in registration methods (*Myronenko and Song, 2009*; *Panaganti and Aravind, 2015*; *Ge et al., 2014*; *Ma, 2015*; *Chui and Rangarajan, 2003*) is similar to

$$E \propto \sum_{ik} -c_{ik} ||x_i - \mathcal{T}(y_k)||^2_{\Sigma_k^{-1}} + R(\mathcal{T}) \tag{32}$$

Here, $x_i \in \mathbb{R}^3$ is the coordinate of $i^{th}$ cell in the image, $\mathcal{T}(y_k) \in \mathbb{R}^3$ is the registered coordinate of $k^{th}$ cell with label $k$ in the atlas, $c_{ik}$ is the posterior probability that $i^{th}$ cell in image matches to the $k^{th}$ in cell in atlas, and $\mathcal{R}(\mathcal{T})$ is the regularization term on the transformation applied to cells to ensure smoothness of transformations for non-rigid registration. If only the first term in the energy function (*Equation 4*) is kept in our model then energy function can be written as

$$\mathrm{E}(\mathrm{y}, \mathbf{x}) = \left( \sum_{i=1}^{V} \sum_a \lambda_a f_a(k; i, \boldsymbol{x}) \right) \tag{33}$$

This model considers only unary (cell specific) features for predicting cell names. Further, if we consider only one unary feature function in the model based on spatial locations of cells in atlas defined as below

$$f_a(k; i, \boldsymbol{x}) = ||x_i - \mathcal{T}(y_k)||^2_{\Sigma_k^{-1}} \tag{34}$$

and substituted back in (*Equation 32*) then the energy function is equivalent to the objective function in registration algorithms.

$$\mathrm{E}(\mathrm{y}, \mathbf{x}) = \left( \sum_{i=1}^{V} \sum_a \lambda_a ||x_i - \mathcal{T}(y_k)||^2_{\Sigma_k^{-1}} \right) \tag{35}$$

Thus, objective function in registration algorithms can be specified using only unary features in

CRF_ID framework. This shows that registration methods do not account for relative positional relationship features. Further, it highlights that CRF_ID framework can easily integrate registration information, as unary feature term, with relative positional relationship features for predicting cell identities.

## S2: Description of different methods that were compared

Below, we provide a brief description of the different methods that we compared. Each model uses a different and specific combination of information for predicting cell labels thus helps in dissecting what information is most useful in predicting cell biological names. Some of these methods consist of methods proposed previously for cell annotation in *C. elegans*. Note, for fair comparison, static 3D OpenWorm (*Szigeti et al., 2014*) atlas or 2D atlas available on wormatlas.org was used as reference for all methods: absolute positions of cells were used from atlases for registration methods, positional relationships derived from the same atlases were used for CRF_ID framework.

### S2.1 Registration

In this case, detected neurons in image stacks or neurons in synthetic data were registered to the atlas using a widely used non-rigid registration algorithm (*Myronenko and Song, 2009*). For synthetic data, the process was repeated ~200 times with new synthetic data generated each time. Here, we provide a brief description of the registration algorithm method. Two point-sets are registered by iteratively applying smooth deformation to one of the point-sets. Let two point-sets be $\mathbf{X} = [\mathbf{x}_1, \mathbf{x}_2, ..., \mathbf{x}_m] \in \mathbb{R}^{3 \times m}$ and $\mathbf{Y} = [\mathbf{y}_1, \mathbf{y}_2, ..., \mathbf{y}_n] \in \mathbb{R}^{3 \times n}$ consisting of $m$ and $n$ points respectively. Here, $\mathbf{x}_i$ and $\mathbf{y}_i$ are coordinates of point-sets. In our case, 3D coordinates of nuclei detected in the image form point set $\mathbf{X}$ and cell positions in the atlas form the point-set $\mathbf{Y}$. In CPD, each point in point-set $\mathbf{X}$ is considered to be a random sample drawn from a mixture of gaussian distributions. Further the point set $\mathbf{Y}$ specifies the centroids of gaussian components of this distribution. Thus, the probability of observing a point $\mathbf{x}_i$ is given by

$$P(\mathbf{x}_i) = \frac{\omega}{m} + \frac{(1-\omega)}{n} \sum_{k=1}^{n} \mathcal{N}(\mathbf{x}_i; \mathbf{y}_k, \Sigma_k) \tag{36}$$

Here, $\omega$ is outlier ratio. A smooth transformation $\mathcal{T}(\mathbf{y}_i, \mathbf{W}): \mathbb{R}^3 \to \mathbb{R}^3$ is applied to points in point-set $\mathbf{Y}$ given by $\mathcal{T}(\mathbf{y}_i, \mathbf{W}) = \mathbf{y}_i + \sum_{k=1}^{n} \mathbf{G}(i,k)\mathbf{W}_k$ so as to maximize the likelihood of point-set $\mathbf{X}$. Here $\mathbf{G} \in R^{n \times n}$ is gaussian kernel matrix defined as $\mathbf{G}(i,k) = exp\left(-\frac{(\mathbf{y}_i - \mathbf{y}_k)^2}{\beta^2}\right)$ and $\mathbf{W}_k$ is the kth column of parameters matrix $\mathbf{W} \in \mathbb{R}^{3 \times n}$. Thus, the transformation $\mathcal{T}(\mathbf{y}_i, \mathbf{W})$ lies in a Reproducible Kernel Hilbert Space (RKHS) with gaussian kernel. The aim of parameterizing transformation in this way is to ensure the smoothness of transformation.

Parameters of transformation matrix $\mathbf{W}$ are estimated by maximizing the joint likelihood of data $\mathbf{X}$ and $n$ latent variables corresponding to mixture components. This is done using Expectation-Maximization algorithm. E-step is equivalent to calculating the posterior probability that point $\mathbf{x}_i$ is generated from component $k$, keeping the current parameters $\mathbf{W}, \Sigma_k$ fixed.

$$P\left(\frac{k}{\mathbf{x}_i}\right) = \alpha_{ik} = \frac{\frac{1-\omega}{n} exp\left(-1/2||\mathbf{x}_i - \mathcal{T}(\mathbf{y}_k, \mathbf{W})||_{\Sigma_k^{-1}}^2\right)}{\frac{\omega}{m} + \frac{(1-\omega)}{n} \sum_{k=1}^{n} exp\left(-\frac{1}{2}||\mathbf{x}_i - \mathcal{T}(\mathbf{y}_k, \mathbf{W})||_{\Sigma_k^{-1}}^2\right)} \tag{37}$$

Here, $||\mathbf{x}_i - \mathbf{y}_k||_{\Sigma_k^{-1}}^2 = (\mathbf{x}_i - \mathbf{y}_k)^T \Sigma_k^{-1}(\mathbf{x}_i - \mathbf{y}_k)$. In M-step, component parameters are determined by maximizing the expected value of complete data log-likelihood $\mathcal{L}$ (objective function). Additional regularization term is added to the objective function to minimize norm of $\mathcal{T}$ in RKHS for controlling the complexity (smoothness) of transformation.

$$\mathcal{L} = -\sum_{i,k}^{m,n} \alpha_{ik}||\mathbf{x}_i - \mathcal{T}(\mathbf{y}_k, \mathbf{W})||_{\Sigma_k^{-1}}^2 - \frac{\lambda}{2} tr(\mathbf{W}\mathbf{G}\mathbf{W}^\mathbf{T}) \tag{38}$$

Prior to registration, cells in image stack and in atlas were aligned by defining the position coordinates of cells in AP, LR, and DV coordinate system (as described in S1.3) to improve the registration accuracy. Cell identities were predicted based on the correspondences generated by the registration algorithm. Note that this case does not consider co-dependent features among cells (as discussed in S1.10) as registration algorithms utilize only absolution positions of cells.

Additionally, we modified the registration method described above to account for missing cells in images, in a manner similar to the CRF_ID framework and for fair comparison with the CRF_ID annotation method. Identities were predicted iteratively with uniformly and randomly selected cells across head-ganglion considered missing thus the labels of those cells were removed from the atlas list. Cells in images were registered to the remaining cells in atlas. This process was repeated ~1000 times to sample multiple combinations of missing cells and identities were predicted by pooling results across each iteration. We did see an improvement in prediction accuracy with this modification compared to the base that does not account for missing cells. Hence all comparisons were performed with this modification and all results shown are for the modified method.

For registration based matching we used a popular registration method (*Myronenko and Song, 2010*). The parameter settings of the method are below.

| Parameter | Value | Description |
|---|---|---|
| opt.method | 'nonrigid' | Non-rigid or rigid registration |
| opt.beta | 1 | The width of gaussian kernel (smoothness) |
| opt.lambda | 3 | Regularization weight |
| opt.viz | 0 | Don't show any iteration |
| opt.outliers | 0.3 | Noise weight |
| opt.fgt | 0 | Do not use FGT (fast gaussian transform) |
| opt.normalize | 1 | Normalize to unit variance and zero mean before registering |
| opt.corresp | 1 | Compute correspondence vector at the end of registration |
| opt.max_it | 100 | Max number of iterations |
| opt.tol | 1e-10 | tolerance |

## S2.2 Relative position (Rel. Pos)

In this case, full CRF-based annotation framework as described in Appendix 1–Extended methods S1 was used. We considered only co-dependent features (i.e. pairwise relative positional features as described in S1.2) between all cells. Optimal labels of cells were predicted using the optimization procedure described in S1.4. Note that the information used by model in this case is different than the information used in Registration method as no information about absolute positions of cells is used. Thus, comparing prediction accuracy across these cases helps in verifying that co-dependent features are more useful in predicting neuron identities.

## S2.3 Registration + Rel position

In this case, we used both cell-specific features (i.e. unary features) and co-dependent features (i.e. pairwise features) to predict cell labels. Pairwise feature terms were the same as described in S1.2. Unary feature term was modified in this case as

$$f_{reg}(k;i,\boldsymbol{x}) = \exp\left(-\frac{||\mathbf{x}_i - \mathcal{T}(\mathbf{y}_k)||_2}{\sigma_{reg}^2}\right) \tag{39}$$

Here, $x_i \in \mathbb{R}^3$ and $y_k \in \mathbb{R}^3$ are the coordinates of cell $i$ in image stack and cell $k$ in the atlas, respectively (in AP-LR-DV coordinate system). Thus, cell $i$ in image has a higher potential to take label $l_k$ of cell $k$ in atlas if the distance between registered cell $i$ in image and cell $k$ in atlas is small. Optimal transformation $\mathcal{T}(\mathbf{y}_k)$ was inferred using the non-rigid registration method as described in S2.1. Here, parameter $\sigma_{reg}$ controls the weight between registration term and CRF-based matching term. For example if $\sigma_{reg}$ is small then the registered cell $i$ in image strongly prefers matching to the

nearest cell in atlas. Thus, the relative position features have little influence on altering the cell label based on relative position consistency criteria. In contrast if $\sigma_{reg}$ is big then the registered cell $i$ has uniform preference of matching to any cell atlas. This allows more flexibility to CRF_ID method to pick optimal labels enforcing the consistency criteria. We set $\sigma_{reg} = 1$ to give equal weightage to the registration term and other features. Optimal labels of cells were predicted using the optimization procedure described in S1.4.

## S2.4 Color

In this case, only color information was used to identify the names of cells without using position information of cells at all. This helps in gauging the prediction accuracy that can be attained by using color information. Further, comparing this case with the cases that combine color information with position information (described below) helps in gauging the contribution of cell position information alone in predicting cell names. We describe below, both the baseline (naive) method for building color atlas and also building ensemble of color atlases with color distributions in training images aligned to test image.

### Baseline color atlas

In this case, color atlas was built by aggregating raw RGB values of cells across images used to build atlas leaving the test image out. Feature function in this case was defined as the kernelized Mahalanobis distance between the colors of cells in image stack and colors of cells in atlas. Note that this is a unary feature (as it is cell specific)

$$f_{col}(k; i, \boldsymbol{x}) = \exp\left(-(\mathbf{C}_i - \mathbf{C}_k)^T \Sigma_k^{-1} (\mathbf{C}_i - \mathbf{C}_k)\right) \tag{40}$$

Here, $\mathbf{C}_i \in \mathbb{R}^3$ is the mean RGB value of the $i^{th}$ cell in test image, $\mathbf{C}_k \in \mathbb{R}^3$ is the mean RGB value of cell with label $k$ in atlas and $\Sigma_k$ is the covariance matrix of the RGB values of cell with label $k$ in atlas. Let $\boldsymbol{r}_k = [r_1, \ldots, r_N] \in \mathbb{R}^{3 \times N}$ be the observed RGB values of cells with label $k$ in $N$ training images used to build the color atlas, then $\mathbf{C}_k = 1/N(\boldsymbol{r}_k \mathbf{1})$ and $\Sigma_k = (\boldsymbol{r}_k - \mathbf{C}_k \mathbf{1}^T)(\boldsymbol{r}_k - \mathbf{C}_k \mathbf{1}^T)^T$ where $\mathbf{1} \in \mathbb{R}^{N \times 1}$ is a vector of ones. Thus, this feature specifies that cell $i$ in image has a high potential of taking label $k$ in atlas if the Mahalonobis distance between the colors of cell $i$ and cell $k$ is small.

### Ensemble atlas with aligned color distributions

We found that when baseline color data-driven atlas was combined with positional relationship features data-driven atlas to annotate cells, the performance increased marginally over the case when only positional relationship features were used for annotation (*Figure 6—figure supplement 1B*). This implies that color information did not contribute to annotation task. We reasoned that this is because distributions of color values vary a lot across images thus color distributions in training data did not reflect the color distribution in test data. This may be due to inherent difference in fluorophore expression across animals, differences in imaging settings across sessions (exposure time, laser power) etc. The problem of different feature distributions in test data compared to training data is solved by domain transfer techniques in machine learning. Here we develop a simple two-step domain transfer method.

First, we align color distributions in training images used to build the atlas to the color distribution in test image using several methods –

1.
    Normalization of each color channel in all images so that pixel values lie between 0 and 1. Let $I_R(x)$ denote raw intensity value of red color channel (mNeptune channel) at location $x$ in image. Then the normalized values are calculated as

$$I_R^{norm}(x) = \frac{I_R(x) - \min(I_R(x))}{\max(I_R(x)) - \min(I_R(x))} \tag{41}$$

    Similar normalization is performed for CyOFP and BFP channels.

2. Histogram matching of each color channel in training image to the corresponding channel in test image. Histogram matching transforms the color distribution in an image so that the color histogram matches the color histogram of reference image. This was implemented using 'imhistmatchn' function in MATLAB.

3. Color invariant transformation of training set images and test image, and subsequent histogram matching of training color invariant images to the test color invariant image. Color invariant transformation transforms the color space of image to remove dependency on lighting geometry and illumination (*Finlayson et al., 1998*). The transformation is performed as follows. Let $C_i = [r_x, g_x, b_x] \in \mathbb{R}^{p \times 3}$ be the matrix that stores RGB values of all voxels in $i^{th}$ image in the training set. Here $p$ is the number of voxels in the image. Then the color invariant image is obtained by sequentially and repeatedly normalizing the rows and columns of $C_i$ for a fixed number of iterations.

Note that aligning color distribution of training image to test image does not require cell identity information at all and is performed using all RGB pixel values in the image. After aligning the color distributions, the color atlas is built similar to the baseline color atlas that is by aggregating the RGB values of cells across images. However, in comparison to the baseline case, now RGB values come from the aligned distributions using one of the methods mentioned above.

Second, an ensemble of data-driven color atlases is used for predicting cell identities that is two data-driven color atlases are used with different color alignment techniques used in each atlas. Feature function in this case was defined as

$$f_{col}(k;i,\boldsymbol{x}) = \sum_l \lambda_l \exp\left(-\left(\mathbf{C}_i - \mathbf{C}_{k,l}\right)^T \Sigma_{k,l}^{-1} \left(\mathbf{C}_i - \mathbf{C}_{k,l}\right)\right) \tag{42}$$

Here, $\mathbf{C}_i \in \mathbb{R}^3$ is the mean RGB value of the $i^{th}$ cell in test image, $\mathbf{C}_{k,l} \in \mathbb{R}^3$ is the mean RGB value of cell with label $k$ in $l^{th}$ atlas, $\Sigma_{k,l}$ is the covariance matrix of the RGB values of cell with label $k$ in $l^{th}$ atlas, and $\lambda_l$ is the mixing weight of each atlas, $\sum_l \lambda_l = 1$. Note that each atlas is built with using RGB values from all training images.

In practice we used two atlases in the ensemble atlas. The first atlas used method 2 that is histogram matching of raw RGB distribution in training images to the test image. The second atlas used method 3 that is color invariant transformation was applied to all images (including test image) and subsequently color histogram of training images was matched to the test image. Mixing weights of 0.2 and 0.8 were selected by cross-validation.

Note that in both cases of color atlas, we do not consider any co-dependent features among cell, thus predicting cell names in this case is equivalent to minimizing the following energy function

$$y = \arg\min_{y \in \mathcal{Y}} \sum_{ik} -f_{col}(k;i,\mathbf{x}) \tag{43}$$

This is equivalent to maximum weight bipartite graph match problem and thus we used Hungarian algorithm to find the optimal solution (*Kuhn, 1955*).

## S2.5 Color + registration

In this case, we used both color information and position information of cells to predict cell identities. The features used in this case were $f_{col}(k;i,\boldsymbol{x})$ and $f_{reg}(k;i,\boldsymbol{x})$ as described in S2.1 and S2.4. Note that in this case too, we do not use co-dependent cell features. Cell labels were predicted by minimizing the following energy function

$$y = \arg\min_{y \in \mathcal{Y}} \sum_{ik} -f_{col}(k;i,\mathbf{x}) - f_{reg}(k;i,\mathbf{x}) \tag{44}$$

Here again, we used Hungarian algorithm (*Kuhn, 1955*) to find the optimal solution. By comparing prediction accuracy in this case to the Color only case (S2.4), we can gauge the contribution of position information of cells on prediction accuracy. Here again, we accounted for missing cells in images by predicting the identities iteratively (similar to S2.1). In this case too, we observed improvement in prediction accuracy by accounting for missing cells hence all the comparisons are shown for modified case.

Below, we briefly show how the objective function in (*Equation 44*) naturally arises in registration algorithms that combine multiple features such as color in spatial registration method (*Danelljan et al., 2016*). First, as defined in registration methods (*Myronenko and Song, 2009*; *Jian and Vemuri, 2005*) that use only spatial location feature $\mathbf{x}_i$ of each cell, the probability of observing each cell is (as described in S2.1)

$$P(\mathbf{x}_i) = \sum_{k=1}^{n} P\left(\frac{\mathbf{x}_i}{z_k}\right) P(z_k) = \sum_{k=1}^{n} \pi_k \mathcal{N}(\mathbf{x}_i; \mu_k, \Sigma_k) \tag{45}$$

Here, $z_k$ denotes a spatial mixture component with mean $\mu_k$ and covariance $\Sigma_k^s$. If color is also available then to maximize complete data log-likelihood using EM method, we need to define the joint probability of observing both spatial location $\mathbf{x}_i$ and color feature $y_i$ of each cell, $p(\mathbf{x}_i, y_i)$. Next, we will discuss two possible cases as discussed in *Danelljan et al., 2016* that differ in the way the joint-probability $p(\mathbf{x}_i, y_i)$ is defined.

First case, if the color information is considered to be independent of spatial information then the joint probability factorizes as

$$P(\mathbf{x}_i, y_i, z_i = k, c_i = l) = P\left(\frac{x_i}{z_i = k}\right) P\left(\frac{y_i}{c_i = l}\right) P(z_i = k) P(c_i = l) = \mathcal{N}\left(\mathbf{x}_i; \mu_k, \Sigma_k^s\right) F(y_i; \theta_l) \pi_k \pi_l \tag{46}$$

Here, $z_i$ and $c_i$ denote the spatial and color mixture components (latent variables) from which $\mathbf{x}_i$ and $y_i$ are drawn. Also, $\pi_l, \theta_l$ and $F(c_i, \theta_l)$ denote the mixture weights, parameters and density of observing $y_i$ from a mixture density. The independence assumption is akin to assuming that distribution of colors observed in images in different cells is not dependent on spatial location of cells. However, this assumption is not valid since in NeuroPAL each cell is color coded thus observed color depends on spatial location of cells.

In second case, dependence of color on spatial location of cell is accounted for. Further, it is modeled, that for each spatial component (cell), color is drawn from a location specific mixture density. Thus, the joint probability factorizes as

$$P(\mathbf{x}_i, y_i, z_i = k, c_i = l) = P\left(\frac{x_i}{z_i = k}\right) P\left(\frac{y_i}{c_i = l, z_i = k}\right) P\left(\frac{c_i}{z_i = k}\right) P(z_i = k) \tag{47}$$

Next, we need to define $P\left(\frac{y_i}{c_i, z_i}\right)$ to define complete data likelihood. For NeuroPAL, this is easy since each cell is assigned a unique color (*Yemini et al., 2021*) that is for each spatial component (cell) the color density mixture has only 1 component. Thus, $P\left(\frac{y_i}{c_i, z_i}\right) = P\left(\frac{y_i}{z_i}\right)$ and $P\left(\frac{c_i}{z_i}\right) = 1$.

With updated definitions, the complete likelihood of data is defined as

$$P(X, Y, Z, C) = \prod_i P(\mathbf{x}_i, y_i, z_i, c_i) = \prod_i P\left(\frac{x_i}{z_i}\right) P\left(\frac{y_i}{c_i}\right) P(z_i) = \prod_i \prod_k \pi_k^{z_{ik}} \mathcal{N}\left(\mathbf{x}_i; \mu_k, \Sigma_k^s\right)^{z_{ik}} F(c_i; \theta_k)^{z_{ik}} \tag{48}$$

Now, if $F(c_i, k)$ is defined as $\exp\left(-(y_i - \mathbf{C}_k)^T \Sigma_k^{c-1}(y_i - \mathbf{C}_k)\right)$ (as described in S2.4) and expected complete data log-likelihood is maximized using the EM method then the M-step is equivalent to maximizing

$$\mathcal{L} = -\sum_{i,k} \alpha_{ik} \left( \|\mathbf{x}_i - \mathcal{T}(\boldsymbol{\mu}_k, \mathbf{W})\|^2_{\Sigma_k^{s-1}} + (y_i - \mathbf{C}_k)^T \Sigma_k^{c-1}(y_i - \mathbf{C}_k) \right) - \frac{\lambda}{2} tr(\mathbf{W}\mathbf{G}\mathbf{W}^T) \tag{49}$$

Thus, it can be seen that the first term in *Equation 49* is equivalent to *Equation 44*.

## S2.6 Color + Rel. position

In this case we combined color information of cells with relative position information of cells. Note that in contrast to S2.5, in this case we use co-dependent features (that is pairwise relative position features) in combination with color information, whereas in S2.5, the feature was dependent on cell-specific information (absolute position of cells). The unary feature in this case is same as the Color only method (S2.4).

$$f_{col}(k;i,\boldsymbol{x}) = \exp\big(-(\mathbf{C}_i - \mathbf{C}_k)^T \Sigma_k^{-1}(\mathbf{C}_i - \mathbf{C}_k)\big) \tag{50}$$

Pairwise features in this case were the same as described in S1.2. Optimization of the objective function in this case was performed using Loopy Belief Propagation algorithm as described in S1.4. Comparing prediction accuracy in this case to the Color + Registration (S2.5) method helps in verifying that the higher accuracy achieved by this method in predicting cell identities is due to co-dependent cell features thus highlighting the importance of such features.

## S2.7 Registration + color + Rel. position

In this case, we combined all the cell independent and co-dependent features in one model. Thus, the unary features in this case were $f_{col}(k;i,\boldsymbol{x})$) and $f_{reg}(k;i,\boldsymbol{x})$ as described in S2.4 and S2.1, and co-dependent features were the same as the described in S1.2. Here again, objective function was optimized using Loopy Belief Propagation algorithm as described in S1.4. We simulated this condition to see if prediction accuracy can be boosted by combining the co-dependent position features with registration algorithm. However, in most cases, we saw a decrease in prediction accuracy. This is because the competition between registration term and relative position features term in objective function decreases accuracy.

