## [Decision Letter]

**Acceptance summary:**

Linking individual neurons to their anatomical names remains a vexing problem that limits efforts to determine patterns of gene expression and neuronal activity at the single-neuron level in *C. elegans*. This paper presents a computational method for solving this problem and, as such, has the potential to advance neurobiology in *C. elegans*. The approach could, in principle, be adapted to any nervous system from which a computational atlas could be derived following the pipeline laid out in this manuscript. The generality of the approach and its ability to incorporate additional datasets in composing the computational atlas, may lend itself to performance improvements by crowd-sourcing datasets.

**Decision letter after peer review:**

Thank you for submitting your article "Graphical Model Framework for Automated Annotation of Cell Identities in Dense Cellular Images" for consideration by *eLife*. Your article has been reviewed by Ronald Calabrese as the Senior Editor a Reviewing Editor, and three reviewers. The reviewers have opted to remain anonymous.

The reviewers have discussed the reviews with one another and the Reviewing Editor has drafted this decision to help you prepare a revised submission.

Summary:

Whole brain imaging in *C. elegans* allows monitoring the activities of a large fraction of worm neurons during behaviors. However, the identity of each cell in the nerve ring remains largely unknown, prohibiting further analysis that integrates structural and functional information of the worm circuit. This problem has been partially amended by manual annotation from experts. However, this is a time-consuming and error-prone solution when one is dealing with a large dataset. Thus, the reviewers were enthusiastic about this manuscript because the authors take one important step towards automated annotation of neuronal identities. Methods in this work are innovative and attractive. By minimizing energy functions imposed by absolute coordinate differences (unary potential) between images and atlases, as well as differences of the pairwise relationship between neurons, the authors show that their CRF_ID model outperforms SOTA registration model in accuracy, speed, and robustness.

Essential revisions:

There were several concerns that necessitate serious revision and clarification. Ultimate acceptance by *eLife* will depend upon how the concerns can be addressed and whether the methodology will be practically useful to others in the field after further clarification. The original reviews are appended, and all concerns should be addressed. Chief among the concerns that emerged on the reviewer consultation were:

1) The authors were not clear about whether any data were held out in developing the data-driven atlas and how this choice will impact the ultimate utility of the method. Moreover, they were not clear about how this atlas was built – the motivation for building it.

2) The authors must be clear when synthetic data is being used and which strain is being used for the biological data.

3) The reviewers found the text inadequate and essentially reviewed the Appendix. For ultimate publication the authors must make the methodology clear to general readers in the main text.

4) The prediction accuracies of the model vary from case to case and was quite low in the scenario of missing neurons. The authors must state clearly, under the worst scenario such as whole brain imaging without colors, the prediction accuracy of their methods. The prediction accuracies of the CRF model, contingent upon different datasets and different noise levels, should be in a self-contained main figure, not in the supplements.

5) A key element of these reagents/strains and the annotation is that, strictly speaking, it annotates nuclei (not neurons). The authors need to explicitly tackle the question of whether the method would work with cytoplasmic markers and to properly annotate the strains they did create as using nuclear localization signals.

6) Important elements of the metrics presented to evaluate the annotation method are poorly described in the manuscript, and it is unclear from the manuscript or from the materials in the code repo the form of the results. By this we mean not the file format, but rather does the algorithm return a single assignment or a ranked list for each cell or something else?

7) Concerning the spatial location of the neurons used in the ground truth datasets. From inspecting the authors' GitHub repo, it seems that the neurons in the hand-annotated ground-truth Neuropal datasets are not evenly sampled throughout the head. Rather they are enriched in easy-to-label locations, like the anterior of the head, with less coverage in the more difficult locations (e.g. few neurons are labeled in the lateral ganglia or retrovisicular ganglia). The authors should explicitly recognize this bias in the hand-annotated ground-truth dataset.

Reviewer #1:

The application of conditional random fields to neuron registration in *C. elegans* represents a significant advance that will impact the field. I have major concerns about the interpretation of the method's performance: some technical, for example test sets were seemingly not held out from the data-driven atlas; and others more related to confusing or potentially misleading presentation. It was often unclear which dataset, method, reagent or performance metric (for example Top 1 vs Top 5) was being discussed. I also have concerns about giving appropriate credit for work and reagents from other groups.

1) Test sets used to assess performance do not seem to have been held out from the data-driven atlas, which raises doubts about the interpretation of Figure 2A and Figure 6E and C. This could artificially inflate performance compared to tests with a held-out animal, which would reduce the utility of the method. This could also explain the dramatic difference in performance observed fr om synthetic data with experimentally derived noise (35% accuracy) Figure 2—figure supplement 4A, and real experimental data matched to the data-driven atlas (>75% accuracy Figure 2A).

2) Lack of clarity could falsely give readers the impression of higher performance:

a) It is often ambiguous as to whether Top 1 or Top 5 performance is being plotted. Example: in Figure 3B the caption makes no mention, so one would assume Top 1 accuracy; but subsection “Cell annotation in gene expression pattern analysis” suggests Top 5. This is a crucial distinction and needs clarity in every subpanel.

b) It is also unclear when synthetic data is being used. Example: subsection “Identity assignment using intrinsic features in CRF_ID outperforms other methods” states that the algorithm "maintains high accuracy even when up to 75 % of cells in atlas were missing for data" but from the figures I gather this is synthetic data with unnaturally low position noise, so this could give the reader the wrong impression.

c) Similarly, it is often unclear from the figures which strain is being used. For example the strain in Figure 4A-C is not labeled, but it appears to be the same as Figure 3A,B. But if that's the case its unclear why performance would be so much higher in Figure 4B than Figure 3B. Please label which strain is used in which figure subpanel throughout the manuscript, or provide a table.

d) Related: Figure 4A prominently displays "calcium imaging," when this is in fact a GFP strain. This misleads the reader

3) The work critically relies on reagents from other groups but could do a better job acknowledging those contributions:

a) The Neuropal strain is crucial to the manuscript and provides the majority of the ground-truth data starting with the Results and Figure 2, but Neuropal is first mentioned only towards the very end of the Results section and even then only in the context of Figure 6.

b) The strain list should reference publications for strains and alleles used.

c) Subsection “Cell annotation in gene expression pattern analysis” misleadingly states that "a reporter strain was crossed with pan neuronal red fluorescent protein" but this cross was not performed by the authors, it was performed by another group, published and is publicly available on CGC. This should be fixed.

4) More details are needed about the ground truth datasets, including the number of neurons hand-annotated in each of the 9 individuals and their spatial location. (Manual annotation of Neuropal is easiest at the periphery and hardest in center of the ganglia, and so it would be important to know coverage).

Reviewer #2:

Whole brain imaging in *C. elegans* allows us to monitor the activities of a large fraction of worm neurons during behaviors. However, the identity of each cell in the nerve ring remains largely unknown, prohibiting further analysis that integrates structural and functional information of the worm circuit. This problem has been partially amended by manual annotation from experts. However, this is a time-consuming and error-prone solution when one is dealing with a large dataset. We are enthusiastic about this manuscript because the authors take one important step towards automated annotation of neuronal identities. Methods in this work are innovative and attractive. By minimizing energy functions imposed by absolute coordinate differences (unary potential) between images and atlases, as well as differences of the pairwise relationship between neurons, the authors show that their CRF_ID model outperforms SOTA registration model in accuracy, speed, and robustness.

We would like to raise several main issues of this manuscript.

1) Writing. The main text of the manuscript is brief, dense, technical, and very difficult to understand. I skipped the main text and went directly to the Appendix. This arrangement is not wise for a general audience, especially biologists. In future revisions, please embed the figures in the text, which would save us a lot of time. Here are some specific suggestions:

a) The CRF framework can be better explained by building upon physical intuition. The mathematical notations sometimes only cause confusion. For example, is it necessary to introduce clique? I thought that every neuron in their graph model is adjacent to every other neuron. If this is not the case, please describe how the graph model is constructed. You find the note on maximum entropy model is out of the blue (subsection “Structured prediction framework”). If you want to make a deep connection with statistical physics, you better explain it well in the text!!

b) The motivation to build and how to build the data-driven atlases are dismissive in the main text. It is briefly mentioned in subsection “Identity assignment using intrinsic features in CRF_ID outperforms other methods”. But these sentences are extremely difficult to understand without reading the Appendix. The Appendix -- extended methods S1.7, however, needs to be revised and better explained. For example, the equations are not quite consistent, such as in Equations 6 and 25, and the notations are quite confusing. Linking some examples like Figure 2B to S1.7 will help understand the concept.

c) The prediction accuracies of the model vary from case to case. It was quite low in the scenario of missing neurons. The authors need to state clearly in their paper, under the worst scenario such as whole brain imaging without colors, the prediction accuracy of their methods. This information is now buried somewhere that is hard to find. The prediction accuracies of the CRF model, contingent upon different datasets and different noise levels, would better be a self-contained main figure, not in the supplements. To follow the logic flow of the text, the figure panels that compared CRF with other registration methods would better be introduced later.

2) Notations. As mentioned above, the symbol consistency is a problem. In the formulation of the model, λ's, trainable parameters in original CRF, have been abused. For the λ appearing in Equations 2/3/4/6/7/8/9/10/32/34/37/46, the author might consider making a clear statement that which of them are trainable and which are hyperparameters. If they are hyperparameters, the setting and tuning strategies need to be disclosed. Also, the keywords "identification" and "annotation" should be considered unified.

3) Loop Belief Algorithms, speed, and stability. The authors should provide details on the computational complexity and the stability of their algorithms and compare it to other registration methods. In their current graph model, does each trial converge and do they all converge to the same solution starting from different initial conditions? If not, how do the authors deal with the discrepancies? Likewise, the authors may want to comment on the memory usage for constructing the data-driven atlases to see if it can be another innovation point comparing to the registration methods.

4) Other missing information and explanations:

a) The authors should post the exact name and its settings of the registration models used for comparison.

b) In Appendix S1.2.1 Unary potentials – Positions along AP axis, as x-y is a 2D plane, we do not see why AP's is always better than LR's and DV's.

c) Following the explanation of generating synthetic data (subsection “Generating synthetic data for framework tuning and comparison 464 against other methods”) is a bit hard. The reason for changing the low bound to zero might need further illustrations. Could you please comment on why the prediction accuracy is much lower in synthetic data (Figure 2A)?

d) For subsection “Whole-brain data analysis”, the author might consider linking the paragraph to Figure 5, making 0.1Hz a reasonable choice.

e) For subsection “Whole-brain data analysis”, "[0.05, 0.05]" seems to be a mistake.

f) Subscripts in Equation 13 are wrong.

g) Providing more annotations and explanations on the meaning of the colors on some worm images (e.g., Figure 4D) would be helpful.

5) Codes on GitHub may need to be slimmed down.

Reviewer #3:

Linking individual neurons to their anatomical names is a vexing problem that limits efforts to determine patterns of gene expression and neuronal activity at the single-neuron level. Given the landmark Mind of the Worm paper (White et al., 1986) and subsequent analyses of its connectome, many imagine that annotating individual neurons in *C. elegans* is a solved problem. Unfortunately, it is not. This paper presents a computational method for solving this problem and, as such, has the potential to advance neurobiology in *C. elegans*. The approach could, in principle, be adapted to any nervous system from which a computational atlas could be derived following the pipeline laid out in this manuscript. There is much to recommend this study and many opportunities for improvement regarding the presentation of the computational method, its limitations and expectations for application, and for considerations of generalization to other nervous systems.

1) Explanation of the computational method

The computational strategy is difficult to follow in the main text, unless one reviews the appendices. We suggest elaborating the process using more generic terms in the main text. The following specific elements need clarification or additional support

- The mixed approach of considering intrinsic and extrinsic similarities between image and atlas is described in a manner that assumes readers are already familiar with this computational strategy. Given the breadth of the *eLife* readership, the authors would increase the accessibility and impact of the paper by providing readers with examples of what features are intrinsic and which are extrinsic in this case.

- The difference between this approach and previous strategies based on image registration is communicated clearly in the Appendix, but not in the main text.

- Much is made of the computational efficiency of the CRF_ID approach, but little is provided in the way of data or analysis supporting this claim. The authors could, for example, compare the computational energy/time needed for this approach compared to prior image matching approaches. They could discuss the nature of this computational framework compared to others with respect to computational efficiency.

Generality of the method

To better support the authors claim that this method is easily adaptable for any nervous system, please apply the framework to estimate the optimal number (and/or density) of landmarks as a fraction of the total number of nuclei or density in 3D. This would assist others in applying the framework to other simple nervous systems (e.g. *Aplysia*, leech, hydra). Additionally, it is not clear if the image stacks used to derive data-driven atlases are distinct from the data analyzed for prediction or whether the same data is used iteratively to build atlases and improve prediction accuracy.

Application of the method to *C. elegans*

Specifying the AP/DV/LR axes from raw data is not completely automatic but rather appears to require some user input based on prior knowledge. This is not stated clearly in the text. In addition, it is also not clear how the algorithm handles data from partially twisted worms (non-rigid rotation around the AP-axis). This problem is more likely to be encountered if the images capture the whole worm. Is it possible to allow rotation of the DV and LR axes at different points along the AP axis?

Specific suggestions for improvement are collected below.

(1) The method depends on the tidy segmentation of nuclei, but the segmentation strategy is not discussed nor is whether or not nuclear localization of landmarks or unknown markers is essential for accurate annotation.

(2) The authors need to explicitly inform readers that the markers in the NeuroPal strains (OH15495, OH15500) and the GT strains are localized to neuronal nuclei. Additionally, the authors need to address (1) whether or not nuclear localization is required for the annotation framework to succeed, and (2) discuss whether or not a cytoplasmic marker would be sufficient, and (3) provide more information on the segmentation algorithm used and recommended for use with the computational framework discuss.

(3) The authors use the AML5 strain as a proxy for GCaMP-labeled strains (subsection “Cell annotation in multi-cell functional imaging experiments”) to demonstrate the utility of the annotation tool for identifying cells used for calcium imaging. This strain differs from the NeuroPal strain expressing GCaMP is several respects that are not discussed by the authors, leaving readers to wonder about the utility of AML5 as a proxy and to question whether or not the success of cell identification in this strain is truly generalizable. Two main differences exist: (1) In AML5, GFP is expressed in both the nucleus and cytoplasm but in in OH15500 GCaMP6s is tagged with an NLS; (2) The basal fluorescence of GCaMP6s, but not GFP depends on basal calcium concentrations-this fact may introduce additional detection noise/variance into the annotation problem. The authors need to discuss how these differences affect (or don't) the ability to generalize from success with the proxy strain to success with NLS::GCaMP6s expression.

(4) Please clarify the nature of the output from the code that implements the framework. In other words, a reader should learn from the paper what kind of output file they might receive if they attempted to apply the tool to their own images (annotated image with top prediction candidate, data structure with all predictions and respective probabilities, etc).

(5) Terminology – There are a number of general computational terms and specific terms relevant to this problem that are not defined for the reader (listed below). Providing readers with implicit or explicit definitions of these terms will improve clarity.

5(a) "count noises" – used to represent variation in the number of nuclei detected in a given image stack, but not defined at first use. Please provide conditions under which the number of nuclei detected might vary along with defining this term. Additionally, this is a singular concept and should be written as "count noise" not "count noises" (by analogy to "shot noise").

5(b) There several other instances in which "noises" is used where "noise" is meant.

5(c) "intrinsic similarity" – which seems to represent pairwise positions in 3D in the image and the atlas.

5(d) "extrinsic similarity" – not clear.

(6) (Subsection “Identity assignment using intrinsic features in CRF_ID outperforms other methods”) "More importantly, automated annotation is unbiased and several orders of magnitude faster than manual annotation by researchers with no prior experience."

(a) Unbiased – It is a common misconception that computational strategies are unbiased. For this claim to be rigorously accurate, the authors would need to demonstrate that each and every nucleus was equally likely to be identified correctly by the automated annotation algorithm. For instance, it is easy to imagine that some nuclei, such as those closer to a landmark, are more likely to identified correctly and with less uncertainty than those further away. If this were true, then automated annotation has a systematic bias in favor of nuclei closer to landmarks and a systematic bias against nuclei further away from landmarks. And would, therefore be biased – even though no human supervision is involved. The authors should back this claim up with data or clarify that what they mean is that automated annotation is an improvement over (biased and subjective) human annotation.

(b) Faster -The authors are free to speculate about the relative speed of automated vs. manual annotation by an inexperienced researchers in the discussion. By writing this as a result, however, data are needed to back it up.

(7) (Materials and methods) From the description of how GT290 and GT298 were made, it seems that both constructs include 2x NLS fragments, but the genotype description under reagents lacks that annotation and most, if not all other references to this strain neglect to mention the NLS.

Presentation concerns that need attention: There are three instances of undisclosed data re-use among the figures that need to be addressed. (1) The image in Figure 4A is identical to the green channel of the image shown in Figure 3A; (2) Some of the data shown in Figure 2—figure supplement 4A – Part of the data shown here is also shown in Figure 2C; (3) Figure 4C – Please indicate if the same number of cells were removed in each run. Also, it appears that this data also may have been used in Figure 4—figure supplement 1. If so, please disclose data re-use.

[Editors' note: further revisions were suggested prior to acceptance, as described below.]

Thank you for submitting your article "Graphical-Model Framework for Automated Annotation of Cell Identities in Dense Cellular Images" for consideration by *eLife*. Your article has been reviewed by Ronald Calabrese as the Senior Editor, a Reviewing Editor, and two reviewers. The reviewers have opted to remain anonymous.

Essential Revisions:

(1) The authors performed additional computational experiments by holding out some of the data used to create atlases – an important and general strategy for computational solutions that depend on real-world data. The authors present a comparison of this approach for reviewers, but not for explicitly for readers. These analyses and plots must be included in the final version main text.

(2) The authors should state clearly the computational speed of their algorithms and compare it with the registration methods. The inference and subsampling procedures (~ 1000 times?) on the data-driven atlas are time-consuming and computationally expensive.

Reviewer #2:

Main Text

(1) Thanks for sending us a much better written manuscript. The authors should consider replotting some of their figures, which can be difficult to read.

(a) The neuron names in Figure 2F and Figure 2—figure supplement 6C are difficult to read. Is there a new way to plot? When I wanted to zoom in to have a better understanding, they caused my Preview crashed multiple times…

(b) The meaning of the error bars throughout the manuscript are not well explained.

(2) The authors should state clearly the computational speed of their algorithms and compare it with the registration methods. The inference and subsampling procedures (~ 1000 times?) on the data-driven atlas are time-consuming and computationally expensive.

(3) Is there a way to make the data-driven atlas publicly available in addition to the raw annotated datasets (n=9)?

Rebuttal

(1) Pipeline (Major Point#6):

I do not quite understand the necessity of single or multi-run, which can be combined into one inference process. Consider the following scenario in a freely behaving animal. At one time, the region to be annotated consists of N neurons. At a different time, a new neuron (now N+1) is squeezed into the region. It is probably always true that the number of neurons in the data is different from that in the atlas.

(2) Color and Robustness and Evaluation Metrics (Essential revisions #1, #4, #6):

It seems that the leverage of color information for annotation is quite limited in the revised manuscript. The procedures for properly normalizing the color distribution are also complicated. I also found that the following sentence was incorrect. "Further, when leave-one-out atlases are used for both positional relationship and color (Author response image 2), the accuracy achieved by top labels is marginally greater than the accuracy achieved by using leave-one-out positional relationship only (Author response image 1)". It was actually significantly lower if I read the figures correctly. This leaves the question how we shall correctly assemble the data-driven atlas efficiently.

Reviewer #3:

The revision is very responsive to the summary and specific reviewers' critiques.

There remain some awkward phrasing and missing details to address. These are noted in order to make the text accessible to a broader audience and to increase the impact of the study both within and beyond the community of *C. elegans* researchers. These include:

(1). The use of the plural "noises" when the authors probably mean either "noise" in the singular e.g. ” Prediction accuracy is bound by position and count 316 noise in data”, replace "amount of various noises" with "amount of noise contributed from various sources". This latter example combines both the concept of the magnitude of noise and the diverse types of noise.

(2) Shorthand terms or jargon. For example, when the text reads "the position of cells AIBL" what is meant is "the position of the cell bodies of AIBL". Similarly, replace "neurons" with "neuronal cell body" or "neuronal somata".

(3) *C. elegans* strains – in response to critiques of the initial submission, the authors have clarified the origin of the transgenic strains used in the study. Thank you! Please also correct the reference to AML70 which is not included in the table of strains. Based on context, it looks like the authors are referring to AML32. Please ensure that this is corrected.

(4) Figure 5, panel E. Please refer to Figure 5—figure supplement 1 panel C noting the cells with non-zero weights in SPC1, SPC2, SPC3

(5) Figure 5 panel H: Indicate the origin of the motion trace shown. It appears to belong to one of the posterior cells shown in Panel F. Please provide a rationale for this choice – since the traces in Panel F certainly indicate that motion signals differ among cells.

---

## [Author Response]

Essential revisions:There were several concerns that necessitate serious revision and clarification. Ultimate acceptance by eLife will depend upon how the concerns can be addressed and whether the methodology will be practically useful to others in the field after further clarification. The original reviews are appended, and all concerns should be addressed. Chief among the concerns that emerged on the reviewer consultation were:1) The authors were not clear about whether any data were held out in developing the data-driven atlas and how this choice will impact the ultimate utility of the method. Moreover, they were not clear about how this atlas was built – the motivation for building it.

We appreciate the reviewers for pointing these out. These points are indeed very important and were an oversight in the initial submission. We have since tested the model with held out data (detailed below) and added more text to emphasize the motivation for building the atlas main text, Appendix – extended methods S1.7.

We present new cell identity prediction results with held out test data. In all new results presented below, we built the atlases with all but one data set (one worm image stack) and used the held out set for testing the accuracy; thus, the atlas each time is a leave-one-out atlas. We then report the average and spread of these predictions. In the text below, we explicitly compare what was reported in the initial manuscript and what is reported in the revision. It is important to note that with the new results, all conclusions stand.

First, for Figure 2A where we build only positional relationship atlases, here we predict identities in experimental whole-brain datasets using data driven atlases. Note in this case, no color information is used for prediction. With leave-one-out-atlases, we see no difference in performance compared to previous results (Author response image 1). This implies that leave-one-out atlases are able to represent positional relationships among cells in test data.

**Author response image 1. respfig1:** Accuracy comparison on experimental datasets when prediction was done using atlas built with all data vs leaveone-out atlases. Prediction was done using positional relationship information only. This figure is update for Figure 2A. Experimental datasets come from NeuroPAL strains.

Second, for Figure 6C, in contrast to Figure 2A, we build both positional relationship atlases and color atlases, *i.e.* we use color information along with positional relationship information to predict identities. We tested accuracy for several kinds of atlases: (1) leave-one-out-positional and allcolor atlas – using only positional relationship leave-one-out atlas and color information comes from all datasets including test data; (2) all-positional and leave-one-out-color atlas – using only color leave-one-out-atlas and positional relationship information comes from all datasets including test data; (3) leave-one-out-positional-and-color atlas – using both positional relationship and color leave-one-out atlases (Author response image 2).

We found that using color leave-one-out atlases decreases accuracy more sharply than using positional relationship leave-one-out atlas. Further, when leave-one-out atlases are used for both positional relationship and color (last bar in Author response image 2), the accuracy achieved by top labels is marginally greater than the accuracy achieved by using leave-one-out positional relationship atlas only (second pair of bars in Author response image 1, and Figure 2A). With the data sets at hand, using color does not contribute much to improving accuracy; this implies that the color variability among animals is far greater than the positional variability.

We see that raw RGB intensity distribution in test image data is different from RGB intensity distribution in atlas. This could be due to several reasons, including inherent differences in NeuroPAL fluorophore expression levels across animals, and differences in image acquisition settings across sessions (exposure times, laser power) etc. As a consequence, raw RGB values that represent colors of cells vary across animals. NeuroPAL manuals also suggest manual contrast and γ adjustment of color channels in images before performing manual annotation of cell identities. Thus, to be able to use color information for automatic annotation we need to first align distribution of features (RGB values) in atlas to distribution of features in test data, also commonly known as domain adaptation problem in machine learning.

In order to increase the utility of the color information, either a much larger dataset would need to be acquired to account for the variability, or that the atlas needs to be built based on images with as close to the experimental conditions as the data to be analyzed (which in our estimates is not a trivial task).We have elaborated on these points in the manuscript.

**Author response image 2. respfig2:** Accuracy achieved on experimental datasets when prediction was done using different combinations of atlases for positional relationships and color. In these cases, leave-one-out atlas was used for either positional relationship, color or both. Experimental datasets come from NeuroPAL strains.

To test whether we can improve the utility of the color information, we explored several simple methods to align color values of images used for building leave-one-out color atlas to color values in test data. Note that color matching is performed using full RGB distributions in image *without* using any cell identity information. The methods we tried included simple normalization (norm) of color channels, color constancy/invariant transformation[1] (colconst), histogram matching (histm), contrast and γ adjustment (imadj), and a combination of these methods (Author response image 3). For each method we found that accuracy was improved for some data sets but not all datasets.

**Author response image 3. respfig3:** Accuracy on experimental datasets when prediction was done using different leave-one-out atlases for color. In each method, a different technique was used to match colors of images used to build atlas to colors of test image. Leave-one-out atlas for both positional relationships and color was used for prediction. Experimental datasets come from NeuroPAL strains.

Finally, we found that using an ensemble of two methods for color atlas gave best accuracy results (Author response image 4), improving top label accuracy by 6% (with an average of 80.5%) compared to the baseline case of color leave-one-out atlas (average of 74.5%). Here, ensemble of two methods means using a combination of two different leave-one-out color atlases with different color alignment manipulations performed in each for predicting cell identities in test data. Weights of 0.2 and 0.8 were assigned to the two atlases in ensemble (determined by cross validation). The methods included in the best ensemble were (1) histogram matching (i.e. histm), and (2) color invariant manipulation, normalization and subsequent histogram matching (i.e. colconst + norm + histm). We have updated the ensemble methods results in Figure 6C, Figure 6—figure supplement 2 and updated the text.

**Author response image 4. respfig4:** Accuracy comparison between base case (when prediction was done using leave-one-out color atlas without any color matching) and ensemble of two leave-one-out color atlases (with different color matching techniques). In both cases, same leave-one-out atlases for positional relationships were used. Experimental datasets come from NeuroPAL strain (n = 9). Results in this figure are updated in Fig. 6C.

Using ensemble of leave-one-out-color atlases also led to improvement in accuracy in non-rigidly rotated animal datasets (related to Figure 6E), improving accuracy of top labels from 39.1% (base case) to 48.8% (ensemble atlases with color matching) (Author response image 5). We have updated these results in Figure 6E, Figure 6—figure supplement 3 and updated the text.

**Author response image 5. respfig5:** Accuracy comparison between base case (when prediction was done using leave-one-out color atlas without any color matching) and ensemble of two leave-one-out color atlases (with different color matching techniques). In both cases, same leave-one-out atlases for positional relationships were used. Experimental data comes from NeuroPAL strains with non-rigidly rotated animals (n = 7). Results in this figure are updated in Fig. 6E.

We also updated results in Figure 6B, where we compare accuracies of several methods on experimental whole-brain datasets. In this figure, for fair comparison across methods OpenWorm atlas was used for positional relationships. For, methods that use color information (i.e. ‘Color’, ‘Reg. + Color’, ‘Color + Rel. Position’, and ‘Reg. + Color + Rel. Position’ methods), data-driven color atlas was built. Now we perform predictions using ensemble of leave-one-out color atlases for methods that use color information. Note that the conclusion remains same – CRF model with positional relationship features along with color outperforms other methods.

We have updated Appendix – extended methods section S2.4 with description of building baseline color atlas, building color atlas after matching color distributions in training data to test data and using ensemble of color atlases for prediction.

Thus, we conclude that in order to be able to utilize color information of cells in NeuroPAL for automatic annotation of cell identities, color consistency across animals needs to be maintained, either experimentally or by post hoc corrections. Experimentally, consistent protocol/imaging settings across sessions should be maintained. Even with consistent protocol, color variation may exist. This can be tackled by (1) collecting large volume of data to capture each cells’ full RGB variations, and (2) using computational domain adaptation techniques. More advancement in image color transfer and domain adaptation techniques will further improve accuracy in future. We have updated these discussions in the text.

[1] Finlayson, Schiele and Crowley. 1998.

To clarify the motivation and methodology of building data-driven atlas, we have now rewritten several sections and included an additional schematic figure –

1) We have text in the Introduction section that talks about the variability of the cell positions in the data which is the main motivation for building such data-driven atlas.

2) We have re-written the “Cell annotation using Structured prediction framework”. Here we describe the motivation and intuition behind building data-driven atlas

3) We have re-written the section “Appendix – extended methods S1.8 Building data-driven atlas”. This section further expands on the intuition and methodology of building data driven atlas for positional relationship features.

4) We have added new details in the section “Appendix – extended methods S2.4” to describe the methodology of building data driven atlases for color features.

5) We have added a new figure as Figure 1—figure supplement 1 that schematically explains how the positional relationship features in the model are modified when data driven atlas is used compared to when static atlas is used.

2) The authors must be clear when synthetic data is being used and which strain is being used for the biological data.

We have now indicated synthetic data or experimental data in each figure subpanel. Strain being used is mentioned in Figure legends. Following figures and figure legends have been changed: Figure 2, Figure 3, Figure 4, Figure 5, Figure 6, Figure 2—figure supplement 1, Figure 2—figure supplement 2, Figure 2—figure supplement 3, Figure 2—figure supplement 4, Figure 2—figure supplement 5, Figure 2—figure supplement 6, Figure 4—figure supplement 1, Figure 5—figure supplement 1, Figure 6—figure supplement 1, Figure 6—figure supplement 2, Figure 6—figure supplement 3.

3) The reviewers found the text inadequate and essentially reviewed the Appendix. For ultimate publication the authors must make the methodology clear to general readers in the main text.

We have rewritten the subsection “Cell annotation formulation using structured prediction framework” in main text, mainly by moving some of the prior text from the appendix and streamlining the description.

1) We describe the methodology to make it more accessible to readers.

2) We describe the motivation and intuition behind building data-driven atlases (subsection “Identity assignment using intrinsic features in CRF_ID outperforms other methods”, Appendix – extended methods S1.7)

3) We have added a new schematic figure as Figure 1—figure supplement 1. This figure clearly explains (1) all features in the CRF model, (2) difference between intrinsic similarity and extrinsic similarity features, and (3) how features are updated with data driven atlas.

We realize that this beginning portion of the text may be math-heavy, but left it as such for the sake of clarity as the reviewers proposed.

4) The prediction accuracies of the model vary from case to case and was quite low in the scenario of missing neurons. The authors must state clearly, under the worst scenario such as whole brain imaging without colors, the prediction accuracy of their methods. The prediction accuracies of the CRF model, contingent upon different datasets and different noise levels, should be in a self-contained main figure, not in the supplements.

We would like to reiterate a few important points that were perhaps obscured in the original manuscript. First, the models we built are data-driven. This means that the accuracy of each model depends on the quality (e.g. signal-to-noise ratio), information content (e.g. availability of labeling of positional information, color information), and quantity of the data. We do not build a single model because the use-case of these models are strongly dependent on the biological questions asked where different reagents are used (e.g. different markers). Second, the accuracy of the models will strongly depend on the information contained in the data that were used to build/train the model. In other words, a model that is built from data of higher quality, more information content, and a larger data set will perform better. Another way to look at this is that models cannot pull information out of thin air (bad data with little information).

In general, missing neurons in the data strongly erode the performance of the models, regardless of the methods used. How exactly missing neurons influence the model performance depends on the information content of the neurons missing. To give an example, neurons that can anchor the AP, DV, and LR axes are highly “valuable”; if they are randomly missing in the simulations, they will have a larger impact than some other neurons. For each scenario that we tested, we ran at least 180-200 random trials to assess the variability.

As we show in the sections “Prediction accuracy is bound by position and count noise” and “Identity assignment using intrinsic features in CRF_ID outperforms other methods”, accuracy of all automatic annotation methods will be degraded by the noise in data. We formalize the concept of types of noise encountered in automatic cell annotation task, i.e. position noise and count noise, and study the effect of these noises on accuracy across a range of noise levels.

Our motivation for testing accuracy across a range of noise levels comes from the fact that the amount of noise that will be encountered practically in experimental data will depend on the use case, and thus affecting the accuracy results. For example, in gene expression pattern analysis, the amount of count noise will be huge since there is no candidate label list available for annotation. In comparison, in multi-cell calcium imaging case, the amount of count noise will be much lower because candidate labels that can be assigned to few cells in image will already be available. Similarly, the effect of position noise on accuracy in whole-brain imaging case (densely packed cells) will be more pronounced than multi-cell calcium imaging case (sparsely labelled and spatially distributed cells). Thus, by systematically varying the noise levels using synthetic data, we show that CRF method outperforms popular registration based methods. The synthetic data illustrate the point about the information content in the data. Again, we emphasize that noise in the data such as missing neurons will affect any method; no one is immune to the problems caused by missing information, and no algorithm can make up information that is not in the data. The important point is whether an algorithm can outperform other algorithms, and get the maximum information from the available data.

Figure 2A is “the worst scenario such as whole brain imaging without colors (of) the prediction accuracy of (the) methods”. New results with leave-one-out atlas are added for the updated Figure 2A.

We have moved the figures and figure supplements closer to the text. We compare the accuracies of methods across different noise levels using synthetic data in Figure 2—figure supplement 5. Since these comparisons are performed using synthetic data, we felt that it should remain as a supplement. Figure 2C-E does contain the major take-away message of the comparisons, i.e. CRF methods performs better than registration based methods in handling both position and count noise. Further it demonstrated superior performance of experimental datasets.

5) A key element of these reagents/strains and the annotation is that, strictly speaking, it annotates nuclei (not neurons). The authors need to explicitly tackle the question of whether the method would work with cytoplasmic markers and to properly annotate the strains they did create as using nuclear localization signals.

We agree with the reviewers that the algorithm is annotating nuclei of neurons currently for whole-brain data (because the datasets are all labeling nuclei). The algorithm should work with cytoplasmic markers, as long as cells can be demarcated as separate entities (e.g. results shown in Figure 3B, Figure 4B, 4C are cytoplasmic markers). For whole-brain imaging, the cytosolic markers will make a bunch of cells look like one big blob, unless perhaps when cell membranes are labeled, in which case, our algorithm should work.

We also note that if the cells cannot be distinguished as separate entities (by eye or by segmentation algorithms), then this will contribute to the count noise that we have characterized in the paper; we have shown that CRF performs better than registration method in handling count noise.

To clarify, our landmark strains contain nuclear localized landmarks for whole-brain imaging application. All nuclei are annotated as shown in Figure 4D.

6) Important elements of the metrics presented to evaluate the annotation method are poorly described in the manuscript, and it is unclear from the manuscript or from the materials in the code repo the form of the results. By this we mean not the file format, but rather does the algorithm return a single assignment or a ranked list for each cell or something else?

We appreciate this comment on clarify. There are two modes of running algorithm: (1) Single run returns top label for each cell; this works when the number of segmented cells/nuclei matches the number of labels available to assign. (2) Accounting for missing neurons returns top candidate for each cell; this requires multiple runs, which we performed in parallel on computing clusters; since neurons missing will dependent on the confidence of knowing which neurons may not show up with the particular reagents (and experimental conditions for that matter), we also report additional top labels (e.g. top 3 and top 5) for users to potentially take advantage of heuristics or prior knowledge that are not hard-coded into the algorithm. We have now described the code output in subsection “Computational workflow for cell identification”. We have also now indicated, in each of the corresponding figure legends, which metric is being shown in the figure.

7) Concerning the spatial location of the neurons used in the ground truth datasets. From inspecting the authors' GitHub repo, it seems that the neurons in the hand-annotated ground-truth Neuropal datasets are not evenly sampled throughout the head. Rather they are enriched in easy-to-label locations, like the anterior of the head, with less coverage in the more difficult locations (e.g. few neurons are labeled in the lateral ganglia or retrovisicular ganglia). The authors should explicitly recognize this bias in the hand-annotated ground-truth dataset.

We agree with this comment. NeuroPAL is at the moment the best available data for neuron identity ground truth since it offers more neurons labeled than other strains available. Since we are not experts in neuroPAL, we tried our best at annotating the ground truth and may be biased in the cells identified. Additional details of number of cells in each of the anterior, middle, and posterior regions of head ganglia annotated in datasets set are provided below. We have added these details in the manuscript as Figure 2—figure supplement 3. As can be seen, we have good coverage in anterior and lateral ganglions. In our experience, annotating cells in the Retrovesicular ganglion (included in posterior region in our analysis) and the Ventral ganglion (included in middle region in our analysis) is the hardest. In animals with rotated orientations we have lower coverage because manually identifying neurons in this case is more difficult. As the reviewers said, the more data available the better the eventual model for prediction.

Reviewer #1:The application of conditional random fields to neuron registration in *C. elegans* represents a significant advance that will impact the field. I have major concerns about the interpretation of the method's performance: some technical, for example test sets were seemingly not held out from the data-driven atlas; and others more related to confusing or potentially misleading presentation. It was often unclear which dataset, method, reagent or performance metric (for example Top 1 vs Top 5) was being discussed. I also have concerns about giving appropriate credit for work and reagents from other groups.1) Test sets used to assess performance do not seem to have been held out from the data-driven atlas, which raises doubts about the interpretation of Figure 2A and Figure 6E and C. This could artificially inflate performance compared to tests with a held-out animal, which would reduce the utility of the method. This could also explain the dramatic difference in performance observed fr om synthetic data with experimentally derived noise (35% accuracy) Figure 2—figure supplement 4A, and real experimental data matched to the data-driven atlas (>75% accuracy Figure 2A).

Please see our explanation above (#1 in the overall response).

2) Lack of clarity could falsely give readers the impression of higher performance:a) It is often ambiguous as to whether Top 1 or Top 5 performance is being plotted. Example: in Figure 3B the caption makes no mention, so one would assume Top 1 accuracy; but subsection “Cell annotation in gene expression pattern analysis” suggests Top 5. This is a crucial distinction and needs clarity in every subpanel.

Please see our explanation above (#6 in the overall response).

b) It is also unclear when synthetic data is being used. Example: subsection “Identity assignment using intrinsic features in CRF_ID outperforms other methods” states that the algorithm "maintains high accuracy even when up to 75 % of cells in atlas were missing for data" but from the figures I gather this is synthetic data with unnaturally low position noise, so this could give the reader the wrong impression.

Please see our explanation above (#2 in the overall response).

The point about the synthetic data accuracy is not to say that the algorithm has such and such accuracy, but rather to compare to existing methods (particularly registration) that for a fixed amount of position noise CRF maintains higher accuracy in handling even high count noise level (75%). Further we show that this is true across all position noise levels (Figure 2—figure supplement 5). We changed the language to say “higher accuracy”, rather than high accuracy.

To demonstrate CRF_ID method’s advantage over registration methods on experimental data, we show accuracy in several cases (Figure 2A, Figure 3B, Figure 4B, 4C, Figure 6B, 6C, 6E).

c) Similarly, it is often unclear from the figures which strain is being used. For example the strain in Figure 4A-C is not labeled, but it appears to be the same as Figure 3A,B. But if that's the case its unclear why performance would be so much higher in Figure 4B than Figure 3B. Please label which strain is used in which figure subpanel throughout the manuscript, or provide a table.

Figure 4B is demonstration of cell annotation in calcium imaging case. In this case, the candidate list of identities of cells are known a priori in practical experiments (based on promoters used to express GCaMP in cells). Thus, the task is to assign labels from the candidate list to the cells in images hence a smaller atlas is used for automatic annotation. A smaller atlas of candidate identities also implies smaller count noise. In comparison, Figure 3B is demonstration of gene-expression pattern analysis. In this case, no candidate list of cell identities is known a priori. Thus, full brain atlas is used for automatic annotation. Assigning identities to few GFP labeled cells using whole-brain atlas also implies large count noise. Thus, the cell annotation in gene-expression task is more difficult and hence lower accuracy.

Labeling methods and strains in the figures made figures look very busy, so we have moved the information into the legends instead. We have now added strain information in figure legends. Changes made in Figure 2, Figure 3, Figure 4, Figure 5, Figure 6, Figure 2—figure supplement 1, Figure 2—figure supplement 2, Figure 2—figure supplement 3, Figure 2—figure supplement 4, Figure 2—figure supplement 5, Figure 2—figure supplement 6, Figure 4—figure supplement 1, Figure 5—figure supplement 1, Figure 6—figure supplement 1, Figure 6—figure supplement 2, Figure 6—figure supplement 3.

d) Related: Figure 4A prominently displays "calcium imaging," when this is in fact a GFP strain. This misleads the reader

We used GFP strain as a proxy for GCaMP strain to demonstrate cell identification (mentioned in subsection “Cell annotation in multi-cell functional imaging 603 experiments”). To clarify, we have now modified the Figure 4A title to “Mock multi-cell calcium imaging”.

3) The work critically relies on reagents from other groups but could do a better job acknowledging those contributions:a) The Neuropal strain is crucial to the manuscript and provides the majority of the ground-truth data starting with the Results and Figure 2, but Neuropal is first mentioned only towards the very end of the Results section and even then only in the context of Figure 6.

We have cited NeuroPAL strain earlier (Introduction).

b) The strain list should reference publications for strains and alleles used.

We have made the changes (Material and Methods section).

c) Subsection “Cell annotation in gene expression pattern analysis” misleadingly states that "a reporter strain was crossed with pan neuronal red fluorescent protein" but this cross was not performed by the authors, it was performed by another group, published and is publicly available on CGC. This should be fixed.

We have made the changes.

4) More details are needed about the ground truth datasets, including the number of neurons hand-annotated in each of the 9 individuals and their spatial location. (Manual annotation of Neuropal is easiest at the periphery and hardest in center of the ganglia, and so it would be important to know coverage).

Please see our explanation above (#7 in the overall response).

Reviewer #2:Whole brain imaging in *C. elegans* allows us to monitor the activities of a large fraction of worm neurons during behaviors. However, the identity of each cell in the nerve ring remains largely unknown, prohibiting further analysis that integrates structural and functional information of the worm circuit. This problem has been partially amended by manual annotation from experts. However, this is a time-consuming and error-prone solution when one is dealing with a large dataset. We are enthusiastic about this manuscript because the authors take one important step towards automated annotation of neuronal identities. Methods in this work are innovative and attractive. By minimizing energy functions imposed by absolute coordinate differences (unary potential) between images and atlases, as well as differences of the pairwise relationship between neurons, the authors show that their CRF_ID model outperforms SOTA registration model in accuracy, speed, and robustness.We would like to raise several main issues of this manuscript.1) Writing. The main text of the manuscript is brief, dense, technical, and very difficult to understand. I skipped the main text and went directly to the Appendix. This arrangement is not wise for a general audience, especially biologists. In future revisions, please embed the figures in the text, which would save us a lot of time. Here are some specific suggestions:a) The CRF framework can be better explained by building upon physical intuition. The mathematical notations sometimes only cause confusion. For example, is it necessary to introduce clique? I thought that every neuron in their graph model is adjacent to every other neuron. If this is not the case, please describe how the graph model is constructed. You find the note on maximum entropy model is out of the blue (subsection “Structured prediction framework”). If you want to make a deep connection with statistical physics, you better explain it well in the text!!

We appreciate this comment from the reviewer. We have re-written several sections to make the methodology clear (please see our response to #3 in overall comments). We have moved the figures in the text to facilitate readability, we have moved some supplemental text into the main manuscript, and perhaps most importantly, we have added cartoons to facilitate the explanation of what is done. We now emphasize the intuition-building for potential users while maintaining some terminologies often used in graphical model literature to be consistent with this body of literature.

To some of the specific points from the reviewer:

- It is true that every neuron is adjacent to every other neuron in the graph because the underlying graph structure in our model is a fully connected graph. We have clarified this in the text.

- In mentioning maximum entropy model, we did not intend to make a deep connection with statistical physics. Our aim was to establish an analogy of our data-driven atlases to sufficient statistics in maximum entropy models (exponential family models in general). We realized that this is confusing and not adding too much to the understanding of our model or the use of it, so we have removed this text. We now emphasize in the text, with intuitive explanation, how the statistics can be updated based on available data using simple averaging operations, which makes this approach computationally advantageous over other methods where updating is computationally costly.

b) The motivation to build and how to build the data-driven atlases are dismissive in the main text. It is briefly mentioned in subsection “Identity assignment using intrinsic features in CRF_ID outperforms other methods”. But these sentences are extremely difficult to understand without reading the Appendix. The Appendix -- extended methods S1.7, however, needs to be revised and better explained. For example, the equations are not quite consistent, such as in Equations 6 and 25, and the notations are quite confusing. Linking some examples like Figure 2B to S1.7 will help understand the concept.

We appreciate the reviewer’s sentiment here. We have changed the text to clarify these points here (Results; Appendix – extended methods S1.7). Briefly, we explain that the current atlases (including the OpenWorm atlas) is too idealistic; this is substantiated by our data as well as other published work. For example, in OpenWorm atlas, left-right cell pairs are exactly symmetric across AP axis whereas this is rarely true for experimental data. Because data-driven atlases can capture the actual observed positional relationships among cells in biological data and they can be updated (or specialized) for new data when they become available, they should prove to be more useful.

Our paper is motivated by the following important point: that data-driven atlases achieve higher accuracy. One example is to compare accuracy achieved on same datasets using data-driven atlas with test data held out (Figure 2A) and OpenWorm atlas (Figure 2C left panel). Data-driven atlases can be built easily using the code provided for as per the experimental need. For example, for multicell calcium imaging, an atlas can be built only for cells of interest. In contrast, for whole-brain imaging experiments a separate atlas can be built. We echo these points in the Discussion.

We have revised several sections in the Appendix – extended methods:

1) Motivation and intuition behind building the data driven atlas is explained clearly in Appendix -extended methods S1.7 and in main text.

2) Equations and notations are now consistent. Particularly for Equation 6 and 25 we explain how these equations are consistent using Equation 26.

3) We have added additional figure as Figure 1—figure supplement 1 that schematically explains the how the features in the model are updated when data-driven atlas is used compare to static atlas.

c) The prediction accuracies of the model vary from case to case. It was quite low in the scenario of missing neurons. The authors need to state clearly in their paper, under the worst scenario such as whole brain imaging without colors, the prediction accuracy of their methods. This information is now buried somewhere that is hard to find. The prediction accuracies of the CRF model, contingent upon different datasets and different noise levels, would better be a self-contained main figure, not in the supplements. To follow the logic flow of the text, the figure panels that compared CRF with other registration methods would better be introduced later.

Please see our response to the overall point #4. Again, we stress that prediction accuracy depends upon position noise and count noise present in data compared to atlas. The levels of these noise depend on application e.g. different amounts of count and position noise in multi-cell calcium imaging experiments compared to whole-brain imaging. We have characterized several position and count noise cases using synthetic data and shown that CRF model performs better than registration method. In fact, CRF model is very robust in handling count noise compared to registration based methods (Figure 2D, Figure 2—figure supplement 5B).

Also, Figure 2A shows the accuracy using experimental datasets for whole-brain imaging accuracy without colors.

2) Notations. As mentioned above, the symbol consistency is a problem. In the formulation of the model, λ's, trainable parameters in original CRF, have been abused. For the λ appearing in Equations 2/3/4/6/7/8/9/10/32/34/37/46, the author might consider making a clear statement that which of them are trainable and which are hyperparameters. If they are hyperparameters, the setting and tuning strategies need to be disclosed. Also, the keywords "identification" and "annotation" should be considered unified.

We have now clearly described in Appendix – extended methods S1.2, what terms in Equation 6/7/8/9/10 are hyperparameters and methodology for their setting. Explanation for lambdas in Equations 2/3/4 follow from this explanation. Briefly, all lambdas in these Equations are hyperparameters. We also note that (old Equations 37/46 and new Equations 38/49) are not our model and lambdas in these equations are just demonstration of previous methods.

In our model, the trainable parameters are either feature function values, e.g. positional relationship feature values in Equations 6/7/8, or inputs to the feature functions, e.g. distance between cells in Equation 9 and angle vector between cells in Equation 10. These parameters are trainable parameters and are updated while-building data-driven atlas. This is described in Appendix – extended methods S1.7.

3) Loop Belief Algorithms, speed, and stability. The authors should provide details on the computational complexity and the stability of their algorithms and compare it to other registration methods. In their current graph model, does each trial converge and do they all converge to the same solution starting from different initial conditions? If not, how do the authors deal with the discrepancies? Likewise, the authors may want to comment on the memory usage for constructing the data-driven atlases to see if it can be another innovation point comparing to the registration methods.

We thank the reviewer for pointing out potential confusing points. Each trial with a defined initial condition yields a converged solution; in other words, the solution to the optimization problem is deterministic. If, however, different missing neurons (typically randomly assigned) are used, then the solutions may be different. This is the origin of the ranked list of assigned names for each cell.

We do want to acknowledge that the CRF method for each round of solution is computationally intensive (because it is a quadratic method, as opposed to linear methods), but updating the model with new data is cheap (while other methods require re-optimization using all data simultaneously).

Separately as the reviewer pointed out, atlas-building compared to registration methods is indeed a big point. For registration methods, the issue resides in that not only the computational requirements are large, but also that there is no systematic method of building atlas. First, not all images are registered simultaneously due to memory constraints. Second, atlases are built iteratively by blockwise registration of groups of images to a reference image. Without systematic methodology of which image should be chosen as reference image, atlas is biased towards the reference image and the choice of the reference image is often not as “principled”. Another way atlas can be biased is by the order in which blocks of images are registered to the reference frame. Tackling this challenge is an active field of research Wang et al., 2008; Evangelidis et al., 2014. In comparison, in the CRF method, atlas building is unbiased towards a specific image set because there is no concept of reference image and atlas can be built from all images simultaneously because of the cheapness of mathematical operation. This is a major take-away from this work. We echo these points in the Discussion.

4) Other missing information and explanations:a) The authors should post the exact name and its settings of the registration models used for comparison.

This information has been added in Appendix-extended methods S2.1.

b) In Appendix S1.2.1 Unary potentials – Positions along AP axis, as x-y is a 2D plane, we do not see why AP's is always better than LR's and DV's.

Here, we wanted to point out that AP is always better because AP axis is always in the xy plane.

In comparison one of the LR or DV (depending on the orientation of worm) is in the z-direction. Since z-sampling in images is lower in resolution compared to xy-sampling, measurements of neuron positions in z-direction contribute to noise in defining positional relationships.

c) Following the explanation of generating synthetic data (subsection “Generating synthetic data for framework tuning and comparison 464 against other methods”) is a bit hard. The reason for changing the low bound to zero might need further illustrations. Could you please comment on why the prediction accuracy is much lower in synthetic data (Figure 2A)?

To generate synthetic data, we start with OpenWorm atlas and add two types of noise to the atlas to mimic noises present in experimental data. These are position noise (random Gaussian noise that is added to spatial positions of cells) and count noise (random combination of cells that are removed from the data). For both position and count noise, we swept through a range of noise level values for comparison against registration methods. Particularly for position noise we swept from zero (zero noise variance) to upper bound noise (seventy-fifth percentile of the position variance of neurons observed experimentally). I.e. we did not change the low bound to zero, we just expanded the range of parameters swept for comparison against other methods. Results for the different levels of position noise are shown in Figure 2—figure supplement 5C.

Figure 2A is not synthetic data. It is experimental data and predictions were performed using data driven atlas built using experimental data with test dataset held out.

In comparison, Figure 2C left-panel is experimental data and prediction is performed using OpenWorm atlas. Lower accuracy achieved in Figure 2C left panel compared to Figure 2A highlights the fact that positional relationships of cells in OpenWorm atlas are not representative of experimental data. This is also the main motivation of building data driven atlases and a major point of the work.

d) For subsection “Whole-brain data analysis”, the author might consider linking the paragraph to Figure 5, making 0.1Hz a reasonable choice.

We have made the changes.

e) For subsection “Whole-brain data analysis”, "[0.05, 0.05]" seems to be a mistake.

This is the bandwidth parameter used in kerned density estimation method for estimating joint probability distributions of neuron activity and motion speed.

f) Subscripts in Equation 13 are wrong.

Thank you. We have made changes.

g) Providing more annotations and explanations on the meaning of the colors on some worm images (e.g., Figure 4D) would be helpful.

Thank you. We have added color annotations in Figure 4D.

5) Codes on GitHub may need to be slimmed down.

We have moved extra files (e.g. examples and extra functions) to a different folder on GitHub.

Reviewer #3:Linking individual neurons to their anatomical names is a vexing problem that limits efforts to determine patterns of gene expression and neuronal activity at the single-neuron level. Given the landmark Mind of the Worm paper (White et al., 1986) and subsequent analyses of its connectome, many imagine that annotating individual neurons in *C. elegans* is a solved problem. Unfortunately, it is not. This paper presents a computational method for solving this problem and, as such, has the potential to advance neurobiology in *C. elegans*. The approach could, in principle, be adapted to any nervous system from which a computational atlas could be derived following the pipeline laid out in this manuscript. There is much to recommend this study and many opportunities for improvement regarding the presentation of the computational method, its limitations and expectations for application, and for considerations of generalization to other nervous systems.1) Explanation of the computational methodThe computational strategy is difficult to follow in the main text, unless one reviews the appendices. We suggest elaborating the process using more generic terms in the main text. The following specific elements need clarification or additional support- The mixed approach of considering intrinsic and extrinsic similarities between image and atlas is described in a manner that assumes readers are already familiar with this computational strategy. Given the breadth of the eLife readership, the authors would increase the accessibility and impact of the paper by providing readers with examples of what features are intrinsic and which are extrinsic in this case.

We thank the reviewer for this comment. We agree that it is a balance between details and confusion. We have made changes in many parts of the text to address this issue e.g. we have re-written the subsection “Cell annotation formulation using structured prediction framework” to make methodology clear. To clarify the point about intrinsic and extrinsic similarities, we have also included a supplemental figure to illustrate these relationships (Figure 1—figure supplement 1).

- The difference between this approach and previous strategies based on image registration is communicated clearly in the supplement (lines 906-910), but not in the main text.

With new additions, we now describe the conceptual differences between our approach and registration based methods in several places including the Introduction and Discussion, Figure 1—figure supplement 1, Figure 2B. Technical details on differences in objective functions used by these methods are present in Appendix – extended methods S1.9.

- Much is made of the computational efficiency of the CRF_ID approach, but little is provided in the way of data or analysis supporting this claim. The authors could, for example, compare the computational energy/time needed for this approach compared to prior image matching approaches. They could discuss the nature of this computational framework compared to others with respect to computational efficiency.

We thank the reviewer for this comment. This indeed can be confusing. We claim the computational efficiency compared to registration methods only in building data-driven atlases. See the Discussion.

In terms of the run time for optimization, CRF is more computationally expensive because it is a quadratic method as opposed to a linear method. We argue, however, that this is ok because of the gain in accuracy, reduction of bias, and the elimination of the need of choosing references. Please also see responses to related comments (#3) from reviewer 2.

Generality of the methodTo better support the authors claim that this method is easily adaptable for any nervous system, please apply the framework to estimate the optimal number (and/or density) of landmarks as a fraction of the total number of nuclei or density in 3D. This would assist others in applying the framework to other simple nervous systems (e.g. Aplysia, leech, hydra).

Our simulation results show that accuracy increases with more landmarks, i.e. there is no saturation point. Thus, there is no optimal number of landmarks from the computational point of view. Practically, however, there are constraints such as the availability of promoters for genetic control of expression and whether landmarks can be easily distinguished. For instance, if too many densely packed cells are labelled as landmarks such that their identity cannot be easily determined manually, then the automated method cannot use these cells as landmarks. The distinguishability of landmarks depends not only on density in 3D but also on geometry of nervous system. Further it is possible that genetic labelling strategies may be too cumbersome or appropriate promoters may not be available to label all desired landmarks. Therefore, it is not possible to provide an estimate of the optimal number of landmarks in general.

Importantly, we want to note that even without landmarks, CRF model can achieve high accuracy (Figure 2A) by using data-driven atlases. Thus, in our opinion, data-driven atlases are more critical for achieving high accuracy compared to landmarks.

Additionally, it is not clear if the image stacks used to derive data-driven atlases are distinct from the data analyzed for prediction or whether the same data is used iteratively to build atlases and improve prediction accuracy.

As discussed in the overall response (#1), we performed the hold-out exercise – holding out a test dataset for prediction and building positional relationship and color atlases using the remaining datasets. We do not iteratively build atlases using the same data on which prediction is to be tested.

Application of the method to *C. elegans*Specifying the AP/DV/LR axes from raw data is not completely automatic but rather appears to require some user input based on prior knowledge. This is not stated clearly in the text. In addition, it is also not clear how the algorithm handles data from partially twisted worms (non-rigid rotation around the AP-axis). This problem is more likely to be encountered if the images capture the whole worm. Is it possible to allow rotation of the DV and LR axes at different points along the AP axis?

AP/DV/LR axes are defined using PCA on positions of cells in images, and thus automatic. The only user input is to determine the polarity – which end of the worm is anterior (dorsal/left); this is because PCA algorithm does not distinguish AP from PA axis (the axes are -1 multiplications of each other); same is true for DV and LR axes. In our framework, this ambiguity is easily resolved by users clicking on any neuron in anterior (dorsal/left) of the worm. No other extra prior knowledge is needed from users. The Materials and methods section has a description of this.

Our strategy of handling non-rigid rotations about AP axis is to appropriately rotate LR and DV axes plane about AP axis (as described in Appendix – extended methods S1.3). This is done by rotating LR axis using easily identifiable LR neuron pairs in image. In the current framework (dealing with head ganglion cells), the same rotation is applied across all points in AP axis. However, we agree with reviewer that different rotations can be applied across different points along AP axis. This is currently not implemented in current framework.

Specific suggestions for improvement are collected below.1) The method depends on the tidy segmentation of nuclei, but the segmentation strategy is not discussed nor is whether or not nuclear localization of landmarks or unknown markers is essential for accurate annotation.

We have added the description of the GMM method in subsection “Whole-brain data analysis”.

Nuclear localization is not necessary as long as cells can be distinguished (e.g. results in Figure 3B, Figure 4B, 4C for strains without nuclear localization). If cells are not detected accurately this leads to count noise. We have characterized various count noise levels (Figure 2—figure supplement 5).

2) The authors need to explicitly inform readers that the markers in the NeuroPal strains (OH15495, OH15500) and the GT strains are localized to neuronal nuclei. Additionally, the authors need to address (1) whether or not nuclear localization is required for the annotation framework to succeed, and (2) discuss whether or not a cytoplasmic marker would be sufficient, and (3) provide more information on the segmentation algorithm used and recommended for use with the computational framework discuss.

Nuclear localization is not necessary as long as cells can be distinguished (e.g. results in Figure 3B, Figure 4B, 4C for strains without nuclear localization). If cells are not detected accurately this leads to count noise. We have characterized count noise (Figure 2—figure supplement 5).

We use a Gaussian Mixture based segmentation algorithm. We have made changes in the text to clarify this point.

3) The authors use the AML5 strain as a proxy for GCaMP-labeled strains (subsection “Computational workflow for automatic cell identification”) to demonstrate the utility of the annotation tool for identifying cells used for calcium imaging. This strain differs from the NeuroPal strain expressing GCaMP is several respects that are not discussed by the authors, leaving readers to wonder about the utility of AML5 as a proxy and to question whether or not the success of cell identification in this strain is truly generalizable. Two main differences exist: 1) In AML5, GFP is expressed in both the nucleus and cytoplasm but in in OH15500 GCaMP6s is tagged with an NLS; 2) The basal fluorescence of GCaMP6s, but not GFP depends on basal calcium concentrations-this fact may introduce additional detection noise/variance into the annotation problem. The authors need to discuss how these differences affect (or don't) the ability to generalize from success with the proxy strain to success with NLS::GCaMP6s expression.

To clarify, according to Nyugen et al., 2016.

1) Nuclear localization: As discussed earlier as well (response to #5 overall comments), CRF_ID model works regardless of marker localization, as long as the cells can be separated/distinguished. If the cells cannot be separated, this leads to count noise. We have demonstrated superior performance of CRF_ID in handling count noise.

2) Cells with only basal fluorescence: Basal fluorescence only could lead to cell detection errors, thus leading to count noise. We have characterized for various amounts of count noise levels for both experimental data (Figure 4C) and synthetic data (Figure 2D, Figure 2—figure supplement 5B), and shown that CRF model performs better than registration methods. In practice, for GCaMP videos, data from multiple frames can be used to get a superset of cells (thus removing count noise) before annotation with CRF_ID. This would directly address the basal fluorescence issue.

3) AML5 example (Figure 4B, 4C) is to demonstrate a use case different than whole-brain imaging (fast annotation of cells in multi-cell calcium imaging videos). To demonstrate success in wholebrain imaging, we show prediction for whole-brain datasets (Figure 2A, Figure 6C).

4) Please clarify the nature of the output from the code that implements the framework. In other words, a reader should learn from the paper what kind of output file they might receive if they attempted to apply the tool to their own images (annotated image with top prediction candidate, data structure with all predictions and respective probabilities, etc).

We run the code in two modes:

Mode 1 – Single run will give a deterministic prediction for each cell.

Mode 2 – multiple runs taking missing cells into account. Each run will generate predicted identities of cells in that run. The results from various runs are used in subsequent step to generate top candidate list for each cell with label consistency score for each candidate.

We have clarified the text about these modes of running the algorithm (subsection “Computational workflow for automatic cell identification”).

(5) Terminology – There are a number of general computational terms and specific terms relevant to this problem that are not defined for the reader (listed below). Providing readers with implicit or explicit definitions of these terms will improve clarity.5(a) "count noises" – used to represent variation in the number of nuclei detected in a given image stack, but not defined at first use. Please provide conditions under which the number of nuclei detected might vary along with defining this term. Additionally, this is a singular concept and should be written as "count noise" not "count noises" (by analogy to "shot noise").5(b) There several other instances in which "noises" is used where "noise" is meant.

Count noise (defined in the Introduction and on) is not variation in the number of nuclei detected. It is the difference in the number of nuclei detected in the image, compared to the number of cells present in atlas to which image is to be matched. For example, if 100 cells are detected in an image whereas 200 cells are present in the atlas (i.e. 200 labels) to be used, count noise would be equal to (200-100)/200 = 50%.

The practical reasons for count noise is in the mosaicism of the expression of the genetically encoded marker/reporter, the incomplete coverage of the marker/reporter (e.g. “pan-neuronal” not covering every neuron), errors in cell segmentation method due to dim cells or tightly clustered cells etc. These reasons are given in the same paragraph.

We have changed all mentions of “count noises” to “count noise”.

5(c) "intrinsic similarity" – which seems to represent pairwise positions in 3D in the image and the atlas.5(d) "extrinsic similarity" – not clear.

We have included Figure 1—figure supplement 1 to clarify these points.

(6) (Subsection “Identity assignment using intrinsic features in CRF_ID outperforms other methods”) "More importantly, automated annotation is unbiased and several orders of magnitude faster than manual annotation by researchers with no prior experience."(a) Unbiased – It is a common misconception that computational strategies are unbiased. For this claim to be rigorously accurate, the authors would need to demonstrate that each and every nucleus was equally likely to be identified correctly by the automated annotation algorithm. For instance, it is easy to imagine that some nuclei, such as those closer to a landmark, are more likely to identified correctly and with less uncertainty than those further away. If this were true, then automated annotation has a systematic bias in favor of nuclei closer to landmarks and a systematic bias against nuclei further away from landmarks. And would, therefore be biased – even though no human supervision is involved. The authors should back this claim up with data or clarify that what they mean is that automated annotation is an improvement over (biased and subjective) human annotation.

We agree with the reviewer that if there are biases in making the atlas, then those biases will be reflected in the predictions. For example, if no information is present for some cells in the atlas, then it may be expected that prediction accuracy of those cells will be lower, compared to cells for which a lot of information is present in the atlas. However, this is true for any machine learning method, as all machine learning methods are as (un)biased as the training data. (The same perhaps can be said about human curation.)

Our use of the terminology is with respect to maximum-entropy models. With log-linear parameterization of features, CRF model is similar to maximum-entropy models. Further, maximum-entropy models produce a maximally unbiased (maximum entropy) probability distribution over labels subject to some constraints, which in our case are observed positional relationships in the atlas.

In addition, the model is unbiased in the sense that it predicts identities using an atlas (that may be built with fully or partially annotated datasets contributed by various researchers). Thus, CRF_ID method combines the knowledge of all researchers in the form of positional relationship atlas and then predicts identities that maximally tries to satisfy the atlas. Thus, it removes individual biases in cell identity annotation. We have clarified this point in text.

(b) Faster -The authors are free to speculate about the relative speed of automated vs. manual annotation by an inexperienced researchers in the discussion. By writing this as a result, however, data are needed to back it up.

We only included discussions on this point in the Discussion section. Our point is that CRF_ID can be run automatically without direct input after the models are trained, so the use is simple.

(7) (Materials and methods) From the description of how GT290 and GT298 were made, it seems that both constructs include 2x NLS fragments, but the genotype description under reagents lacks that annotation and most, if not all other references to this strain neglect to mention the NLS.

We apologize for the confusion; we have clarified the genotypes. The strains are nuclear localized.

Presentation concerns that need attention: There are three instances of undisclosed data re-use among the figures that need to be addressed. (1) The image in Figure 4A is identical to the green channel of the image shown in Figure 3A; (2) Some of the data shown in Figure 2—figure supplement 4A – Part of the data shown here is also shown in Figure 2C; (3) Figure 4C – Please indicate if the same number of cells were removed in each run. Also, it appears that this data also may have been used in Figure 4—figure supplement 1. If so, please disclose data re-use.

Same strain, AML5, was used for demonstration of cell annotation in gene expression and multicell calcium imaging cases. For gene expression case (Figure 3A) both red and green channels were used for cell identity prediction whereas for multi-cell calcium imaging case (Figure 4A) only green channel was used. We have replaced Figure 4A with a new image.

Data in Figure 2C (right panel) and Figure 2—figure supplement 5A come from the same experiment that is predicting cell identities in experimental datasets using OpenWorm atlas and comparing accuracy across methods. Figure 2C (right panel) shows comparison for only top labels predicted by methods, in comparison Figure 2—figure supplement 5A shows comparison for top 3 and top 5 labels as wells.

Figure 2C is indeed related to Figure 2—figure supplement 5A. Figure 2C highlights the most important comparisons. For the sake of easy side-by-side comparisons, we left Figure 2—figure supplement 5A as is, and added in the legend that part of data are replotted in Figure 2C for simple comparison.

For Figure 4C, the same fraction (3 out of 16) but a different combination of randomly selected cells was removed in each run. We have mentioned this in Figure panel now. Figure 4—figure supplement 1 shows additional results for different numbers of cells removed (2, 4, and 5). We have now mentioned in the legend of Figure 4—figure supplement 1 that part of data are replotted in Figure 4C.

[Editors' note: further revisions were suggested prior to acceptance, as described below.]

Essential Revisions:(1) The authors performed additional computational experiments by holding out some of the data used to create atlases – an important and general strategy for computational solutions that depend on real-world data. The authors present a comparison of this approach for reviewers, but not for explicitly for readers. These analyses and plots must be included in the final version main text.

We have included the new data in the revised manuscript in the last round (Figure 2A, Figure 6B, C, E, Figure 6—figure supplement 1, Figure 6—figure supplement 2, Figure 6—figure supplement 3).

(2) The authors should state clearly the computational speed of their algorithms and compare it with the registration methods. The inference and subsampling procedures (~ 1000 times?) on the data-driven atlas are time-consuming and computationally expensive.

We have now included a new figure (Figure 2—figure supplement 7) comparing the optimization runtimes of CRF and registration methods. A discussion is added.

Reviewer #2:Main Text(1)Thanks for sending us a much better written manuscript. The authors should consider replotting some of their figures, which can be difficult to read.(a) The neuron names in Figure 2F and Figure 2—figure supplement 6C are difficult to read. Is there a new way to plot? When I wanted to zoom in to have a better understanding, they caused my Preview crashed multiple times…

We have increased the size of Figure 2F and the Figure 2—figure supplement 6C. They are now available on github. We enlarged them to page width but perhaps still not clear enough to see details. Here in the figures we mainly want to show the general shape of the data. For readers interested in the exact details, they may download the figure from github.

(b) The meaning of the error bars throughout the manuscript are not well explained.

Description of boxplots and error bars is included now in all figure legends.

(2) The authors should state clearly the computational speed of their algorithms and compare it with the registration methods. The inference and subsampling procedures (~ 1000 times?) on the data-driven atlas are time-consuming and computationally expensive.

We have now compared the runtimes of CRF_ID framework with registration methods. The discussion is added in subsection “Identity assignment using intrinsic features in CRF_ID outperforms other methods” and on and results are added as Figure 2—figure supplement 7.

(3) Is there a way to make the data-driven atlas publicly available in addition to the raw annotated datasets (n=9)?

The datasets are all on our github. Positional relationship atlases are available here https://github.com/shiveshc/CRF_Cell_ID/tree/master/Runs/Data_driven_atlases/PositionalRelationship_ atlases, and Color atlases are available here https://github.com/shiveshc/CRF_Cell_ID/tree/master/Runs/Data_driven_atlases/Color_atlases.

Rebuttal(1) Pipeline (Major Point#6):I do not quite understand the necessity of single or multi-run, which can be combined into one inference process. Consider the following scenario in a freely behaving animal. At one time, the region to be annotated consists of N neurons. At a different time, a new neuron (now N+1) is squeezed into the region. It is probably always true that the number of neurons in the data is different from that in the atlas.

We agree with the reviewer that the number of neurons in the dataset is likely to be different from that in the atlas in many experimental conditions. There are, however, use-cases when the count-noise is zero (i.e. the number of cells is exactly the same as in the available labels). For instance, we talk about multicell imaging scenario (Figure 4A/B) – in some situations where multiple cells are labeled and the identity set of the cells is well defined (e.g. well-known well-behaved promoters).

(2) Color and Robustness and Evaluation Metrics (Essential revisions #1, #4, #6):It seems that the leverage of color information for annotation is quite limited in the revised manuscript. The procedures for properly normalizing the color distribution are also complicated. I also found that the following sentence was incorrect. "Further, when leave-one-out atlases are used for both positional relationship and color (Figure 6—figure supplement 1), the accuracy achieved by top labels is marginally greater than the accuracy achieved by using leave-one-out positional relationship only (Author response image 1)". It was actually significantly lower if I read the figures correctly. This leaves the question how we shall correctly assemble the data-driven atlas efficiently.

We apologize for the confusion. The statement was comparing the yellow bar in Figure 6—figure supplement 1 (leave-one-out for both position and color, which is at 74.5%) and the first column orange bar in Author response image 1 (leave-oneout for position only, which is at 73.4 %). That is why we say including color is only marginally better. The reviewer makes a good point, which we agree, that not all information is additive – either position or color alone can carry quite a bit of information, but in this case, together, they did not provide a whole lot more extra information. With color normalization and using ensemble of color atlases the leave-one-out accuracy is 80.7% for top labels.

Reviewer #3:The revision is very responsive to the summary and specific reviewers' critiques.There remain some awkward phrasing and missing details to address. These are noted in order to make the text accessible to a broader audience and to increase the impact of the study both within and beyond the community of *C. elegans* researchers. These include:(1). The use of the plural "noises" when the authors probably mean either "noise" in the singular e.g. “Prediction accuracy is bound by position and count 316 noise in data”, replace "amount of various noises" with "amount of noise contributed from various sources". This latter example combines both the concept of the magnitude of noise and the diverse types of noise.

Changes are made.

(2) Shorthand terms or jargon. For example, when the text reads "the position of cells AIBL" what is meant is "the position of the cell bodies of AIBL". Similarly, replace "neurons" with "neuronal cell body" or "neuronal somata".

Changes are made.

(3) *C. elegans* strains – in response to critiques of the initial submission, the authors have clarified the origin of the transgenic strains used in the study. Thank you! Please also correct the reference to AML70 which is not included in the table of strains. Based on context, it looks like the authors are referring to AML32. Please ensure that this is corrected.

We have updated the strains reference table and description of making strains in the subsection “Construction of landmark strains”.

(4) Figure 5, panel E. Please refer to Figure 5—figure supplement 1 panel C noting the cells with non-zero weights in SPC1, SPC2, SPC3

Changes are made.

(5) Figure 5 panel H: Indicate the origin of the motion trace shown. It appears to belong to one of the posterior cells shown in Panel F. Please provide a rationale for this choice – since the traces in Panel F certainly indicate that motion signals differ among cells.

We have now indicated the cell whose motion trace was used for mutual-information (Figure 5G) and cross-correlation analysis (Figure H, I) in Figure 5H. We chose one of the posterior cells because velocity traces of those cell were in phase with neuron activity. We agree with reviewer that motion signal among cells are different in terms of phase shifting and magnitude; this could be because the motion of animal is restricted in the device.